# Do Contemporary Causal Inference Models Capture Real-World Heterogeneity? Findings from a Large-Scale Benchmark

**Haining Yu**
Amazon
`hainiy@amazon.com`

**Yizhou Sun**
Department of Computer Science
University of California, Los Angeles
`yzsun@cs.ucla.edu`

## Abstract

We present unexpected findings from a large-scale benchmark study evaluating Conditional Average Treatment Effect (CATE) estimation algorithms, i.e., CATE models. By running 16 modern CATE models on 12 datasets and 43,200 sampled variants generated through diverse observational sampling strategies, we find that: (a) 62% of CATE estimates have a higher Mean Squared Error (MSE) than a trivial zero-effect predictor, rendering them ineffective; (b) in datasets with at least one useful CATE estimate, 80% still have higher MSE than a constant-effect model; and (c) Orthogonality-based models outperform other models only 30% of the time, despite widespread optimism about their performance. These findings highlight significant challenges in current CATE models and underscore the need for broader evaluation and methodological improvements.

Our findings stem from a novel application of *observational sampling*, originally developed to evaluate Average Treatment Effect (ATE) estimates from observational methods with experiment data. To adapt observational sampling for CATE evaluation, we introduce a statistical parameter, $Q$, equal to MSE minus a constant and preserves the ranking of models by their MSE. We then derive a family of sample statistics, collectively called $\hat{Q}$, that can be computed from real-world data. When used in observational sampling, $\hat{Q}$ is an unbiased estimator of $Q$ and asymptotically selects the model with the smallest MSE. To ensure the benchmark reflects real-world heterogeneity, we handpick datasets where outcomes come from field rather than simulation. By integrating observational sampling, new statistics, and real-world datasets, the benchmark provides new insights into CATE model performance and reveals gaps in capturing real-world heterogeneity, emphasizing the need for more robust benchmarks.

## 1 Introduction

Conditional Average Treatment Effect (CATE) models are increasingly used to answer causal inference questions in fields such as medicine, economics, and policy. But how well do these models capture real-world heterogeneity? We present unexpected findings from a large-scale benchmark study on contemporary CATE estimation algorithms. Based on 43,200 sampled variants derived from 12 datasets and evaluated across 16 CATE models, we find: (a) 62% of CATE estimates have a higher Mean Squared Error (MSE) than a trivial estimator that consistently predicts zero effect, rendering them ineffective; (b) in cases where at least one useful CATE estimate exists, 80% have higher MSE than a constant-effect estimator; and (c) orthogonality-based models outperform other models only 30% of the time. These findings raise important questions about the current models' ability to fully reflect the complexities of real-world heterogeneity.

Rather than introducing new models, our benchmark study focuses on evaluating existing CATE estimation models. The past decade has seen significant advancements in CATE estimation, with new methods emerging from statistics, econometrics, and machine learning (see Chernozhukov et al. (2017); Athey & Imbens (2016); Kennedy (2023); Shalit et al. (2017); Alaa & van der Schaar

(2017); Chernozhukov et al. (2023); Künzel et al. (2019)). Widely available and easy-to-use tools like EconML (Battocchi et al., 2019) and DoubleML (Bach et al., 2022) have made CATE models accessible to users with minimal expertise, leading to their broad application in high-stakes business and scientific decisions, where accuracy is critical.

Understanding real-world model accuracy is challenging because CATE estimation models lack access to ground truth CATE. To compensate, these models rely on estimates of potential outcomes and/or propensity, known as *nuisance functions*. As a result, models face two key risks - *inaccurate potential outcome estimation* and *inaccurate propensity estimation* - forcing difficult trade-offs. When potential outcome risk approaches zero, the ground truth CATE estimate can be recovered, which is the core idea behind S-learner and T-learner models (Künzel et al., 2019). Conversely, when propensity is known, the Horvitz-Thompson estimator (Horvitz & Thompson, 1952) provides unbiased CATE estimates. Most contemporary CATE models attempt a hybrid approach, estimating both potential outcomes and propensities to generate the final CATE estimate. The error of nuisance estimates impacts the accuracy of CATE estimates. To minimize the effect of errors, modern CATE models employ loss functions with robustness guarantees, ensuring that errors in nuisance estimates do not have a first-order effect on the CATE estimate. Such guarantees come from a family of closely related theories, including but not limited to doubly robustness (Kennedy, 2023), Neyman Orthogonality (Chernozhukov et al., 2017; Nie & Wager, 2020; Foster & Syrgkanis, 2023), and Influence function (Alaa & Van Der Schaar, 2019). We refer to such models from these theories as *orthogonality-based models*. These theories rely on stringent assumptions regarding smoothness, Lipschitz continuity, sparsity, convexity, and orthogonality of the true potential outcomes, propensities, and loss functions. While mathematically elegant, their effectiveness is typically validated using simulation data.

Despite theoretical guarantees, evaluating CATE models in practice remains challenging. CATE model *evaluation* methods aim to assess the accuracy of CATE estimation models but encounter the same challenges as the models they evaluate. Ideally, we would compute the MSE for CATE estimates and rank estimators by their MSE relative to the ground truth CATE, a process known as *oracle ranking*. However, in most real-world datasets, only the factual outcome is observed, not the counterfactual, making the ground truth CATE unknown unless the counterfactual generation process is explicitly known. It is widely accepted that without observing both factual and counterfactual outcomes, ground truth CATE cannot be computed, making it difficult to select the most accurate CATE estimators (Curth & van der Schaar, 2021; 2023; Neal et al., 2021; Mahajan et al., 2023). Without understanding CATE estimate accuracy, users cannot effectively evaluate the quality of estimators or the risk of inaccurate estimates.

To address challenge of evaluating CATE models without ground truth, two main approaches are used, each with drawbacks. The first approach simulates potential outcomes and ranks CATE estimators by their MSE on semi-synthetic datasets, effectively removing the risk of potential outcome estimation (see Hill (2011); Shalit et al. (2017); Künzel et al. (2019); Diemert et al. (2021)). But doubts remain about whether promising simulation results translate to equally promising outcomes in real-world cases. The second approach ranks CATE estimators using proxy loss functions (see details in Section 2), particularly those with doubly robust or Neyman orthogonal properties. This approach attempts to manage both potential outcome and propensity estimation risks by exploiting orthogonality properties in the loss. Doubts remain about proxy loss functions, particularly concerning their stringent assumption requirement, finite sample property, and self-serving bias (Curth & van der Schaar, 2023). The last bias occurs when a CATE estimator is evaluated using a loss function sharing common assumptions. For example, if we use R-loss to score CATE estimators and find R-learner (which optimizes R-loss) performs best, we cannot determine whether R-learner has indeed the lowest MSE or simply shares assumptions with R-loss. This situation is analogous to a sports player also acting as the referee. Note that, this situation is unique to causal inference where ground truth is missing in test dataset, making it impossible to compute MSE there like one would do in supervised learning. To summarize, from a risk perspective, current CATE evaluation methods either try to remove potential outcome estimation risk, or manage both outcome estimation and propensity risks. Meanwhile, few work explores removing the risk of propensity estimation.

Given the limitations of current CATE evaluation methods, we seek new approaches that can assess CATE estimator performance on real-world heterogeneous data while relying on fewer and simpler assumptions. Drawing from the risk discussion, we ask: *could eliminating the risk of propensity estimation be the solution?* At first glance, this seems counter-intuitive. Observational methods inherently work with unknown propensity, thus the risk cannot be removed. This is where the method

of observational sampling comes in (LaLonde, 1986; Gentzel et al., 2021). In observational sampling, an observational dataset is created by sampling from an Randomized Controlled Trial (RCT) dataset through a carefully designed process that introduces selection bias. This approach allows CATE estimators to be trained on the observational sub-sample (with propensity estimation risk) while their performance is evaluated using the full RCT data (without propensity estimation risk).

The history of observational sampling dates back as long as the field of causal inference itself. For instance, LaLonde (1986) used it to construct the IHDP dataset for evaluating ATE estimates from observational data. Since then, large RCT datasets have become more available in certain domains (Gordon et al., 2019; 2022). However, few researchers (except Gentzel et al. (2021)) explore observational sampling for CATE evaluation, as mainstream research often relies on small semi-synthetic datasets, which have the drawbacks discussed earlier.

Recognizing the untapped potential of observational sampling, we hypothesize that it offers opportunities to develop new statistics for identifying the most accurate CATE estimators and assessing their ability to capture real-world heterogeneity. This forms the central hypothesis of our research. To rigorously test this, we aim to develop new theoretical results and create a benchmark procedure using observational sampling for CATE evaluation.

This paper offers three key contributions to the field of CATE evaluation:

1. **Benchmark Findings:** Our primary contribution is new findings from the large-scale benchmark study, evaluating sixteen contemporary CATE estimation methods across 43,200 variants sampled from 12 unique datasets with real-world outcomes. As noted in the opening paragraph, these findings reveal current CATE models' limitations in capturing real-world heterogeneity. They highlight the need for further research.

2. **New Evaluation Metrics:** Our second contribution is new CATE evaluation metrics. We define a new statistical parameter, $Q$, which equals MSE minus a constant, and develop a family of statistics, collectively called $\hat{Q}$, that converge to $Q$. We prove that, for RCT data, $\hat{Q}$ is an unbiased estimator to $Q$ and achieves a $O(1/\sqrt{N})$ asymptotic convergence rate. Additionally, we introduce a control-variates-based framework to reduce the variance of $\hat{Q}$, showing that common CATE estimation losses, such as R-loss and DR-loss, are special cases of this framework.

3. **Novel Evaluation Procedure:** Our third contribution is a novel CATE evaluation procedure based on observational sampling and the newly developed $Q$. This method allows for the training of CATE estimators on observational sub-samples and evaluates their performance using $\hat{Q}$ on the full RCT dataset. Unlike previous benchmarks, our approach does not rely on simulated potential outcomes, addressing concerns about real-world heterogeneity and mitigating the risk of self-serving bias.

## 2 PRELIMINARY AND RELATED WORK

### 2.1 PRELIMINARIES

We formalize our problem setting using the potential outcomes framework Rubin (2005). All notations can be found in Table 2 in Appendix A. Let $(X, T, Y)$ be a tuple of random variables following distribution $\Pi$, where $X \in \mathcal{X}$ is the pre-treatment covariates, $Y \in \mathcal{R}$ is the observed outcome, and $T \in \{0, 1\}$ is the treatment assignment. Each tuple is associated with two potential outcomes $Y(0)$ and $Y(1)$. However, we observe only the outcome associated to the factual treatment $T \in \{0, 1\}$, $Y = Y(T)$. We denote $\mu^{(0)}(x) = \mathbb{E}[Y(0)|X = x]$ and $\mu^{(1)}(x) = \mathbb{E}[Y(1)|X = x]$ as the expected potential outcome functions given the covariate $x$, and $e(x) = \Pr(T = 1|X = x)$ as the *treatment propensity function*. The *Conditional Average Treatment Effect (CATE)* is then defined as: $\tau(x) = \mathbb{E}[Y(1) - Y(0)|X = x] = \mu^{(1)}(x) - \mu^{(0)}(x)$. The *Average Treatment Effect (ATE)* is then $\tau_{ATE} = \mathbb{E}_X[\tau(X)]$.

Let $D$ denote a dataset with $N$ i.i.d samples $\{(x_n, t_n, y_n)\}$ drawn from $\Pi$. When the treatment assignment is independent of covariates, i.e., $T \perp X$, we call such dataset a RCT dataset and define the constant treatment propensity $E_1 = e(x) = \Pr(T = 1|X = x) = \Pr(T = 1)$ and $E_0 = 1 - E_1$.

The goal of *CATE estimation* is to train a CATE estimator $\hat{\tau}(x)$ using an observational dataset that approximates $\tau(x)$ as close as possible. Given a trained CATE estimator $\hat{\tau}(\cdot) : \mathcal{X} \to \mathbb{R}$, the goal

of *CATE evaluation* is to evaluate the quality of $\hat{\tau}(\cdot)$ by comparing it to $\tau(\cdot)$. One commonly used evaluation criterion is its MSE, also known as *Precision in Estimating Heterogeneous Effects (PEHE)* (Hill, 2011): $P(\hat{\tau}(\cdot)) = \mathbb{E}_X[(\tau(X) - \hat{\tau}(X))^2]$, which is a functional that maps an estimator $\hat{\tau}$ to a non-negative real number. Note that, MSE can be calculated only when $\tau(\cdot)$ is available, which requires the observation of both factual and counterfactual outcomes. To ensure that effects are identifiable from observational data, we rely on the standard ignorability assumptions (Rosenbaum & Rubin, 1983):

**Assumption 2.1.** (i) Consistency: for a sample with treatment assignment $T$, we observe the associated potential outcome, i.e. $Y = Y(T)$. (ii) Unconfoundedness: there are no unobserved confounders, so that $Y(0), Y(1) \perp T | X$. (iii) Overlap: treatment assignment is non-deterministic, i.e., $0 < \Pr(T = 1 | X = x) < 1$.

## 2.2 RELATED WORK

**CATE estimation.** There exist many methods to construct CATE estimator $\hat{\tau}(x)$. Here we cover three most popular strategies. The first *outcome prediction* strategy predicts potential outcomes $\mu^{(0)}(x)$ and $\mu^{(1)}(x)$, and uses their difference as the CATE estimate, i.e., $\hat{\tau}(x) = \tilde{\mu}^{(1)}(x) - \tilde{\mu}^{(0)}(x)$. [1] This approach essentially minimizes $\mathcal{L}_{OP}(\tilde{\mu}^{(0)}, \tilde{\mu}^{(1)}) = \frac{1}{N} \sum_n \left(y_n - \tilde{\mu}^{(t_n)}(x_n)\right)^2$ by any regression model. Examples include S-learner, which regress $Y$ on $X$ and $T$, and T-learner, which regress $Y$ on $X$ for $T = 0$ and $T = 1$ separately (Künzel et al., 2019). Solving the minimization problem of $\arg\min_{\tilde{\mu}^{(0)}, \tilde{\mu}^{(1)}} \mathcal{L}_{OP}$ yields the estimator of $\hat{\tau}_{OP}(x) = \tilde{\mu}^{(1)}(x) - \tilde{\mu}^{(0)}(x)$. For observational datasets, learning a shared representation $\phi(x)$ for both treatment groups can improve CATE estimates. This approach regresses $Y$ on $\phi(X)$ to estimate potential outcomes; Johansson et al. (2018) provides bounds on generalization error. Dragonnet (Shi et al., 2019), a variation of this approach, learns the representations of $\mu^{(0)}(\phi(x))$, $\mu^{(1)}(\phi(x))$, and $e(\phi(x))$ using a three-head neural network. The loss function for dragonnet is $\mathcal{L}_{RL}(\tilde{\mu}^{(0)}, \tilde{\mu}^{(1)}) = \frac{1}{N} \sum_n \left[(y_n - \tilde{\mu}(t_n, \phi(x_n)))^2 + \lambda \mathrm{BCE}(t_n, \tilde{e}(\phi(x_n)))\right]$. Solving the problem of $\arg\min_{\phi, \tilde{\mu}^{(0)}, \tilde{\mu}^{(1)}, \tilde{e}} \mathcal{L}_{RL}$ yield the estimator $\hat{\tau}_{RL}(x) = \tilde{\mu}^{(1)}(\phi(x)) - \tilde{\mu}^{(0)}(\phi(x))$.

The second *semi-parametric regression* strategy estimates CATE based on transformed outcomes. When the true model is $Y = f(X) + T\tau(X) + \epsilon$, it can be rewritten as $Y - m(X) = (T - e(X))\tau(X) + \epsilon$ where $m(x) = \mathbb{E}[Y | X = x] = f(x) + e(x)\tau(x)$. This reformulation then estimates $\tau(x)$ by regressing transformed outcome $Y - m(X)$ on transformed covariate $T - e(X)$. Robinson (1988) and subsequent work show that these estimates are $\sqrt{N}$-consistent. Historically, this approach carried different names such as residual-on-residual, partialling-out estimators, Double Machine Learning (Chernozhukov et al., 2017), and Neyman orthogonality (Newey, 1994), to name a few. In its simplest form, the loss function is $\mathcal{L}_R(\hat{\tau}) = \frac{1}{N} \sum_n \left[((y_n - \tilde{m}(x_n)) - (t_n - \tilde{e}(x_n))\hat{\tau}(x_n))^2\right]$ with plug-in estimates $\tilde{m}(x)$ and $\tilde{e}(x)$; this is called R-loss in Nie & Wager (2020). Minimizing $\mathcal{L}_R$ yields the estimator $\hat{\tau}_R(x)$. A notable extension of this method is causal forests (Athey et al., 2018), which adaptively partition the data to maximize the difference between CATE estimates from different tree partitions, improving the accuracy of the estimates.

The third approach is *Inverse-Propensity Weighting* (IPW). IPW is based on the Horvitz-Thompson estimator, defined as $\eta(x, t, y) = \left(\frac{t}{e(x)} - \frac{(1-t)}{1-e(x)}\right) y$. When the propensity score $e(x)$ is known, this estimator is an unbiased estimate of $\tau(x)$ (Horvitz & Thompson, 1952). That is, $\mathbb{E}[\eta(X, T, Y) | X = x] = \tau(x)$. However, IPW is known to have high variance (Robins et al., 1994). To reduce variance, Kennedy (2023) suggests constructing doubly robust loss $\mathcal{L}_{DR}(\hat{\tau}) = \frac{1}{N} \sum_n [\eta(x_n, t_n, y_n) + \gamma(x_n, t_n) - \hat{\tau}(x_n)]^2$, where $\gamma(x, t) = \left(1 - \frac{t}{\tilde{e}(x)}\right) \tilde{\mu}^{(1)}(x) - \left(1 - \frac{1-t}{1-\tilde{e}(x)}\right) \tilde{\mu}^{(0)}(x)$ is a shorthand function with plug-in estimates $\tilde{\mu}^{(1)}, \tilde{\mu}^{(0)}$ and $\tilde{e}(x)$ that we will use later. Solving the problem of $\arg\min_{\hat{\tau}} \mathcal{L}_{DR}$ yields the Doubly Robust estimator $\hat{\tau}_{DR}(x)$.

When explaining the strategies mentioned above, we omit the details for sample splitting and regularization to improve readability. Modern implementation of these methods use standard ML

---

[1]Let $f$ be a ground truth function defining the data generation process; $f$ is often unobservable. We use $\hat{f}$ to represent the main estimator, and $\tilde{f}$ to represent the plug-in.

regression and classification models as components. In different literature components are called *plug-ins*, *base learners*, or *nuisance functions*.

**CATE model evaluation.** There is extensive literature on CATE model evaluation. In principle, any score function $S(\hat{\tau})$ can rank and evaluate CATE estimators. However, the key question is whether the ranking helps users identify the best estimators for their needs. While this paper focuses on score functions that rank by MSE, it's useful to first explore the broader landscape of score functions.

The first category of score functions includes hypothesis testing statistics. Studies such as Chernozhukov et al. (2023); Bartolomeis et al. (2024); Hussain et al. (2023) develop statistics to detect heterogeneity, unobserved confounding, or transportability. These statistics can rank CATE estimators but do not guarantee finding the estimator with smaller MSE; this makes them unsuitable for general CATE evaluation. For instance, the BLP statistic from Chernozhukov et al. (2023) is ineffective when the CATE estimator has small heterogeneity. The second category covers rank-based metrics, commonly used in uplift modeling and CATE calibration. Examples include Radcliffe (2007); Dwivedi et al. (2020); Yadlowsky et al. (2023); Imai & Li (2021); Xu & Yadlowsky (2022). While useful in specific contexts, these metrics lack a direct connection to MSE. For example, the Qini index (Radcliffe, 2007) assigns the same score to two estimators $\hat{\tau}(x)$ and $\hat{\tau}(x) + 1$, even if their MSE differs. The third category consists of score functions that compare CATE estimators to ATE estimates from experimental data, as discussed in Gentzel et al. (2021) and related work.

Now, let us turn to methods designed to find model with smallest MSE; see Curth & van der Schaar (2021; 2023); Mahajan et al. (2023); Neal et al. (2021) for reviews. The most common approach is simulation using semi-synthetic datasets, such as IHDP and Jobs (Hill, 2011; LaLonde, 1986), where simulated potential outcomes make MSE calculation feasible. Extensions of this approach include generative models for synthetic data (Neal et al., 2021; Athey et al., 2020; Parikh et al., 2022). As noted before, simulation misrepresents real-world heterogeneity, the precise risk we want to evaluate.

Another strategy is hold-out validation, which constructs CATE estimation loss functions on test datasets. For instance, by estimating potential outcomes $\tilde{\mu}^{(0)}(x)$ and $\tilde{\mu}^{(1)}(x)$, one can compute the proxy $\tilde{\tau}(x)$ and calculate the MSE as $\mathcal{L}_{PL}(\hat{\tau}) = \frac{1}{N} \sum_n (\hat{\tau}(x_n) - \tilde{\tau}(x_n))^2$. Other loss functions can also apply; theorem 15.2.1 in Chernozhukov et al. (2024) is an example. The primary risk with this approach is still self-serving bias: model may unfairly benefit from being judged by losses that favor its own design. Furthermore, hold-out validation introduces complexity, as there are numerous choices: base regression models, hyperparameters, regularization, sample-splitting, and bias correction techniques (e.g., Neyman orthogonality (Newey, 1994), influence functions (Alaa & Van Der Schaar, 2019)). These choices further aggravate the self-serving bias.

Recent studies (Curth & van der Schaar, 2023; Mahajan et al., 2023; Neal et al., 2021; Athey et al., 2020; Parikh et al., 2022), largely based on semi-synthetic datasets, have evaluated various CATE evaluation criteria, including $\mathcal{L}_{OP}$, $\mathcal{L}_R$, $\mathcal{L}_{DR}$, and $\mathcal{L}_{PL}$. However, consensus is yet form.

## 3 THE PROPOSED EVALUATION METRIC $Q$

As discussed, existing CATE evaluation criteria fall short of selecting the best models, particularly when it comes to capturing real-world heterogeneity. This makes it difficult for practitioners to choose the most suitable models and prevents the community from making substantial breakthroughs in CATE estimation. We propose to break this dilemma by introducing a new statistical parameter $Q$, equal to MSE minus a constant, and a family of statistics $\hat{Q}$ that can be computed from real-world data. We show that, when propensity is known (as in observational sampling), $\hat{Q}$ is an *unbiased* estimator to $Q$, thus asymptotically preserving the same order as MSE when ranking different CATE models. We also discuss the generalization property, variance reduction strategies, and ways to use $\hat{Q}$ to evaluate CATE estimators.

### 3.1 $Q$ AND ITS STATISTICAL ESTIMATOR $\hat{Q}$

For a given CATE estimator $\hat{\tau}$, we refactor MSE $P(\hat{\tau})$ into three parts:

$$P(\hat{\tau}) = \mathbb{E}_X[(\tau(X) - \hat{\tau}(X))^2] = \underbrace{\mathbb{E}_X[\tau^2(X)]}_{\text{unobservable constant}} + \underbrace{\mathbb{E}_X[\hat{\tau}^2(X)] - 2\mathbb{E}_X[\tau(X)\hat{\tau}(X)]}_{\text{can be approximated from real-world dataset}} \quad (1)$$

The first part, $P_1 = \mathbb{E}_X[\tau^2(X)]$, is unobservable but independent from the CATE estimator, thus can be dropped when we evaluate *relative performance* of models. The two other parts, $P_2(\hat{\tau}) = \mathbb{E}_X[\hat{\tau}^2(X)]$, and $P_3(\hat{\tau}) = \mathbb{E}_X[\tau(X)\hat{\tau}(X)]$, can be approximated with confidence. This yields the statistical parameter that drives oracle model ranking:

$$Q(\hat{\tau}) = P_2(\hat{\tau}) - 2P_3(\hat{\tau}) = P(\hat{\tau}) - P_1 \tag{2}$$

Let $q(x, t, y; \hat{\tau}) = \hat{\tau}^2(x) - 2\hat{\tau}(x)\eta(x, t, y)$, where $\eta$ is the shorthand for Horwitz-Thompson estimator, and let $q_n(\hat{\tau}) = q(x_n, t_n, y_n; \hat{\tau})$ for $n$-th sample. We now define the sample statistic:

$$\hat{Q}(\hat{\tau}) = \frac{1}{N} \sum_n q_n(\hat{\tau}) = \frac{1}{N} \sum_n [\hat{\tau}^2(x_n) - 2\hat{\tau}(x_n)\eta(x_n, t_n, y_n)] \tag{3}$$

Note that we can compute the value of $\hat{Q}$ without counterfactual ground truth. It follows that:

**Lemma 3.1.** *Unbiasedness. When the propensity function $P(T = 1|X = x) = e(x)$ is known,* $\mathbb{E}[\hat{Q}(\hat{\tau})] = Q(\hat{\tau})$.

See Appendix B for all proofs. In particular, Theorem B.2 establish the consistency of $\hat{Q}$ when propensity needs to be estimated.

*Remark* 3.2. *Relationship with orthogonal ML.* Lemma 3.1 is connected to orthogonal ML methods (Foster & Syrgkanis, 2023). For example, Theorem 15.2.1 in Chernozhukov et al. (2024) discuss similar CATE evaluation techniques based on orthogonality assumptions, requiring the triple product of the propensity error, plug-in outcome estimate error, and the difference between two CATE estimates to converge at an $O(1/\sqrt{N})$ rate.

While Lemma 3.1 may initially appear similar to results in orthogonal ML (by zero-ing out propensity risk), it is essential to establish these results without relying on orthogonal ML assumptions. As discussed in Section 1, CATE evaluation should be based on fewer and simpler assumptions than CATE estimation. Furthermore, as Section 4 will show, orthogonality-based estimators often fail to capture real-world heterogeneity, raising doubts about their reliability in CATE evaluation. This makes it critical to develop results on stronger foundation. To our knowledge, our result is the first to provide asymptotic guarantees for oracle ranking under such general conditions.

*Remark* 3.3. *Local Effect.* Lemma 3.1 can be easily extended to the case where $Q$ is weighted by function $w(x) > 0$. That is, $\hat{Q}(\tau; w) = \sum_n [w(x_n)(\hat{\tau}^2(x_n) - 2\eta(x_n, t_n, y_n)\hat{\tau}(x_n))] / \sum_n w(x_n)$ is an unbiased estimator of $Q(\hat{\tau}; w) = \mathbb{E}_X[w(X)(\hat{\tau}^2(X) - 2\hat{\tau}(X)\tau(X))]$. This result is useful when some samples are more important than others.

Intuitively, $\hat{Q}$ are relative performance metrics. Meanwhile, they can be used to measure absolute performance of CATE estimators, in three ways below. We will demonstrate their use in Section 4.

*Remark* 3.4. *Degeneracy.* A CATE estimator is useless if $Q(\hat{\tau}) \geq 0$; when this happens, we call the estimator is *degenerate*. To see that, let $\hat{\tau}_0 = 0$ be a (trivial) CATE estimator that estimates no CATE constantly. If $Q(\hat{\tau}) \geq 0$, the CATE estimator $\hat{\tau}$ has a higher MSE than $\hat{\tau}_0$. This suggests the model is useless. As a result, $\hat{Q}(\hat{\tau}) \geq 0$ can be used to detect severe errors in CATE estimation.

*Remark* 3.5. *Heterogeneity screening.* Secondly, let $Q(\hat{\tau}_B)$ be a constant effect (ATE) estimator. A CATE estimator with $Q(\hat{\tau}) \geq Q(\hat{\tau}_B)$ is useless as its MSE is higher than a constant effect estimator.

*Remark* 3.6. *Approximate MSE.* Finally, we can construct an MSE estimate, $\hat{P}(\hat{\tau})$, by decorating $\hat{P}(\hat{\tau})$ with plug-in estimates of potential outcomes $\tilde{\mu}^{(0)}(x)$ and $\tilde{\mu}^{(1)}(x)$ as follows. $\hat{P}(\hat{\tau})$ helps us understand the error magnitude of CATE estimator and retains same ranking property as $\hat{Q}(\hat{\tau})$
$\hat{P}(\hat{\tau}) = \hat{Q}(\hat{\tau}) + \frac{1}{N} \sum_n (\tilde{\mu}^{(1)}(x) - \tilde{\mu}^{(0)}(x))^2$.

Does CATE evaluation result generalize to new distributions? We answer in the next two theorems:

A scientist with access to a dataset generated by one distribution may want to use it to find the best CATE estimator for a second, different but related distribution, without the cost of second data selection. Theorem 3.7 below shows that we can use data from one distribution to estimate $Q$ on another, via Inverse Propensity Weighting, when the density ratio between two distributions are known or can be reliably estimated.

**Theorem 3.7.** *Generalization via Inverse Propensity Weighting. Let $\Pi_1$ and $\Pi_2$ be two data distributions sharing the same $\tau(x)$. Let $\mathcal{X}_1$ and $\mathcal{X}_2$ be their marginal distribution of $X$ with density $\rho_1(x)$ and $\rho_2(x)$ respectively. Also assume both distributions share common support and let*

$\zeta(x) = \rho_2(x)/\rho_1(x)$ be the density ratio. Let $\{(x_n, t_n, y_n)\}(1 \leq n \leq N)$ be $N$ i.i.d samples drawn from $\Pi_1$. We have $Q(\hat{\tau}; \mathcal{X}_2) = \mathbb{E}_{\Pi_1} \left[ \frac{1}{N} \sum_n \zeta(x_n) \left[ \hat{\tau}^2(x_n) - 2\eta(x_n, t_n, y_n)\hat{\tau}(x_n) \right] \right]$.

When density ratio is difficult to estimate, or when the potential outcome distribution changes, a scientist may wonder if the best CATE estimator identified for one distribution is also the best for another. Theorem 3.8 below states that the CATE estimator with smaller $Q$ on one distribution is also the CATE estimator with smaller $Q$ on the second, when the two distributions are close enough:

**Theorem 3.8.** *Ranking Generalization. Let $\Pi_1$ and $\Pi_2$ be two different joint distribution of $X, Y^{(0)}, Y^{(1)}$. Let $\hat{\tau}_1$ and $\hat{\tau}_2$ be two deterministic CATE estimators. Let $h_0(x, y_0, y_1; \hat{\tau}) = \hat{\tau}^2(X) - 2\hat{\tau}(x)(y_1 - y_0))$ be a shorthand function. Let $D_H(\Pi_1, \Pi_2) := \sup_{h \in H} |E_{\Pi_1}[h(X, Y^{(0)}, Y^{(1)})] - E_{\Pi_2}[h(X, Y^{(0)}, Y^{(1)})]| < \Delta$ be the Integral Probability Metric bounded by a finite constant $\Delta$, where $H$ is a set of real-valued functions such that $h_0(\hat{\tau}_1), h_0(\hat{\tau}_2) \in H$. When $Q(\hat{\tau}_1; \Pi_1) - Q(\hat{\tau}_2; \Pi_1) \geq 2\Delta$, we have $Q(\hat{\tau}_1; \Pi_2) - Q(\hat{\tau}_2; \Pi_2) > 0$.*

## 3.2 RESULTS FOR OBSERVATIONAL SAMPLING

In case of observational sampling (to be used in Section 4), results can be further improved. In observational sampling, we sample a subset from experiment data to train CATE estimators. We then evaluate the CATE estimators on the remaining RCT sub-sample. These results help the benchmark.

First we explore opportunities of variance reduction. Even with Lemma 3.1, the variance of $\hat{Q}$ can still be large in finite-sample settings, driven by the high-variance nature of Horwitz-Thompson estimator $\eta$. In this section we provide a general control variates framework to reduce variance of $\hat{Q}$ while preserving its desirable unbiased property. To start, we introduce the basic concepts of control variates (Glynn & Szechtman, 2002). Let $U$ be a real-valued random variable and we want to estimates its mean $\mathbb{E}[U]$. We can use the sample mean estimator $\bar{U} = \sum_n u_n/N$ where $u_n$ are i.i.d samples of $U$. Suppose that there exists a zero-mean random variable $V, \mathbb{E}[V] = 0$. Then, the control variate $\bar{U}(\theta) = \sum_n (u_n + \theta v_n)/N = \bar{U} + \theta \sum_n v_n/N$ is also an unbiased estimator of $\mathbb{E}[U]$. Moreover, the variance-minimizing choice is $\theta^* = -\text{Cov}(U, V)/\text{Var}[V]$.

To apply control variates on $\hat{Q}$, note that $\hat{Q} = \frac{1}{N} \sum_n q_n$ is the sample mean of $q(X, T, Y; \hat{\tau})$. Let $r(x, t, y; \hat{\tau})$ be a *control variates function* with zero mean, i.e., $\mathbb{E}[r(X, T, Y; \hat{\tau})] = 0$. Therefore

$$\hat{Q}(r(\cdot); \hat{\tau}) = \frac{1}{N} \sum_n \left[ q(x_n, t_n, y_n; \hat{\tau}) + \theta r(x_n, t_n, y_n; \hat{\tau}) \right] \tag{4}$$

has the same expectation as $\hat{Q}(\hat{\tau})$, i.e., $\mathbb{E}[\hat{Q}(r(\cdot); \hat{\tau})] = \mathbb{E}[\hat{Q}(\hat{\tau})]$.

Next we show that location invariance and commonly used CATE estimation losses are special cases of this control variates framework.

**Proposition 3.9.** *Location Invariance. Assume $X \perp T$. Let the location invariance control variates function be $r_{LI}(x, t, y; \hat{\tau}) = 2 \left( \frac{t}{E_1} - \frac{(1-t)}{E_0} \right) \hat{\tau}(x)$. $\hat{Q}(r_{LI}) = \hat{Q} + \theta \frac{1}{N} \sum_n r_{LI}(x_n, t_n, y_n; \hat{\tau})$ is an unbiased estimator of $Q$.*

**Proposition 3.10.** *Doubly Robust loss. Assume $X \perp T$. Define the control variates function as $r_{DR}(x, t, y; \hat{\tau}) = -2\gamma(x, t)\hat{\tau}(x)$ and $\hat{Q}(r_{DR}) = \hat{Q} + \frac{1}{N} \sum_n r_{DR}(x_n, t_n, y_n; \hat{\tau})$, where $\gamma(x, t)$ is the shorthand function defined in Section 2. We have $\mathbb{E}[\hat{Q}(r_{DR})] = Q$ and $\mathcal{L}_{DR}(\hat{\tau}) = \hat{Q}(r_{DR}) + \frac{1}{N} \sum_n \left[ \eta(x_n, t_n, y_n) + \gamma(x_n, t_n) \right]^2$ is equal to $\hat{Q}(r_{DR})$ plus a constant independent from $\hat{\tau}$.*

**Proposition 3.11.** *R-loss. Assume $X \perp T$. Define the control variates function as $r_R(x, t) = -4(1 - 2t)\tilde{m}(x)\hat{\tau}(x)$ and $\hat{Q}(r_R) = \hat{Q} + \frac{1}{N} \sum_n r_R(x_n, t_n, y_n; \hat{\tau})$. We have $\mathbb{E}[\hat{Q}(r_R)] = Q$. Moreover when $E_1 = \Pr(T = 1) = 0.5$, $\mathcal{L}_R(\hat{\tau}) = \frac{\hat{Q}(r_R)}{4} + \frac{1}{N} \sum_n (y_n - \tilde{m}(x_n))^2$ is equal to $\hat{Q}(r_R)/4$ plus a constant independent from $\hat{\tau}$.*

Variations of $\hat{Q}$ are rank-preserving when used to evaluate CATE models. When the dataset grows large, their difference disappears. Which variant to use depends on theoretical and practical considerations: the original $\hat{Q}$ and $\hat{Q}(r_{LI})$ are model-free and easy to implement. Meanwhile, the variance

reduction variants $\hat{Q}(r)$, including its special cases $\hat{Q}(r_R)$ and $\hat{Q}(r_{DR})$, offers the potential benefit of even lower variance, at the price of fitting and saving the extra plug-in estimators. Finally, note that Chernozhukov et al. (2023) proved results similar to Propositions 3.11 and 3.10, based on stringent orthogonality assumptions; their result do not generalize to other control variates.

Finally, we prove that all variants of $\hat{Q}$ achieves $O(1/\sqrt{N})$ convergence rate:

**Theorem 3.12.** *Convergence Rate. Assume $Y$ is a bounded random variable, $\hat{\tau}(x)$ is a bounded function, propensity score is bounded, i.e., $0 < \bar{e} < e(x) = \Pr(T = 1|X = x) < 1 - \bar{e} < 1$, and that the control variate function $r(x, t, y; \hat{\tau})$ is bounded. We have $\sqrt{N}(\hat{Q}(r; \hat{\tau}) - Q) \to \mathcal{N}(0, \sigma^2(r, \hat{\tau}))$ where $\sigma^2(r, \hat{\tau})$ is the finite variance for $q(r; \hat{\tau})$.*

## 4 BENCHMARK AND FINDINGS

### 4.1 BENCHMARK DESIGN VIA OBSERVATIONAL SAMPLING

Now that we have studied the statistical properties of variations of $\hat{Q}$, we are ready to use it to evaluate CATE estimation models using real-world datasets. We emphasize that the evaluation is more than a *post-mortem* examination. Theorems 3.7 and 3.8 suggests that results obtained from one distribution can generalize to a new one under the right conditions. Performance of CATE estimators on a carefully selected portfolios of observational sampling study are predictive indicators to their future performance on new and similar distributions.

*Dataset generation.* We use twelve large RCT datasets for this evaluation study. They are listed in Table 4. These datasets were selected to represent diverse real-world data generation processes; the rationale for their inclusion and additional details are in Appendix E.

*Observational sampling.* For each RCT dataset $D$, we sample it to generate the estimation $D_{est}$ with selection bias, and an evaluation RCT dataset $D_{eval}$; there is no overlap between them. See Appendix F for details. We vary three sampling parameters, 4 variations in estimation dataset size, 3 variations in treatment %, and 3 variations in assignment mechanism nonlinearity; this results in 36 settings. For every setting, we sample $D_{est}$ and $D_{eval}$ jointly 100 times, yielding 3,600 pairs of $D_{est}$ and $D_{eval}$. Repeating same process for the 12 RCT datasets yields $12 \times 3,600 = 43,200$ benchmark datasets.

*Estimation model selection.* We evaluate 16 CATE estimation models on $D_{est}$. These models include variations of S, R, and T learners (Künzel et al., 2019), Doubly Robust learners (Kennedy, 2023), Double Machine Learning models (Chernozhukov et al., 2017; Nie & Wager, 2020), representation learning-based models (Shi et al., 2019), and causal random forest models (Athey et al., 2018). We use the format `<model-name>.<base-learner>.<details>` as the model code when presenting results. Full model details can be found in Table 3 in Appendix C. We use code from Curth & van der Schaar (2023); Curth (2023); Battocchi et al. (2019) for reproducibility.

*Estimation and Evaluation.* We train 16 models listed in Table 3 on $D_{est}$. See Appendix C for details. We then evaluate the trained models on $D_{eval}$, using $\hat{Q}(r_{DR})$.

### 4.2 FINDINGS

Table 1 summarizes the benchmark findings. For each dataset, we calculate $\hat{Q}$ for every model. Out of 43,200 datasets, 41,499 (96%) have at least one non-degenerate model with $\hat{Q}(\hat{\tau}) < 0$. For these datasets, models are ranked from best (rank 1 for the most negative $\hat{Q}$) to worst. A model "wins" if it ranks 1, and its "win share" reflects how often it outperforms other models. We also compute each model's average degenerate rate.

The benchmark reveals critical insights into the current landscape of CATE estimation models:

1. *CATE models produce degenerate estimators more than half the time.* We found 62% of fitted CATE estimators are degenerate; Among them, 94% are statistically different from zero at 5% significance level. This highlights the need for problem-specific fine-tuning. It also suggests that using using $\hat{Q}(\hat{\tau}) \leq 0$ as a guardrail is crucial for avoiding poor performance, when possible.

2. *CATE estimators fail to outperform a constant-effect benchmark 80% of the time.* We use Double ML with a Lasso base learner (`dml.lasso`) to construct $\hat{\tau}_B$, a constant-effect estimator. Among

Table 1: Model comparison: summary of 43,200 datasets

| Model | Wins | Win share | Degenerate | Degenerate rate | Avg rank |
|---|---|---|---|---|---|
| `s.xgb.cv` | 10,491 | 25.5% | 2,600 | 6.3% | 4.4 |
| `s.ridge.cv` | 5,327 | 12.9% | 12,837 | 31.2% | 4.2 |
| `dragon.nn` | 4,976 | 12.1% | 18,021 | 43.8% | 5.1 |
| `s.ext.ridge.cv` | 4,582 | 11.1% | 20,760 | 50.4% | 5.6 |
| `dml.elastic` | 3,413 | 8.3% | 19,913 | 48.4% | 5.6 |
| `dml.lasso` | 3,279 | 8.0% | 19,916 | 48.4% | 5.7 |
| `s.ext.xgb.cv` | 2,648 | 6.4% | 21,344 | 51.8% | 6.9 |
| `r.ridge.cv` | 2,532 | 6.2% | 25,209 | 61.2% | 8.7 |
| `dr.ridge.cv` | 2,499 | 6.1% | 24,384 | 59.2% | 7.1 |
| `t.ridge.cv` | 1,780 | 4.3% | 26,383 | 64.1% | 8.2 |
| `dr.xgb.cv` | 476 | 1.2% | 29,409 | 71.4% | 9.9 |
| `cforest` | 209 | 0.5% | 31,286 | 76.0% | 10.5 |
| `t.xgb.cv` | 187 | 0.5% | 31,561 | 76.7% | 11.4 |
| `r.xgb.cv` | 110 | 0.3% | 34,814 | 84.6% | 12.4 |
| `dml.xgb` | - | 0.0% | 40,741 | 99.0% | 15.9 |
| `dml.linear` | - | 0.0% | 38,796 | 94.2% | 14.3 |

25,440 datasets with non-degenerate $\hat{\tau}_B < 0$, only 20% of CATE estimators ($\hat{\tau}$) outperform $\hat{\tau}_B$. This finding is striking, given that these methods are explicitly designed to capture heterogeneity. It also highlights the underappreciated value of heterogeneity detection methods (Crump et al., 2008; Chernozhukov et al., 2023), which deserve significantly more attention.

3. *Orthogonality-based learners underperform.* Despite their theoretical advantages, these models (model name `dml`, `r`, `dr`, and `cforest`) have an average degenerate rate of 71%, and win only 30% of the time. This underperformance raises concerns about the self-serving bias inherent in using their proxy losses as CATE evaluation criteria. Their performance, we hypothesize, arises from a combination of factors, including the data-generating process and modeling choices. While the sample-splitting and debiasing mechanisms should, in theory, mitigate risks from poorly specified outcome models, other practical challenges—such as violations of assumptions required for orthogonality conditions to hold—may play a role. This is a topic for our ongoing research.

4. *All learners are weak learners.* Among the sixteen models evaluated, `s.xgb.cv` had the highest win share at 25.5%. Unlike prior studies, our findings are based on real-world data rather than simulated outcomes, reinforcing the relevance of these results for practical applications. Detailed performance analysis is available in Appendix I.1.

### 4.3 RESULT VALIDITY AND CONSIDERATIONS

We were surprised by the findings. While we anticipated relative accuracy varies, we did expect contemporary CATE estimators to provide generally useful estimates. Before concluding that these results reflect fundamental issues with CATE estimation, we consider several alternative explanations:

*Is $\hat{Q}$ really performing oracle ranking?* We present simulation results on the agreement between $\hat{Q}$ and MSE $P$ when used to select best models. We tested using semi-synthetic datasets based on the Hillstrom dataset (Hillstrom, 2008). Synthetic potential outcomes and treatments were generated, and the dataset was split into an estimation set ($D_{est}$) and an evaluation set ($D_{eval}$) of varying sizes (1,000 to 64,000 samples). We trained the same 16 CATE models on $D_{est}$ and evaluated them on $D_{eval}$ using $\hat{Q}$ variants as the evaluation criteria. To assess the accuracy of $\hat{Q}$, we compared model rankings from $\hat{Q}$ with oracle rankings available in the simulated data using ranking metrics; see Figure 1 for MRR and Appendix D for more details. As predicted by theory, the agreement between $\hat{Q}$ and the oracle improved with larger evaluation datasets, and $\hat{Q}$ consistently outperformed alternative evaluation metrics. See Appendix D.2 for more results.

*Implementation Accuracy.* The possibility of implementation errors is a valid concern, but we minimize the risk by re-using the codebase (Curth, 2023) that has been used for recent large-scale

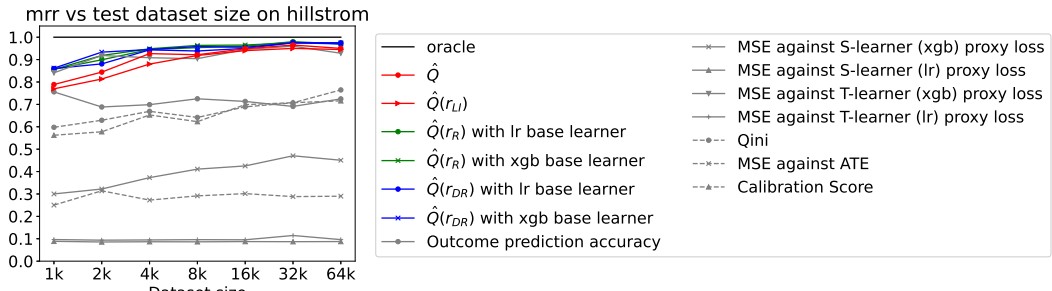

Figure 1: Ranking agreement between $\hat{Q}$ variants and oracle.

benchmarks (Curth & van der Schaar, 2023). We relied on existing CATE estimators and evaluation criteria when possible and used EconML for additional implementations. We are committed to releasing our code following proper approval to further ensure transparency and reproducibility.

*Model Selection.* Our evaluation focused on 16 widely used CATE models; they span the major strategies in CATE estimation discussed in Section 2.2. While resource constraints limited the number of models we could include, this selection offers a representative evaluation of contemporary methods. However, we acknowledge that additional models, particularly from deep learning and Gaussian Process approaches (Alaa & van der Schaar, 2017), could provide further insights.

*Context-Specific Generalizability.* While the datasets used in our benchmark may not cover every researcher's specific needs, they represent a diverse range of real-world data generation processes. Our results expose significant risks in CATE estimation, particularly for practitioners without the deep domain expertise necessary for rigorous model fine-tuning. These findings provide crucial insights into the limitations of widely used CATE methods in capturing real-world heterogeneity.

## 5 CONCLUSIONS

We introduce a new approach to evaluating CATE estimators using observational sampling, centered around the statistical parameter $Q$ to identify the estimator with the lowest MSE. The $\hat{Q}$ family of statistics are computable from real-world RCT datasets, allowing us, for the first time, to evaluate CATE estimator's ability to capture real-world heterogeneity without counterfactual ground truth. However, the most important contribution of this work is not just the method, but the empirical findings themselves. These findings reveal that contemporary CATE estimators often fail to outperform trivial baselines, raising fundamental concerns about the field's reliance on limited benchmarks and simulation-driven validation. They highlight significant challenges in CATE estimation and underscore the need for a re-examination of evaluation practices.

Our work brings renewed attention to foundational principles and highlights new opportunities. First, it underscores the central role of RCTs in causal inference and renews interest in observational sampling methods, an underutilized tool for CATE evaluation. Second, it reveals the need for broader, more representative benchmarks. Our study, while large by current standards, is an early step toward this goal. Systematic evaluation across more datasets, including high-sample regimes, may clarify when and why models fail. Third, the observed performance gaps suggest future work should explore alternative modeling assumptions, improved regularization, or new architectures tailored to real-world heterogeneity. Ultimately, our findings do not suggest that progress in CATE estimation has stalled, but that evaluation practices must evolve. The reliance on small benchmarks and simulated comparisons may have created an artificial sense of model superiority. Moving forward, the field requires both methodological innovation and stronger empirical validation on diverse, high-quality datasets. We hope this work catalyzes broader benchmarking efforts, ensuring that future models and model evaluation better capture real-world heterogeneity.

## ACKNOWLEDGMENTS

We sincerely thank Randall Lewis, Lihong Li, Rui Song, Ryan Shyu, Max Farrell, Mathias Cattaneo, and Andrew Gelman for their valuable discussions and contributions to this work. We appreciate the constructive feedback from the anonymous reviewers, which helped improve this manuscript.

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

## A    NOTATION TABLE

Let $f$ be a ground truth function defining the data generation process; $f$ is often unobservable. We use $\hat{f}$ to represent the main estimator, and $\tilde{f}$ to represent the plug-in.

| Notation | Definition |
|---|---|
| $X$ | Pre-treatment random vector |
| $T$ | Binary treatment |
| $Y(0)$ and $Y(1)$ | Potential outcomes |
| $Y = Y(T)$ | Outcome |
| $\mu^{(0)}(x)$ and $\mu^{(1)}(x)$ | Expectations of potential outcomes |
| $\tilde{\mu}^{(0)}(x)$ and $\tilde{\mu}^{(1)}(x)$ | Plug-in estimates of potential outcomes |
| $\tau(x) = \mu^{(1)}(x) - \mu^{(0)}(x)$ | ground truth CATE function |
| $\hat{\tau}(x)$ | CATE estimate |
| $e(x) = \Pr(T = 1 \mid X = x)$ | Propensity function |
| $\tilde{e}(x)$ | Plug-in estimate of $e(x)$ |
| $E_1$ | Treatment probability in RCT, i.e., $E_1 = \Pr(T = 1)$. Also $E_0 = 1 - E_1$ |
| $D$ | A dataset, a list of $(X, T, Y)$ tuples |
| $N$ | Number of samples in dataset |
| $(x_n, t_n, y_n)$ | $n$-th sample in $D$ |
| $D_{est}$ | The estimation dataset, a subset of $D$ |
| $D_{eval}$ | The evaluation dataset, a subset of $D$ |
| $P$ | Mean Squared Error of CATE estimator. Also known as PEHE |
| $Q$ | The statistical parameter; also the part of MSE that depends on $\hat{\tau}$ |
| $\hat{Q}$ | The sample statistics computed on dataset $D$ |
| $r(\cdot)$ | The zero-mean control variate function |
| $\hat{Q}(r)$ | The sampled statistics with control variates function $r$ |
| $\mathcal{L}_R$ | R loss |
| $\mathcal{L}_{DR}$ | DR loss |
| $m(x)$ | $\mathbb{E}[Y \mid X = x]$, used in R loss |
| $\tilde{m}(x)$ | Plug-in estimate of $m(x)$ |
| $\eta(x, t, y)$ | Shorthand function used for IPW estimator |
| $\gamma(x, t)$ | Shorthand function used in DR learner |

Table 2: Notations

## B    PROOFS

### B.1    PROOF OF LEMMA 3.1

*Proof.* To see that, notice

$$
\begin{aligned}
P_3(\hat{\tau}) &= \mathbb{E}_X[\hat{\tau}(x)\tau(x)] \\
&= \mathbb{E}_X[\hat{\tau}(x)\mathbb{E}_{T,Y \mid X}[\eta(X, T, Y)]] \\
&= \mathbb{E}_X[\mathbb{E}_{T,Y \mid X}[\hat{\tau}(x)\eta(X, T, Y)]] \\
&= \mathbb{E}_{X,T,Y}[\hat{\tau}(x)\eta(X, T, Y)] \\
&= \mathbb{E}\Big[\frac{1}{N} \sum_n \hat{\tau}(x_n)\eta(x_n, t_n, y_n)\Big]
\end{aligned}
$$

where the second equation is due to unbiasedness of Horwitz-Thompson estimator $\eta$. It follows that

$$
\begin{aligned}
Q &= P_2 - 2P_3 \\
&= \mathbb{E}\Big[\frac{1}{N}\sum_n (\hat{\tau}(x_n) - 2\hat{\tau}(x_n)\eta(x_n, t_n, y_n))\Big] \\
&= \mathbb{E}\Big[\frac{1}{N}\sum_n q(x_n, t_n, y_n; \hat{\tau})\Big] \\
&= \mathbb{E}[\hat{Q}]
\end{aligned}
$$

Thus $E[\hat{Q}] = Q$ holds as long as the ground truth propensity function $e(x)$ is known, even if it is not constant. $\square$

*Remark* B.1. The proof can be extended to other unbiased CATE estimators.

## B.2  PROOF OF THEOREM B.2

Based on Lemma 3.1, We can establish the consistency of $\hat{Q}$ when propensity needs to be estimated:

**Theorem B.2.** *Consistency. Assume propensity $e(x)$ and its estimate $\hat{e}(x)$ are both bounded away from zero and way on their support:* $0 < \bar{e} \le e(x), \hat{e}(x) \le 1 - \bar{e} < 1$. *Also assume* $\lim_{n\to\infty} \Pr(\mathbb{E}_X[|\hat{e}_n(x) - e(X)|] > \epsilon) = 0$ *for all* $\epsilon > 0$. *We have for all* $\epsilon > 0$,

$$
\lim_{n\to\infty} \Pr(|\hat{Q}_n(\hat{\tau}) - Q(\hat{\tau})| > \epsilon) = 0 \tag{5}
$$

*Proof.* First, notice

$$
\begin{aligned}
\hat{Q}(e(x)) - \hat{Q}(\hat{e}(x)) &= \frac{1}{N}\sum_n \Big[\hat{\tau}^2(x_n) - 2\hat{\tau}(x_n)(\frac{t_n}{e(x_n)} - \frac{1-t_n}{1-e(x_n)})y_n\Big] \tag{6} \\
&\quad - \frac{1}{N}\sum_n \Big[\hat{\tau}^2(x_n) - 2\hat{\tau}(x_n)(\frac{t_n}{\hat{e}(x_n)} - \frac{1-t_n}{1-\hat{e}(x_n)})y_n\Big] \tag{7} \\
&= \frac{1}{N}\sum_n 2\hat{\tau}(x_n)y_n\left(\frac{t_n}{\hat{e}(x_n)} - \frac{t_n}{e(x_n)} + \frac{1-t_n}{1-e(x_n)} - \frac{1-t_n}{1-\hat{e}(x_n)}\right) \tag{8}
\end{aligned}
$$

It follows that

$$
|\hat{Q}(e(x)) - \hat{Q}(\hat{e}(x))| \tag{9}
$$
$$
\le \frac{1}{N}\sum_n |2\hat{\tau}(x_n)y_n|\left(\left|\frac{t_n}{\hat{e}(x_n)} - \frac{t_n}{e(x_n)}\right| + \left|\frac{1-t_n}{1-e(x_n)} - \frac{1-t_n}{1-\hat{e}(x_n)}\right|\right) \tag{10}
$$
$$
= \frac{1}{N}\sum_n |2\hat{\tau}(x_n)y_n|\left(\frac{t_n}{e(x_n)\hat{e}(x_n)}|\hat{e}(x_n) - e(x_n)| + \frac{1-t_n}{(1-e(x_n))(1-e(x_n))}|\hat{e}(x_n) - e(x_n)|\right) \tag{11}
$$
$$
\le \left\{\frac{1}{N}\sum_n |2\hat{\tau}(x_n)y_n|\left(\frac{t_n}{e(x_n)\hat{e}(x_n)} + \frac{1-t_n}{(1-e(x_n))(1-e(x_n))}\right)|e(x_n) - e(x_n)|\right\} \tag{12}
$$
$$
\le C_1 \frac{1}{N}\sum_n |e(x_n) - \hat{e}(x_n)| \tag{13}
$$
$$
= C_1\left(\mathbb{E}_X[|e(X) - \hat{e}(X)|] + \varepsilon_1\right) \tag{14}
$$
$$
\le C_1\left(\mathbb{E}_X[|e(X) - \hat{e}(X)|] + |\varepsilon_1|\right) \tag{15}
$$

where

$$
C_1 = 4\max\left(\frac{1}{\bar{e}^2}, \frac{1}{(1-\bar{e})^2}\right)\max_n |\hat{\tau}(x_n)y_n| \tag{16}
$$

is a constant and

$$
\varepsilon_1 = \mathbb{E}_X[|e(X) - \hat{e}(X)|] - \frac{1}{N}\sum_n |e(x_n) - \hat{e}(x_n)| \tag{17}
$$

$\varepsilon_1$ is a zero-mean random variable with asymptotic variance on the order of $o(1/\sqrt{N})$ due to Central Limit Theorem. As a result we have $\lim_{n\to\infty} \Pr(\varepsilon_1 > \epsilon) = 0$. It follows that

$$\lim_{n\to\infty} \Pr(\mathbb{E}\,|e(X) - \hat{e}(X)| > \varepsilon) = 0 \tag{18}$$

Combining the two yields

$$\lim_{n\to\infty} \Pr(|\hat{Q}(e(x)) - \hat{Q}(\hat{e}(x))| > \epsilon) = 0 \tag{19}$$

Secondly, by Theorem 3.12, we have

$$\sqrt{N}(\hat{Q}(e(x)) - Q(e(x))) \to N(0, \sigma^2) \tag{20}$$

It follows that

$$\lim_{n\to\infty} \Pr(|\hat{Q}(e(x)) - Q(e(x))| > \epsilon) = 0 \tag{21}$$

Finally, notice that

$$\hat{Q}(\hat{e}(x)) - Q(e(x)) = [\hat{Q}(\hat{e}(x)) - \hat{Q}(e(x))] + [\hat{Q}(e(x)) - Q(e(x))] \tag{22}$$

is the sum of two parts. The convergence for the first part is given by (19) and the second part by (21). It follows that

$$\lim_{n\to\infty} \Pr(|\hat{Q}(\hat{e}(x)) - Q(e(x))| > \epsilon) = 0 \tag{23}$$

$\square$

### B.3 PROOF OF THEOREM 3.7

*Proof.* Recall $Q = P_2 - 2P_3$. We have:

$$\begin{aligned}
P_2(\hat{\tau}; \mathcal{X}_2) &= \mathbb{E}_{\mathcal{X}_2}[\hat{\tau}^2(X)] & (24)\\
&= \mathbb{E}_{\mathcal{X}_1}[\zeta(X)\hat{\tau}^2(X)] & (25)\\
&= \mathbb{E}_{\Pi_1}\left[\frac{1}{N}\sum_n \zeta(x_n)\hat{\tau}^2(x_n)\right] & (26)
\end{aligned}$$

where the second equation is is due to inverse propensity weighting and the definition of density ratio, and the third equation is due to the unbiasedness nature of sample mean. Similarly,

$$\begin{aligned}
P_3(\hat{\tau}; \mathcal{X}_2) &= \mathbb{E}_{\mathcal{X}_2}[\tau(X)\hat{\tau}(x)] & (27)\\
&= \mathbb{E}_{\mathcal{X}_1}[\zeta(X)\tau(X)\hat{\tau}(x)] & (28)\\
&= \mathbb{E}_{\mathcal{X}_1}[\zeta(X)\eta(X)\hat{\tau}(x)] & (29)\\
&= \mathbb{E}_{\Pi_1}\left[\frac{1}{N}\sum_n \zeta(x_n)\eta(x_n, t_n, y_n)\tau(x_n)\right] & (30)
\end{aligned}$$

Combining the two yields

$$Q(\hat{\tau}; \mathcal{X}_2) = \mathbb{E}_{\Pi_1}\left[\frac{1}{N}\sum_n \zeta(x_n)\left[\hat{\tau}^2(x_n) - 2\eta(x_n, t_n, y_n)\hat{\tau}(x_n)\right]\right] \tag{31}$$

To summarize, if we compute $\hat{Q}(\hat{\tau})$ on $\Pi_1$ and weight it by IPW density ratio $\zeta$, we get an unbiased estimator of $Q(\hat{\tau})$ on $\Pi_2$. $\square$

### B.4 PROOF OF THEOREM 3.8

*Proof.* First note that, for a given CATE estimator $\hat{\tau}$, the difference between $Q(\hat{\tau})$ under $\Pi_1$ and $\Pi_2$ is bounded by $\Delta$:

$$
\begin{aligned}
|Q(\hat{\tau};\Pi_1) - Q(\hat{\tau};\Pi_2)| &= |E_{\mathcal{X}_1}[\hat{\tau}^2(X) - 2\hat{\tau}(X)\tau(X)] - E_{\mathcal{X}_2}[\hat{\tau}^2(X) - 2\hat{\tau}(X)\tau(X)]| \\
&= |E_{\mathcal{X}_1}[\hat{\tau}^2(X) - 2\hat{\tau}(X)E_{\mathcal{Y}_1|}[Y^{(1)} - Y^{(0)}]] - E_{\mathcal{X}_2}[\hat{\tau}^2(X) - 2\hat{\tau}(X)E_{\mathcal{Y}_2}[Y^{(1)} - Y^{(0)}]]| \\
&= |E_{\Pi_1}[h_0(X, Y^{(0)}, Y^{(1)})] - E_{\Pi_2}[h_0(X, Y^{(0)}, Y^{(1)})]| \\
&\leq \sup_{h \in H} |E_{\Pi_1}[h(X, Y^{(0)}, Y^{(1)})] - E_{\Pi_2}[h(X, Y^{(0)}, Y^{(1)})]| \\
&= D_H(\Pi_1, \Pi_2) \\
&< \Delta
\end{aligned}
$$

where the third equality is due to the definition of $h_0$ and the fifth step is due to the definition of IPM $D_H(\Pi_1, \Pi_2)$.

Let us assume we have two CATE estimators, $\hat{\tau}_1(x)$ and $\hat{\tau}_2(x)$, and

$$Q(\hat{\tau};\Pi_1)(\hat{\tau}_1) - Q(\hat{\tau};\Pi_1)(\hat{\tau}_2) \geq 2\Delta \tag{32}$$

It follows that

$$
\begin{aligned}
Q(\hat{\tau}_1;\Pi_2) - Q(\hat{\tau}_2;\Pi_2) &> Q(\hat{\tau}_1;\Pi_1) - \Delta - (Q(\hat{\tau}_2;\Pi_1) + \Delta) \\
&> Q(\hat{\tau}_1;\Pi_1) - Q(\hat{\tau}_2;\Pi_1) - 2\Delta \\
&> 0
\end{aligned}
$$

That is, $\hat{\tau}_2$ is also better on $\Pi_2$. $\qquad\square$

### B.5 PROOF OF PROPOSITION 3.9

*Proof.* First we show that $\mathbb{E}[r_{LI}(X,T,Y;\hat{\tau}] = 0$, i.e., it is a control variates function. This is obvious because

$$
\begin{aligned}
\mathbb{E}[r_{LI}(X,T,Y;\hat{\tau})] &= 2\theta\mathbb{E}\left[\left(\frac{T}{E_1} - \frac{(1-T)}{E_0}\right)\hat{\tau}(X)\right] \tag{33} \\
&= 2\theta\mathbb{E}\left[\frac{T}{E_1} - \frac{(1-T)}{E_0}\right]\mathbb{E}[\tau(\hat{X})] \tag{34} \\
&= 2\theta(1-1)\mathbb{E}[\tau(\hat{X})] \tag{35} \\
&= 0 \tag{36}
\end{aligned}
$$

where the first step is by definition of $r_{LI}(\cdot)$, and the second step is by property of RCT dataset.

It follows that

$$
\begin{aligned}
\mathbb{E}[\hat{Q}(r_{LI})] &= \mathbb{E}\left[\hat{Q} + \frac{1}{N}\sum_n r_{LI}(x_n, t_n, y_n; \hat{\tau})\right] \tag{37} \\
&= \mathbb{E}[\hat{Q}] + \mathbb{E}[r_{LI}(X,T,Y;\hat{\tau})] \tag{38} \\
&= \mathbb{E}[\hat{Q}] \tag{39} \\
&= Q \tag{40}
\end{aligned}
$$

$$\square$$

### B.6 PROOF OF PROPOSITION 3.10

*Proof.* First we prove $\mathbb{E}[r_{DR}(X,T,Y;\hat{\tau})] = 0$. First note that,

$$
\begin{aligned}
\mathbb{E}\left[(1 - \frac{T}{E_1})\hat{\mu}^{(1)}(X)\hat{\tau}(X)\right] &= \mathbb{E}[(1 - \frac{T}{E_1})]\mathbb{E}[\hat{\mu}^{(1)}(X)\hat{\tau}(X)] \tag{41} \\
&= (1-1)\mathbb{E}[\hat{\mu}^{(1)}(X)\hat{\tau}(X)] \tag{42} \\
&= 0 \tag{43}
\end{aligned}
$$

Similarly

$$\mathbb{E}\left[1 - \frac{1-T}{1-E_0}\hat{\mu}^{(0)}(X)\hat{\tau}(X)\right] = 0 \tag{44}$$

Combining the results above yields $\mathbb{E}(r_{DR}(X,T,Y;\hat{\tau})) = 0$. It follows that $\mathbb{E}[\hat{Q}_{DR}] = \mathbb{E}[\hat{Q}] = Q$

Next we show that $\mathcal{L}_{DR}$ is a linear function of $\hat{Q}(\hat{\tau})$. Recall

$$\mathcal{L}_{DR}(\hat{\tau}) = \frac{1}{N}\sum_n [\eta(x_n,t_n,y_n) + \gamma(x_n,t_n) - \hat{\tau}(x_n)]^2 \tag{45}$$

$$= \frac{1}{N}\sum_n \left\{[(\eta(x_n,t_n,y_n) + \gamma(x_n,t_n)]^2 + \hat{\tau}^2(x_n) - 2[\eta(x_n,t_n,y_n) + \gamma(x_n,t_n)]\hat{\tau}(x_n)\right\} \tag{46}$$

$$= \frac{1}{N}\sum_n \left[\eta(x_n,t_n,y_n) + \gamma(x_n,t_n)\right]^2 + \hat{Q} + \frac{1}{N}\sum_n \left[r_{DR}(x_n,t_n)\hat{\tau}(x_n)\right] \tag{47}$$

$$= \sum_n \frac{1}{N}\left[\eta(x_n,t_n,y_n) + \gamma(x_n,t_n)\right]^2 + \hat{Q}(r_{DR}) \tag{48}$$

where the first term is independent from $\hat{\tau}$ and thus can be omitted for ranking purposes. $\qquad\square$

### B.7   PROOF OF PROPOSITION 3.11

*Proof.*  First we prove the zero-mean property:

$$\mathbb{E}[(1-2T)m(X)\hat{\tau}(X)] = \mathbb{E}[m(X)\hat{\tau}(X)]\mathbb{E}[1-2T] \tag{49}$$
$$= \mathbb{E}[m(X)\hat{\tau}(X)]\cdot 0 \tag{50}$$
$$= 0 \tag{51}$$

It follows that $\mathbb{E}[\hat{Q}_R] = \mathbb{E}[\hat{Q}] = Q$.

Next, note that when $E_1 = E_0 = \tilde{e}(x) = 0.5$

$$\hat{Q} = \frac{1}{N}\sum_n \left[\hat{\tau}^2(x_n) + 4(1-2t_n)y_n\hat{\tau}(x_n)\right] \tag{52}$$

It follows that

$$\mathcal{L}_R(\hat{\tau}) = \frac{1}{N}\sum_n \left[((y_n - \tilde{m}(x_n)) - (t_n - \tilde{e}(x_n))\hat{\tau}(x_n))^2\right] \tag{53}$$

$$= \frac{1}{N}\sum_n \left[(y_n - \tilde{m}(x_n))^2 + (t_n - \tilde{e}(x_n))^2\hat{\tau}^2(x_n) - 2(y_n - \tilde{m}(x_n))(t_n - \tilde{e}(x_n))\hat{\tau}(x_n)\right] \tag{54}$$

$$= \frac{1}{N}\sum_n (y_n - \tilde{m}(x_n))^2 + \frac{1}{N}\sum_n \left[(t_n - \tilde{e}(x_n))^2\hat{\tau}^2(x_n) - 2(y_n - \tilde{m}(x_n))(t_n - \tilde{e}(x_n))\hat{\tau}(x_n)\right] \tag{55}$$

$$= \frac{1}{N}\sum_n (y_n - \tilde{m}(x_n))^2 + \frac{1}{N}\sum_n \left[\frac{1}{4}\hat{\tau}^2(x_n) + (y_n - \tilde{m}(x_n))(1-2t_n)\hat{\tau}(x_n)\right] \tag{56}$$

$$= \frac{1}{N}\sum_n (y_n - \tilde{m}(x_n))^2 + \frac{1}{4N}\sum_n \left[\hat{\tau}^2(x_n) + 4(y_n - \tilde{m}(x_n))(1-2t_n)\hat{\tau}(x_n)\right] \tag{57}$$

$$= \frac{1}{N}\sum_n (y_n - \tilde{m}(x_n))^2 + \frac{1}{4N}\left\{\sum_n \left[\hat{\tau}^2(x_n) + 4y_n(1-2t_n)\hat{\tau}(x_n)\right] - \sum_n 4\tilde{m}(x_n)(1-2t_n)\hat{\tau}(x_n)\right\} \tag{58}$$

$$= \frac{1}{N}\sum_n (y_n - \tilde{m}(x_n))^2 + \frac{1}{4}\hat{Q}(r_R) \tag{59}$$

where the first term is a constant without impact on ranking. $\qquad\square$

### B.8 PROOF OF THEOREM 3.12

*Proof.* Based on the assumptions, it is easy to see that the random variable $\eta(X, T, Y)$ is bounded. Combining this with the boundedness of $\hat{\tau}(x)$ we get $q(x, t, y|\tau) = \hat{\tau}^2(x) - 2\hat{\tau}(x)\eta(x, t, y)$ is bounded. It follow that $q(r; \hat{\tau}) = q + r$ is also bounded and thus have finite expectation and finite variance. Let us denote its variance as $\sigma^2(r, \hat{\tau})$. By Lindeberg–Lévy CLT, the sample mean $\hat{Q}(r; \hat{\tau})$ converges in distribution to its expectation $Q$: $\sqrt{N}(\hat{Q}(r; \hat{\tau}) - Q) \to \mathcal{N}(0, \sigma^2(r, \hat{\tau}))$ $\square$

## C DETAILED CONFIGURATION OF CATE ESTIMATION MODELS

We train sixteen CATE estimation models listed in Table 3. This includes two S-learners, two T-learners, two R-learners, two Doubly Robust learners, four Double ML learners, a causal tree (forest) learner, and one representation learning learner, discussed in Section 2.

We hope the selection covers mainstream CATE estimation methods. We include meta-learner (e.g., S and T learners in Künzel et al. (2019)) in our empirical study, because they represent the outcome prediction strategy, arguably the most simple and direct methods for causal inference. We include Double ML methods (Chernozhukov et al., 2017) because they represent econometric/semi-parametric view of causal inference, as well as the recent development of orthogonal and debiased Machine Learning. We include Doubly Robust learners (Kennedy, 2023) because they represent Inverse Propensity Weighting (Robins et al., 1994), a classic causal inference technique. We include causal forest because they are one of the earliest ML-based work with theoretical guarantee. Finally, we include DragonNet to represent recent trend using deep learning and representation learning for causal inference.

Our limited resource prevents us from including more CATE estimation methods. As a result we do not claim this list to be complete or "optimal". Due to resource constraint, we were unable to include more variations of deep learning models following Shalit et al. (2017), causal tree models following Athey et al. (2018), or Gaussian Process models following Alaa & van der Schaar (2017). The list may not be "optimal" because some of the modeling approaches are related, most notably between Double Machine Learning and R learners.

Table 3: CATE estimation Models

| CATE estimation Method | Base Learner | Model Code |
|---|---|---|
| DR learner | Ridge Regression | `dr.ridge.cv` |
| DR learner | XGBoost | `dr.xgb.cv` |
| R learner | Ridge Regression | `r.ridge.cv` |
| R learner | XGBoost | `r.xgb.cv` |
| S learner | Ridge Regression | `s.ridge.cv` |
| S learner | XGBoost | `s.xgb.cv` |
| S learner | Ridge Regression | `s.ext.ridge.cv` |
| S learner | XGBoost | `s.ext.xgb.cv` |
| T learner | Ridge Regression | `t.ridge.cv` |
| T learner | XGBoost | `t.xgb.cv` |
| Double ML | Linear Regression | `dml.linear` |
| Double ML | Lasso | `dml.lasso` |
| Double ML | Elastic Net | `dml.elastic` |
| Double ML | XGBoost | `dml.xgb` |
| Generalized Causal Forest | Random Forest | `cforest` |
| Representation Learning | Neural Net | `dragon.nn` |

Note that, `s.ext.xgb.cv` and `s.ext.ridge.cv` models are variants of S learners where the interaction $X \cdot T$ are constructed explicitly as model inputs.

We use codebase in Curth (2023) for model estimation and evaluation. Out of the sixteen models listed in Table 3, eight meta-learners (S-learners, T-learners, R-learners) and two Doubly Robust learners directly come from Curth (2023) implementation. Following Curth & van der Schaar (2023), we use two base learners, linear regression (implemented as an sklearn `RidgeCV` object), and XGBoost

(implemented as an XGBoost `XGBRegressor` object with grid search implemented by sklearn `GridSearchCV`). See Curth (2023) for details. The five Double Machine Learning learners are implemented using `LinearDML` and `NonParamDML` classes in `EconML` package, using sklearn `GradientBoostingRegressor` as the potential outcome learner and respective base learner as the residual model learner.

All treatment propensity estimators are implemented using sklearn `RandomForestClassifier` class; we clip propensity outputs between 0.05 and 0.95.

# D    SECTION 4.3 SUPPLEMENTAL DETAILS

## D.1    SEMI-SYNTHETIC DATASET GENERATION DETAILS

We follow the steps below to transform raw covariates into feature $x$:

- Apply one-hot encoding on all categorical covariates
- Linearly scale all float covariates between 0 and 1
- Colume stack all covariates to generate a feature vector
- If feature vector has more than 100 elements, randomly select 100.

We follow the steps below to generate synthetic outcome:

- Generate two random vectors $\beta_0$ and $\beta_1$ with discrete values of $[0, 1, 2, 3, 4]$ and discrete probability of $[0.5, 0.2, 0.15, 0.1, 0.05]$
- Compute transformed feature and outcome using the one of following three approaches:

  - *Linear.* First compute $z_0(x) = x, z_1(x) = e^x$ then generate $\mu_0(x) = \beta_0^T z_0(x)$ and $\mu_1 = e^{\beta_1^T z_1(x)}$

  - *Interaction.* First compute $z_0(x) = [x_0 x_1, x_1 x_2, ..., x_{D-1} x_0]$ and $z_1(x) = [x_0 x_2, x_1 x_3, ..., x_{D-1} x_1]$ then generate $\mu_0(x) = \beta_0^T z_0(x)$ and $\mu_1 = \beta_1^T z_1(x)$

  - *Sine.* First compute $z_0(x) = [x_0 x_1, x_1 x_2, ..., x_{D-1} x_0]$ and $z_1(x) = [x_0 x_2, x_1 x_3, ..., x_{D-1} x_1]$ then generate $\mu_0(x) = \cos(\beta_0^T z_0(x))$ and $\mu_1 = \sin(\beta_1^T z_1(x))$

- Scale $\mu_0(x)$ and $\mu_1(x)$ to have zero mean and unit standard deviation
- generate $y_0 = \mu_0(x) + N(0, 1)$ and $y_1 = \mu_1(x) + N(0, 1) + \tau$, where $\tau$ is now the ATE estimate.

We generate synthetic treatment as follows:

- Generate random vector $\beta_T$
- Calculate $\Pr(T|X = x) = \frac{1}{1 + e^{\beta_T x + 1.}}$
- Sample $T$ using the $\Pr(t|x)$

## D.2    AGREEMENT BETWEEN MODEL SELECTION CRITERIA AND ORACLE

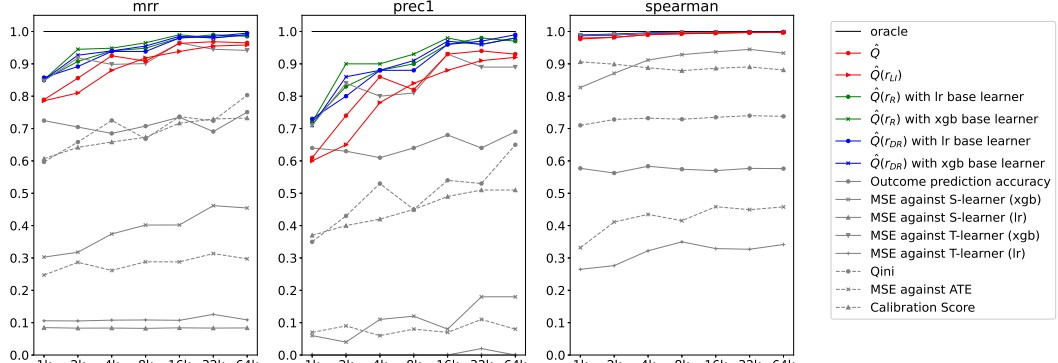

Figure 2: Interaction transformation; $\tau = 2.0$

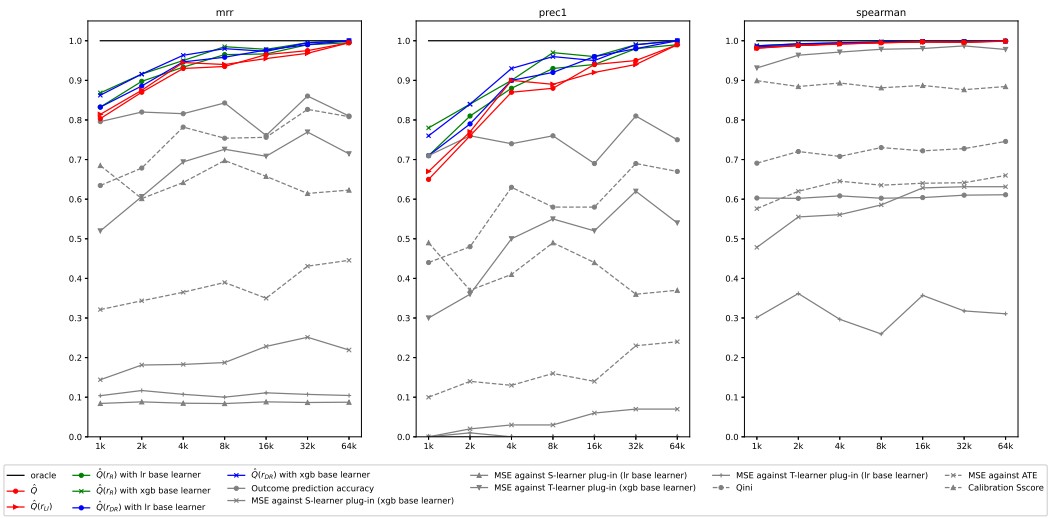

Figure 3: Sine transformation; $\tau = 0.5$

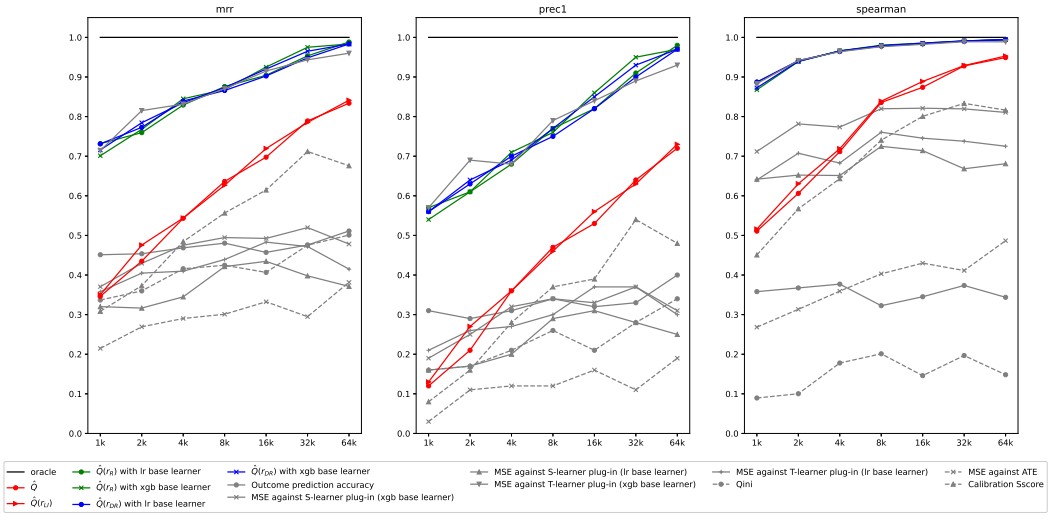

Figure 4: Linear transformation; $\tau = 0.5$

# E   DATASET INTRODUCTION

We use twelve RCT datasets in Section 4.3 and 4, listed in Table 4. In this section we discuss the rationale of data selection, and provide a brief introduction to individual datasets and data handling.

The goal for the dataset selection is to ensure that, collectively, they better represent real applications of causal inference than those datasets studied by current state of the art (e.g, Curth & van der Schaar (2023); Mahajan et al. (2023); Neal et al. (2021)): including IHDP, ACIC, and LaLonde. We achieve that by applying the following four factors when selecting the datasets:

- *Real-world heterogeneity.* We select datasets collected from real-world, and forgo datasets with simulated outcomes, to allow evaluation performance of CATE estimator on *real-world* heterogeneity. In comparison, simulated outcome on IHDP or ACIC do not achieve the same rigor.

- *Dataset size.* we prefer large datasets to allow the asymptotic property of $\hat{Q}$ to kick in. The smallest in the selection `sandercock` has  19,000 samples.

- *Diversity.* The datasets are curated to cover a diverse sources. They differ in domains (e.g., marketing in `criteo` and `hilstrom`, consumer behavior in `ferman`, medical science in `sandercock`, and sociology and political science in GSS datasets), geography (GSS from the United States, `ferman` from Brazil, `sandercock` from Europe, and `criteo` from Russia), and form of experiments (traditional RCT in `sandercock`, online A/B testing in `criteo` and `hillstrom`, and field survey in GSS)

- *Prior work.* We select datasets previously studied by causal inference and related literature. For example `criteo` is used for uplift modeling in Diemert et al. (2021); the `natfare` is used for regression adjustment in Wager et al. (2016)

Table 4: RCT Datasets

| Dataset | Samples | Treatment % | Features | References |
|---------|---------|-------------|----------|------------|
| `criteo` | 13,979,592 | 85% | 12 | Diemert et al. (2021) |
| `ferman` | 103,116 | 82% | 9 | Ferman (2015) |
| `hillstrom` | 64,000 | 67% | 12 | Hillstrom (2008) |
| `sandercock` | 19,435 | 50% | 24 | IST Collaborative Group & Sandercock (1997) |
| `nataid` | 51,957 | 50% | 20 | Davern et al. (2023) |
| `natarms` | 51,987 | 50% | 20 | Davern et al. (2023) |
| `natcity` | 51,915 | 50% | 20 | Davern et al. (2023) |
| `natcrime` | 51,977 | 50% | 20 | Davern et al. (2023) |
| `natdrug` | 51,961 | 50% | 20 | Davern et al. (2023) |
| `nateduc` | 52,017 | 50% | 20 | Davern et al. (2023) |
| `natenvir` | 52,027 | 50% | 20 | Davern et al. (2023) |
| `natfare` | 51,993 | 50% | 20 | Davern et al. (2023) |

*Criteo* dataset Diemert et al. (2021) captures advertising related online shopping behavior for 13,979,592 web users (identified by a browser cookie) in RCT. Each user is randomly assigned to either treatment or control group. Pre-assignment user activities before assignment is used to construct covariates. If a user is in treatment, they are subject to an ad exposure; if they are in control group, they are expose to the ad. The dataset tracks multiple outcomes such as visits and conversion. We use visit as the outcome in the current analysis.

*Hillstrom* dataset Hillstrom (2008) contains email marketing related activity for 64,000 shoppers who had purchase records within a year. Through randomization, one third of the shoppers receive a marketing e-mail campaign featuring Men's merchandise; one third receive an email featuring women's; and the last one third received no marketing email. Covariates include past purchase history, gender, geo location, etc. In the current paper, we combine the two groups who receive marketing email into one treatment group; the remaining group (who receives no marketin email) is the control group. We use visit as the outcome variable.

*Sandercock* dataset IST Collaborative Group & Sandercock (1997) includes data on 19,435 patients with acute stroke. Patients in treatment group are treated with aspirin; patients in control group are not. Covariates include age, gender, and other medical information. The binary outcome is whether patient is dead or dependent on other people for activities of daily living at six months after randomisation.

*Ferman* dataset Ferman (2015) includes shopping activity related to credit card payment plan on 103,116 customers of a Brazilian credit company. Customers are randomly assigned into three groups: 34,743 customers in the first group were offered a menu of payment plans with interest rate equal to 6:39%, 49,573 customers in second group were offered plans with interest rate equal to 9:59%, and the third group of 18,800 customers did not receive any payment plan offer. The outcome is whether the customer defaults within 12 months after the offer. We combine the first and second group into one treatment group.

*GSS* datasets include responses to eight questions from more than 50,000 respondents surveyed by The General Social Survey (GSS) Davern et al. (2023) between 1986 and 2022; response to each question consistutes a RCT dataset. GSS is an annual sociological survey created in 1972 by the National Opinion Research Center (NORC) at the University of Chicago. It collects information biannually and keeps a historical record of the concerns, experiences, attitudes, and practices of residents of the United States. GSS Survey regularly include randomized wording experiments to capture heterogeneity in respondent's opinion on social issues. For a given randomized question, the question variation forms different treatment arms, the answer to the question forms the outcome. GSS also collects hundreds of high-quality demographic variables, capturing demographic, work, family and spouse, household, racial, and region related information. These variables become the pre-treatment covariates. We use eight wording experiments (`nataid`, `natarms`, `natcity`, `natcrime`, `natdrug`, `nateduc`, `natenvir`, `natfare`)) to construct the binary outcome. Its value is equal to 1 if and only if when a respondent answers "too much" to a question, and 0 otherwise.

## F    SECTION 4 ADDITIONAL DETAILS ON OBSERVATIONAL SAMPLING

For every original dataset in Table 4, we vary three parameters when generating $D_{est}$: first we set the estimation dataset size to be one of the following value [1000, 2000, 4000, 8000], to test if certain models perform better with more (or less) data. Secondly, we set the expected treatment % to be one of the following values [0.1, 0.5, 0.9], to test if certain models are sensitive to treatment imbalance. [2] Third, we use a MLP in generating assignment mechanism, and set the number of MLP layers to be 1, 2, or 3. This tests if models are sensitive to nonlinearity in assignment mechanism. We enumerate all parameter combinations, leading to $4 \times 3 \times 3 = 36$ settings. For every setting, we sample $D_{est}$ and $D_{eval}$ jointly 100 times. This gives 3,600 pairs of $D_{est}$ and $D_{eval}$. Repeating same process for the 12 datasets yields $12 \times 3,600 = 43,200$ datasets.

We train model on small estimation datasets with selection bias, and evaluate model performance on large unbiased RCT datasets. The starting point is an RCT dataset $D$. We first randomly split $D$ into $D_{eval}$ and $D - D_{eval}$. We then sample $D - D_{eval}$ to get estimation dataset: for every sample $(x, t, y)$, define a random variable $K \in \{0, 1\}$ with $\Pr(K = 1 | T = t, X = x) = G(t, x)$. We keep the $n$-th sample in $D_{est}$ if and only if $k_n = 1$. $G(t, x)$ is the *biasing function* since it introduces selection bias to the original RCT dataset. This creates the estimation dataset $D_{est}$, a subset of $D$ with selection bias. We apply any CATE estimation method on $D_{est}$ to obtain an CATE estimator $\hat{\tau}(x)$, and use $\hat{\tau}(x)$ on $D_{eval}$ to compute $\hat{Q}$. Fig. 5 illustrates the process. Note that, in estimation dataset, the treatment is a function of covariates; in evaluation dataset, the treatment is randomly generated based on standard binary distribution. For every dataset and evaluation dataset size, we repeat simulation 100 times.

The complete algorithm of creating $D_{est}$ is summarized below:

---

[2]Treatment % of 0.9 can be different from 0.1 because different potential outcomes may be different in real-world datasets.

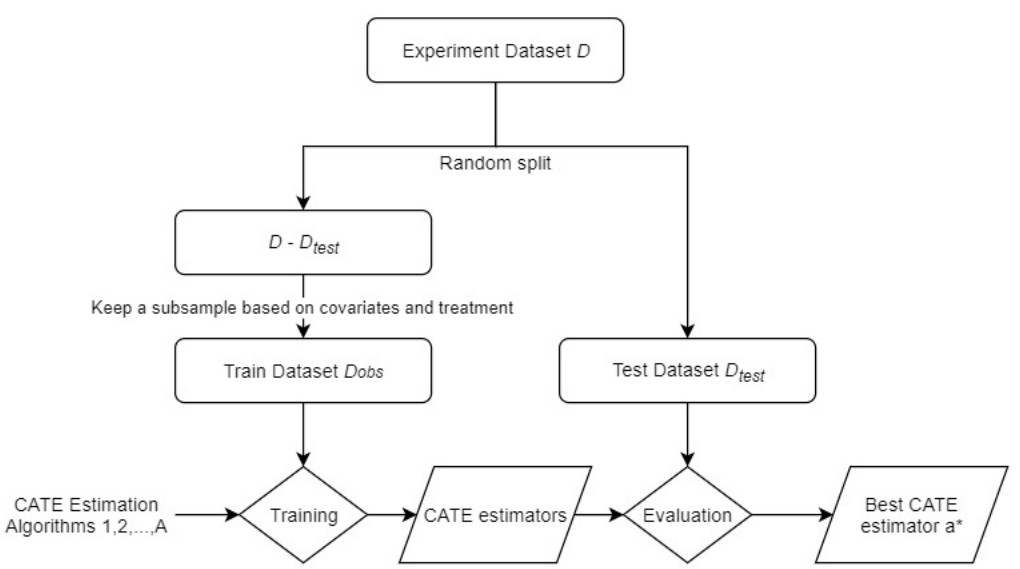

Figure 5: Overall approach

---

**Algorithm 1** Creating estimation dataset $D_{est}$

---

**Input:** RCT dataset $D$
**Input:** function $G(t, x), 0 < G(t, x) < 1$
**Output:** Observational dataset $D_{est}$
$D_{est} = \emptyset$
**for** every sample $n$ in $D$ **do**
    Sample a binary random variable $K_n$ with $\Pr(K = 1 | X = x_n, T = t_n) = G(t_n, x_n)$
    If $K_n = 1$, add sample $n$ to $D_{est}$
**end for**
Return $D_{est}$

---

Note that, the biasing function $G(t, x)$ function generates the following assignment mechanism for $D_{est}$:

$$\Pr(T = 1 | X = x, K = 1) \tag{60}$$

$$= \frac{\Pr(T = 1, X = x, K = 1)}{\Pr(X = x, K = 1)} \tag{61}$$

$$= \frac{\Pr(T = 1, X = x, K = 1)}{\Pr(X = x, K = 1)} \tag{62}$$

$$= \frac{\Pr(T = 1)\Pr(X = x)\Pr(K = 1 | X = x, T = 1)}{\Pr(X = x)\Pr(K = 1 | X = x)} \tag{63}$$

$$= \frac{\Pr(T = 1)\Pr(X = x)\Pr(K = 1 | X = x, T = 1)}{\Pr(X = x)(\Pr(T = 1)\Pr(K = 1 | X = x, T = 1) + \Pr(T = 0)\Pr(K = 1 | X = x, T = 0))} \tag{64}$$

$$= \frac{\Pr(T = 1)G(x, 1)}{\Pr(T = 1)G(x, 1) + \Pr(T = 0)G(x, 0)} \tag{65}$$

$$= \frac{1}{1 + \frac{\Pr(T=0)}{\Pr(T=1)} \frac{G(x,0)}{G(x,1)}} \tag{66}$$

As a result, $T$ is dependent on $X$, achieving selection bias on $D_{est}$.

Note that

$$
\begin{aligned}
\Pr(X \leq x | K = 1) &= \Pr(X \leq x | K = 1, T = 1) \Pr(T = 1) + \Pr(X \leq x | K = 1, T = 0) \Pr(T = 0) \quad (67)\\
&= \frac{\Pr(x \leq x, K = 1, T = 1)}{\Pr(K = 1, T = 1)} + \frac{\Pr(X \leq x, K = 1, T = 0)}{\Pr(K = 1, T = 0)} \quad (68)\\
&= \frac{\int_0^x f(x) G(x,1) dx}{\int_0^\infty f(x) G(x,1) dx} + \frac{\int_0^x f(x) G(x,0) dx}{\int_0^\infty f(x) G(x,0) dx} \quad (69)\\
&= \frac{\int_0^x f(x) G(x,1) dx}{\mathbb{E}_X[G(X,1)]} + \frac{\int_0^x f(x) G(x,0) dx}{\mathbb{E}_X[G(X,0)]} \quad (70)
\end{aligned}
$$

where $f$ is density of $X$ on $\Pi$. It follows that

$$
f_{est}(x) = f(x) \left( \frac{G(x,1)}{\mathbb{E}_X[G(X,1)]} + \frac{G(x,0)}{\mathbb{E}_X[G(X,0)]} \right) \quad (71)
$$

The complete CATE estimator evaluation algorithm is summarized below.

---

**Algorithm 2** Selecting best CATE Estimator

---

**Input:** A list of $A$ CATE estimation models $a = 1, 2, ..., A$
**Input:** RCT dataset $D$
**Input:** Biasing function $G(t, x)$
**Output:** $a^*, 1 \leq a^* \leq A$ the best performing model
Randomly split $D$ into $D_{train}$ and $D_{eval}$
Generate $D_{est}$ from $D_{train}$ using $G$ as the biasing function
**for** model $a, 1 \leq a \leq A$ **do**
    Train a CATE estimator $\hat{\tau}_a(x)$ using data $D_{est}$
    Compute $q_n$ on every sample $(x_n, t_n, y_n) \in D_{eval}$
    Compute $\hat{Q}(\hat{\tau}_a, D_{eval})$
**end for**
Return $a^* = \arg\min_a \hat{Q}(\hat{\tau}_a, D_{eval})$

---

# G    SECTION 4 RESULTS FOR ALL RCT DATASETS COMBINED

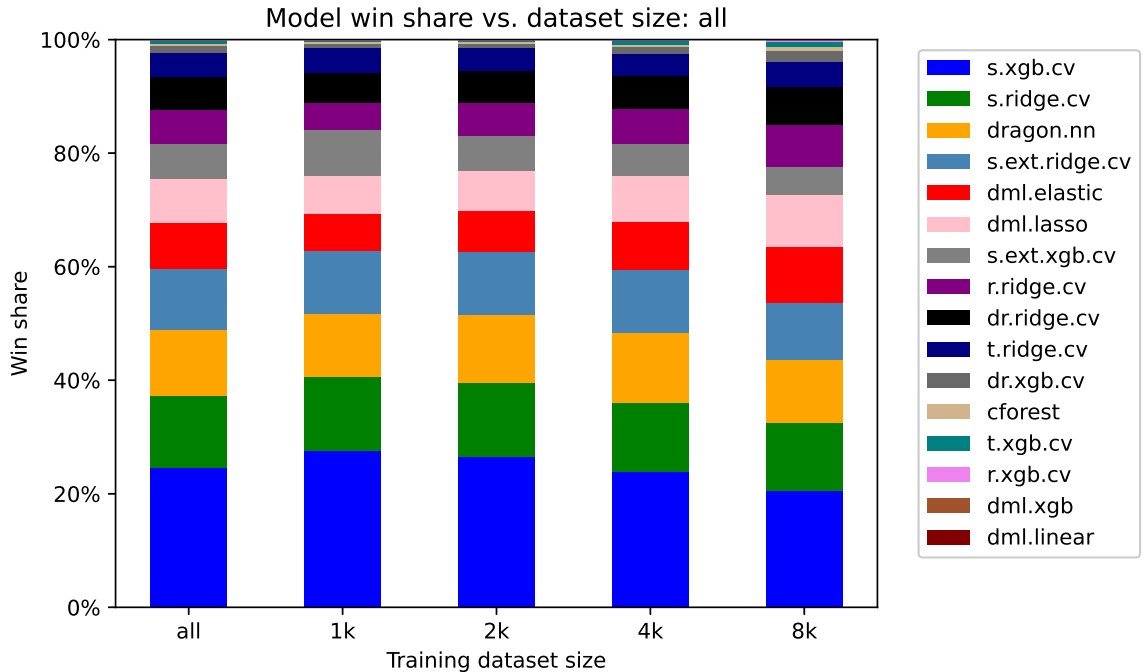

Figure 6: Model win share by training dataset size: all RCT datasets

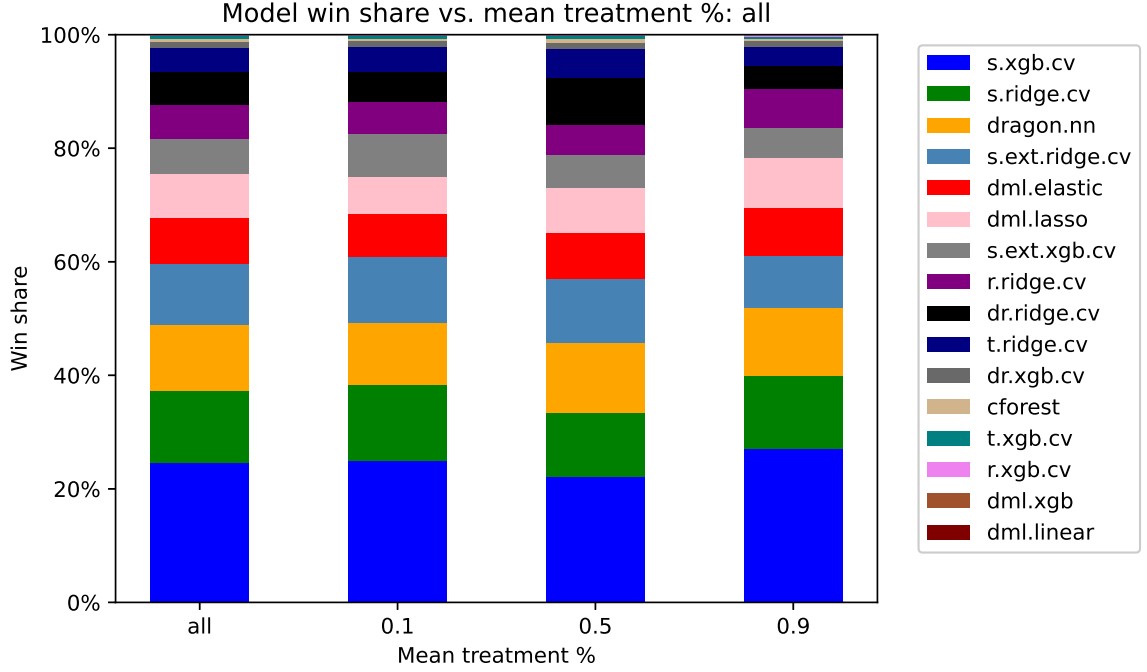

Figure 7: Model win share by treatment ratio: all RCT datasets

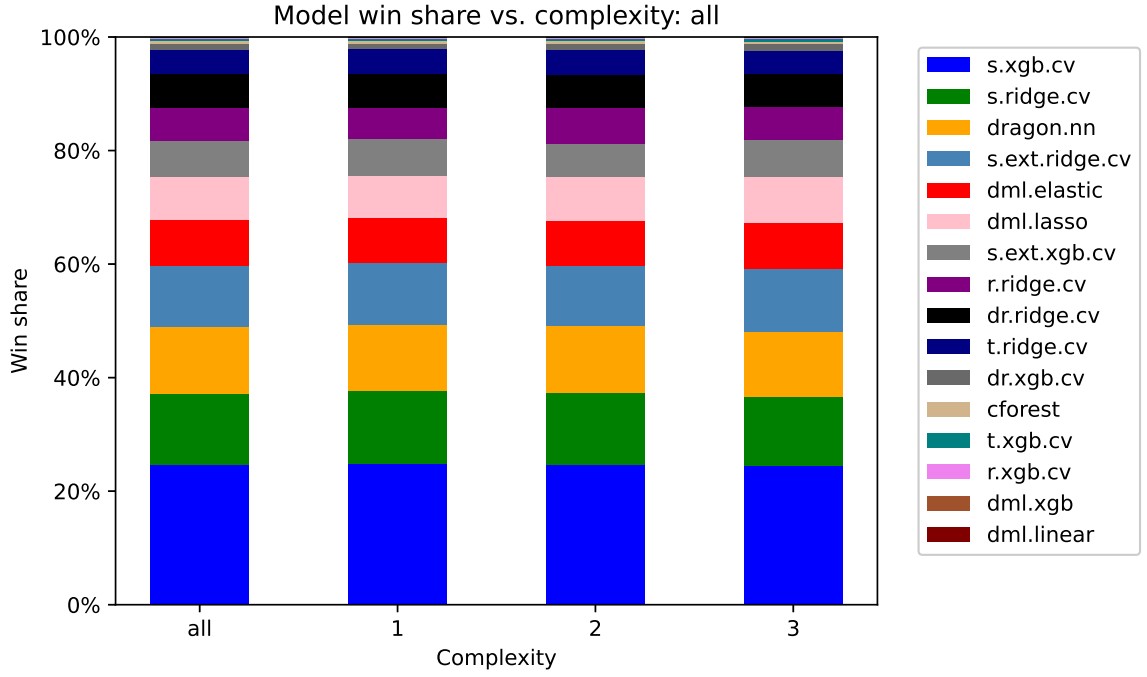

Figure 8: Model win share by assignment mechanism complexity: all RCT datasets

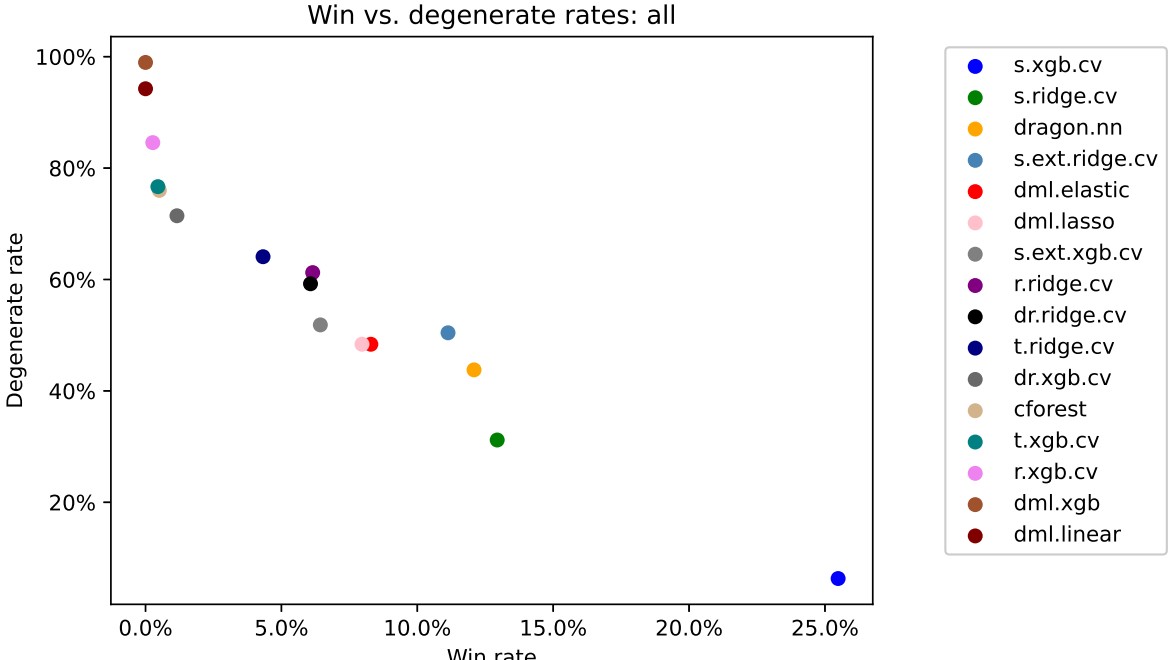

Figure 9: Win share vs. degenerate rate, by model, all RCT datasets

# H    TABLE 1 WITH STANDARD ERROR AND 95% CONFIDENCE INTERVALS

Table 5: Model comparison: summary of 43,200 datasets. The standard error and 95% confidence interval come from 10,000 bootstrapping

| Model | Win Rate Mean (SE) | Win Rate 95% CI | Degeneracy Rate Mean (SE) | Degeneracy Rate 95% CI | Avg Rank Mean (SE) | Avg Rank 95% CI |
|---|---|---|---|---|---|---|
| cforest | 0.005 ( 0.000 ) | (0.004, 0.006) | 0.760 ( 0.002 ) | (0.756, 0.764) | 10.5 ( 0.0 ) | (10.4, 10.5) |
| dml.elastic | 0.083 ( 0.001 ) | (0.080, 0.086) | 0.484 ( 0.002 ) | (0.479, 0.488) | 5.6 ( 0.0 ) | (5.6, 5.6) |
| dml.lasso | 0.080 ( 0.001 ) | (0.077, 0.082) | 0.484 ( 0.002 ) | (0.479, 0.489) | 5.7 ( 0.0 ) | (5.6, 5.7) |
| dml.linear | 0.000 ( 0.000 ) | (0.000, 0.000) | 0.942 ( 0.001 ) | (0.940, 0.945) | 14.3 ( 0.0 ) | (14.3, 14.3) |
| dml.xgb | 0.000 ( 0.000 ) | (0.000, 0.000) | 0.990 ( 0.000 ) | (0.989, 0.991) | 15.9 ( 0.0 ) | (15.9, 15.9) |
| dr.ridge.cv | 0.061 ( 0.001 ) | (0.058, 0.063) | 0.592 ( 0.002 ) | (0.588, 0.597) | 7.1 ( 0.0 ) | (7.1, 7.2) |
| dr.xgb.cv | 0.012 ( 0.001 ) | (0.011, 0.013) | 0.714 ( 0.002 ) | (0.710, 0.719) | 9.9 ( 0.0 ) | (9.8, 9.9) |
| dragon.nn | 0.121 ( 0.002 ) | (0.118, 0.124) | 0.438 ( 0.002 ) | (0.433, 0.443) | 5.1 ( 0.0 ) | (5.1, 5.2) |
| r.ridge.cv | 0.062 ( 0.001 ) | (0.059, 0.064) | 0.612 ( 0.002 ) | (0.608, 0.617) | 8.7 ( 0.0 ) | (8.7, 8.8) |
| r.xgb.cv | 0.003 ( 0.000 ) | (0.002, 0.003) | 0.846 ( 0.002 ) | (0.842, 0.849) | 12.4 ( 0.0 ) | (12.4, 12.4) |
| s.ext.ridge.cv | 0.111 ( 0.002 ) | (0.108, 0.114) | 0.504 ( 0.002 ) | (0.499, 0.509) | 5.6 ( 0.0 ) | (5.6, 5.6) |
| s.ext.xgb.cv | 0.064 ( 0.001 ) | (0.062, 0.067) | 0.518 ( 0.002 ) | (0.514, 0.523) | 6.9 ( 0.0 ) | (6.9, 6.9) |
| s.ridge.cv | 0.129 ( 0.002 ) | (0.126, 0.133) | 0.312 ( 0.002 ) | (0.307, 0.316) | 4.2 ( 0.0 ) | (4.2, 4.2) |
| s.xgb.cv | 0.255 ( 0.002 ) | (0.251, 0.259) | 0.063 ( 0.001 ) | (0.061, 0.066) | 4.4 ( 0.0 ) | (4.4, 4.4) |
| t.ridge.cv | 0.043 ( 0.001 ) | (0.041, 0.045) | 0.641 ( 0.002 ) | (0.636, 0.646) | 8.2 ( 0.0 ) | (8.2, 8.3) |
| t.xgb.cv | 0.005 ( 0.000 ) | (0.004, 0.005) | 0.767 ( 0.002 ) | (0.763, 0.771) | 11.4 ( 0.0 ) | (11.4, 11.4) |

# I SECTION 4 DATASET LEVEL STATISTICS

## I.1 SUMMARY OF DATASET SPECIFIC BEHAVIOR

*Does models' relative performance vary by amount of training data, treatment imbalance, or level of nonlinearity in assignment mechanism?* Looking at the aggregated results (Appendix G), we found moderate fluctuation of model performance when varying these drivers. This is likely because variations averages out. Dataset level statistics shows a different picture. We present selected results in this section and leave more in Appendix I.

*Training dataset size* can have large impact on model performance. On `criteo`, Win share of R-learner (`r.ridge.cv`) increase from 7% with 1k estimation data, to 26% with 8k estimation data. See Figure 10.

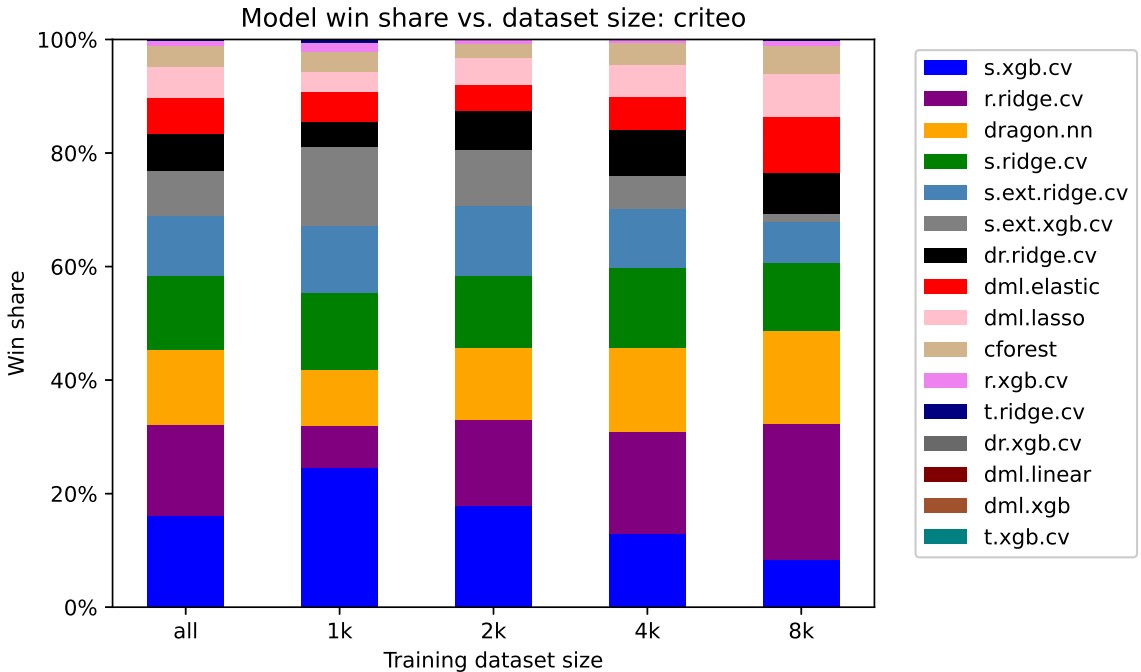

Figure 10: Training size on `criteo`

*Treatment imbalance* can have large impact on model performance. On `natcity`, win share of DR learner is 40% with balanced treatment, and 16% when treated ratio is 0.9. See Figure 11.

*Level of nonlinearity in assignment mechanism*, we found, has limited impact on model performance. This is partly, we think, because our propensity estimator (Random Forest with propensity clipping) is flexible enough to fit different degrees of nonlinearity. See Figure 8 in Appendix.

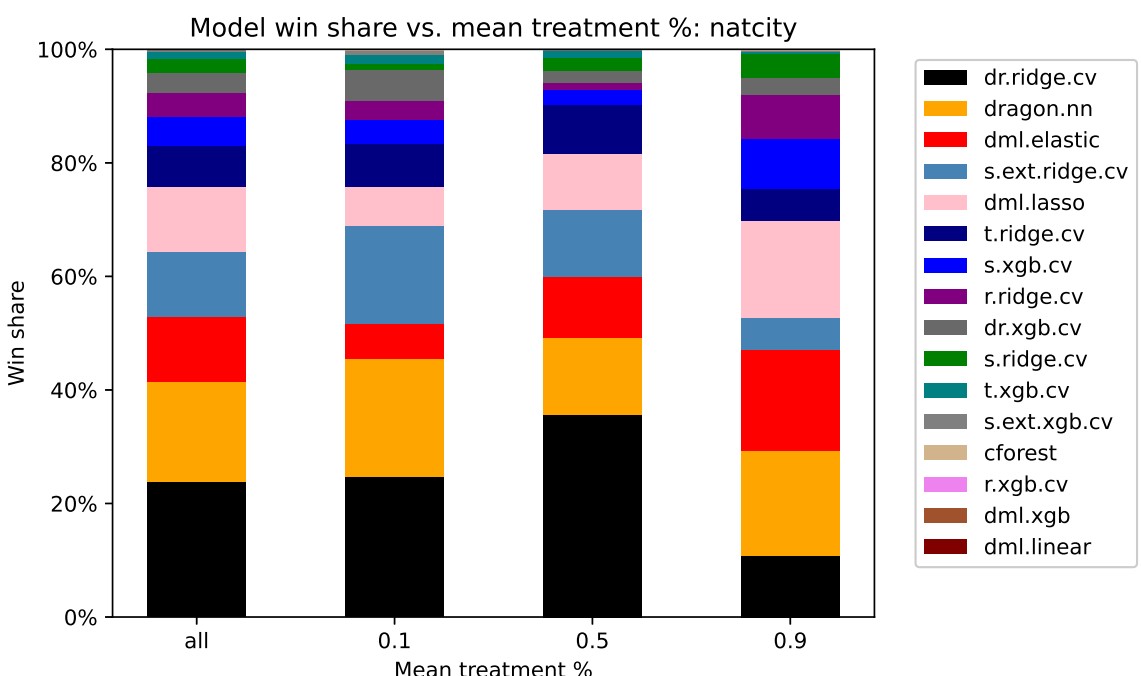

Figure 11: treatment % on `natcity`

## I.2 CRITEO

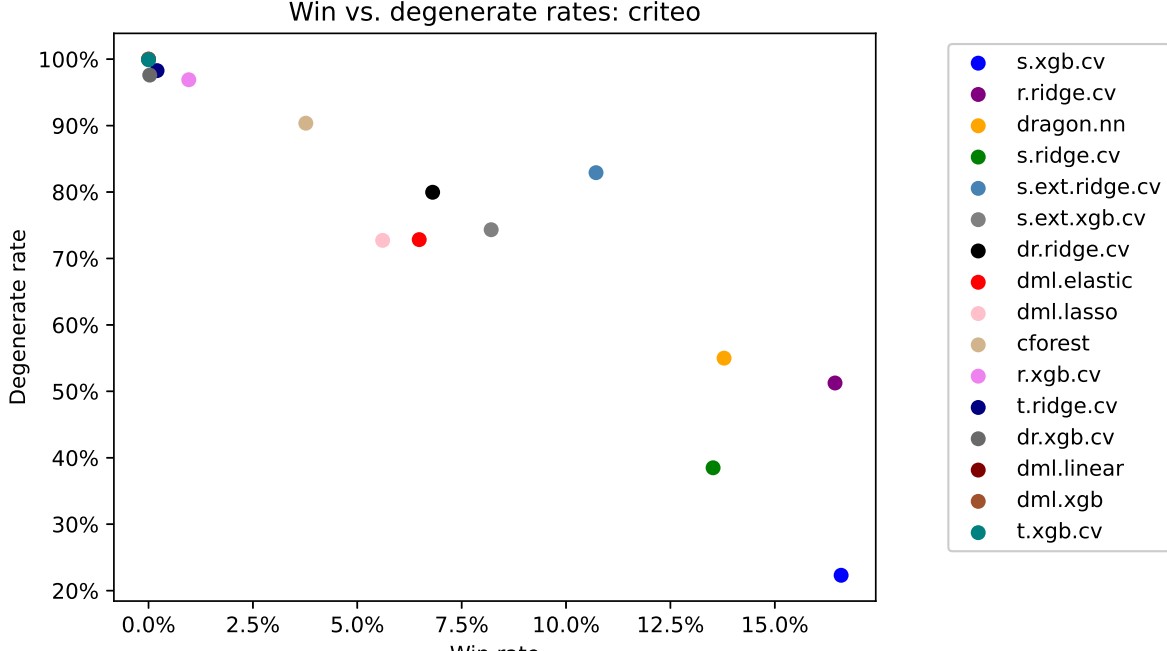

Figure 12: Win share vs. degenerate percentage, by model on `criteo`

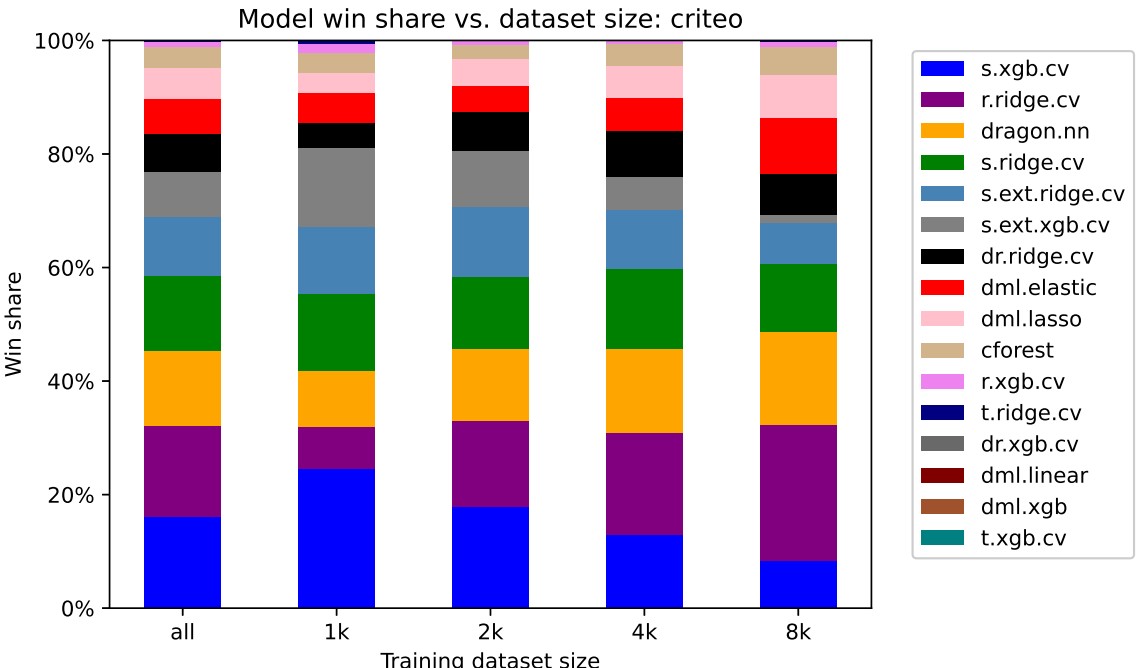

Figure 13: Model win share by estimation data, `criteo`

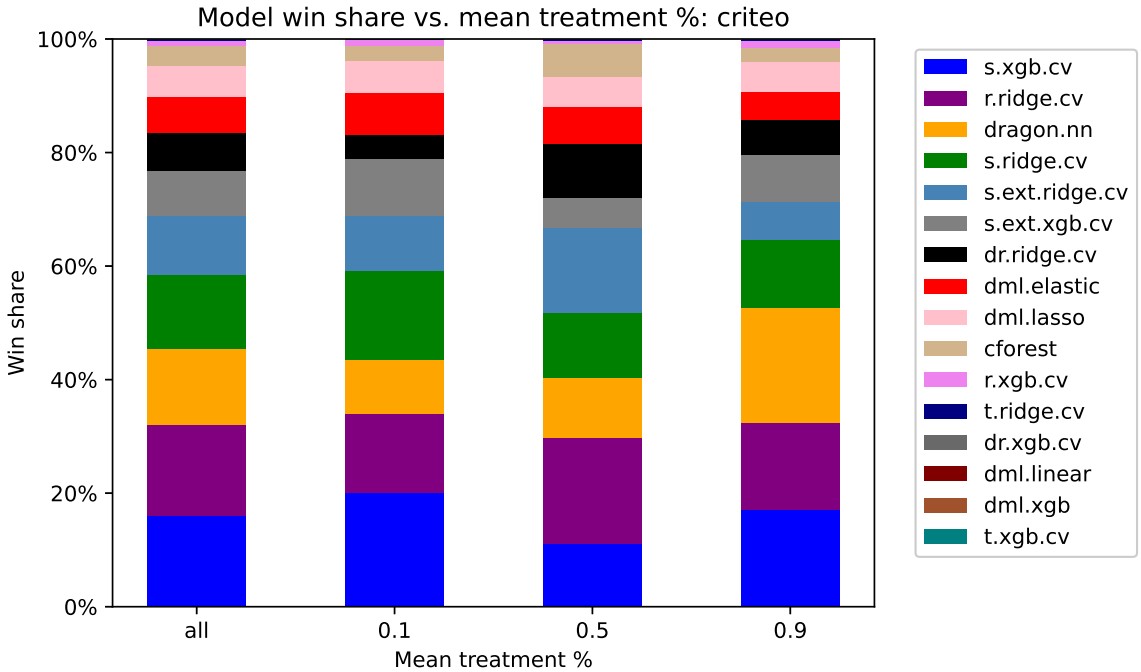

Figure 14: Model win share by treatment ratio, `criteo`

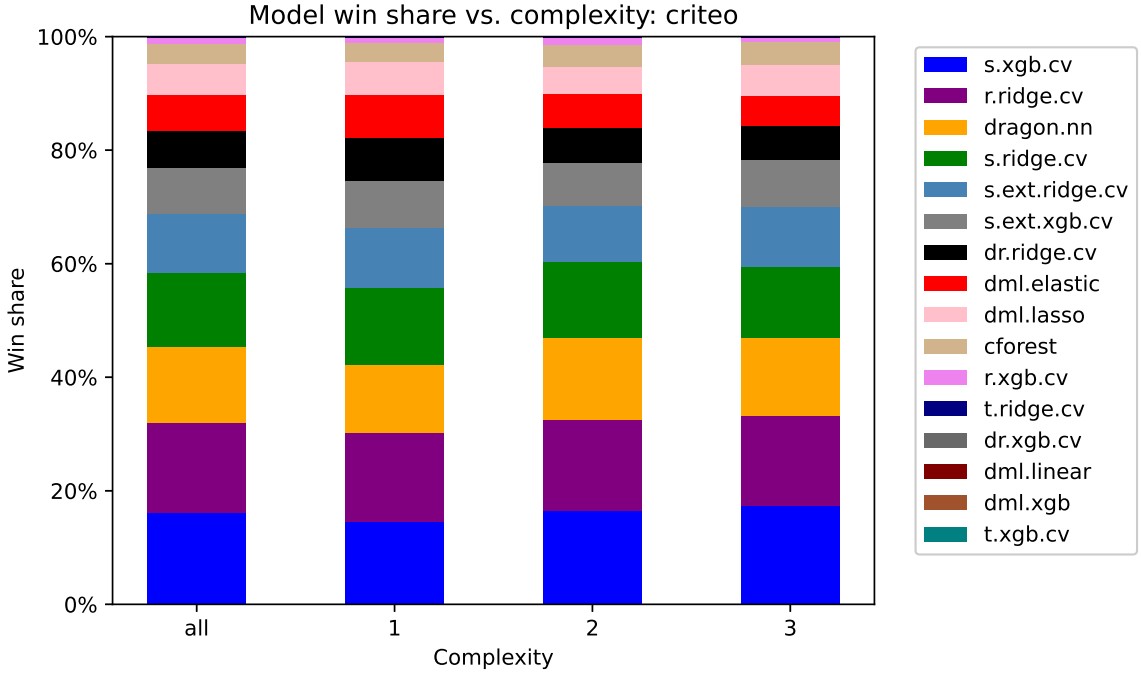

Figure 15: Model win share by assignment mechanism complexity, `criteo`

I.3  `HILLSTROM`

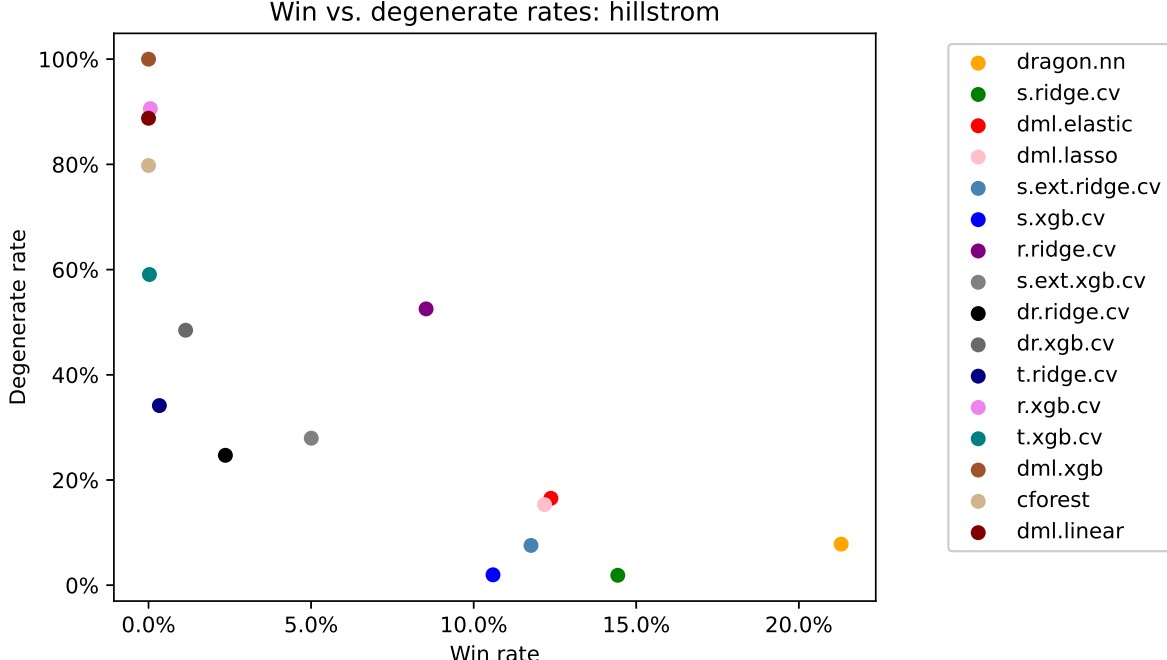

Figure 16: Win share vs. degenerate percentage, by model on `hillstrom`

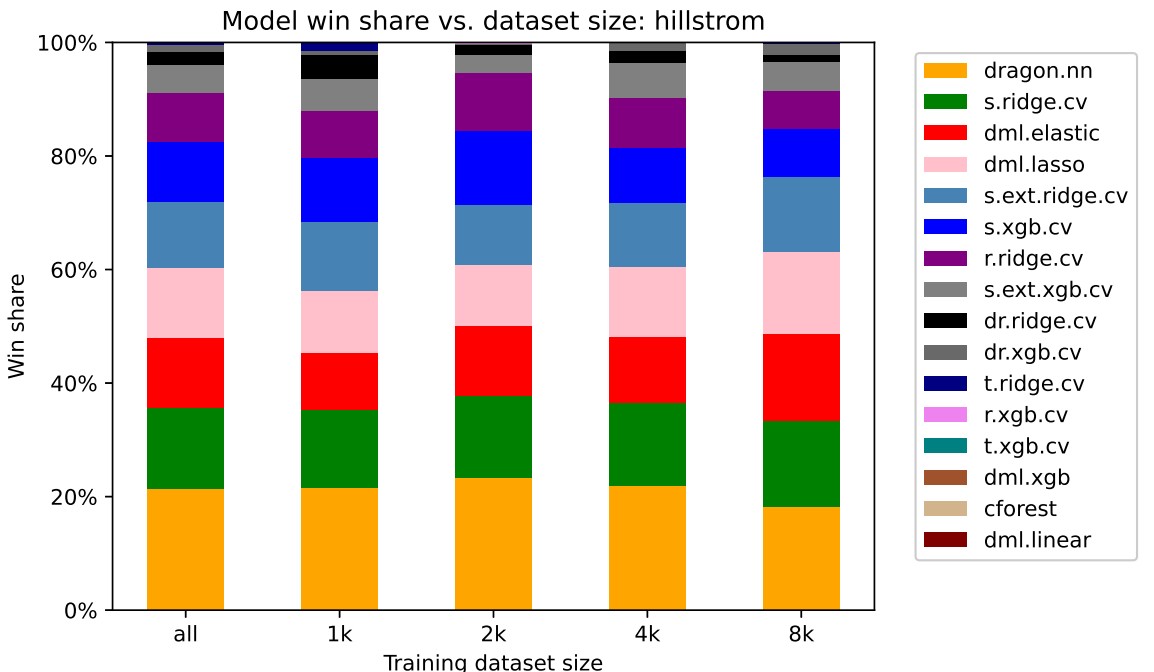

Figure 17: Model win share by estimation data, `hillstrom`

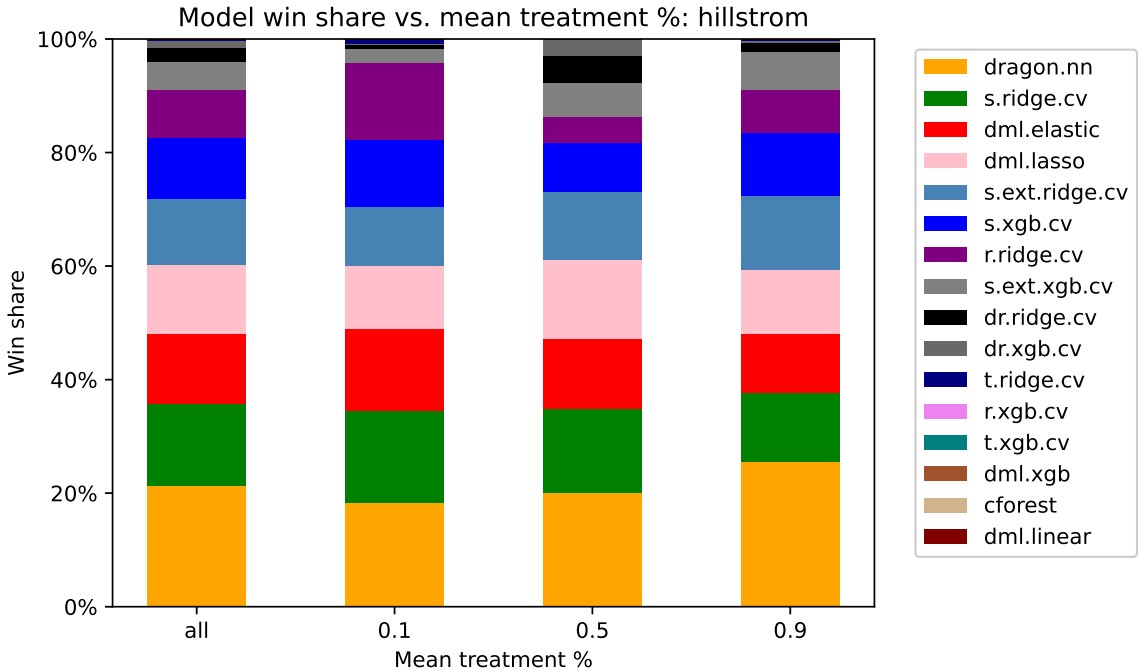

Figure 18: Model win share by treatment ratio, `hillstrom`

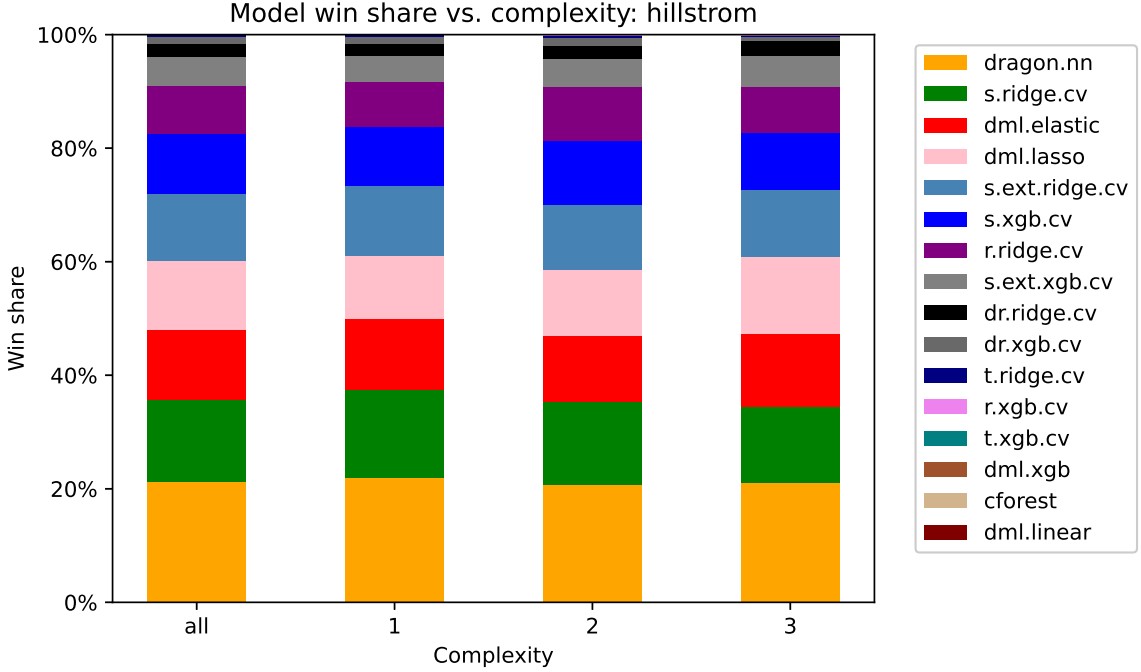

Figure 19: Model win share by assignment mechanism complexity, `hillstrom`

I.4 FERMAN

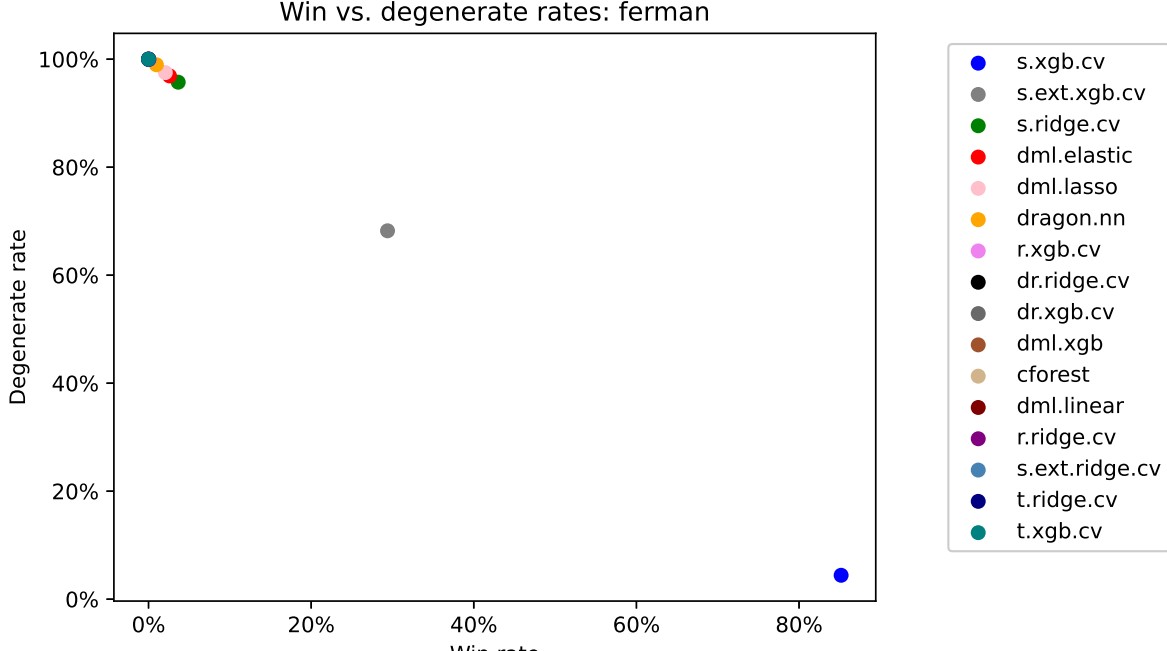

Figure 20: Win share vs. degenerate percentage, by model on ferman

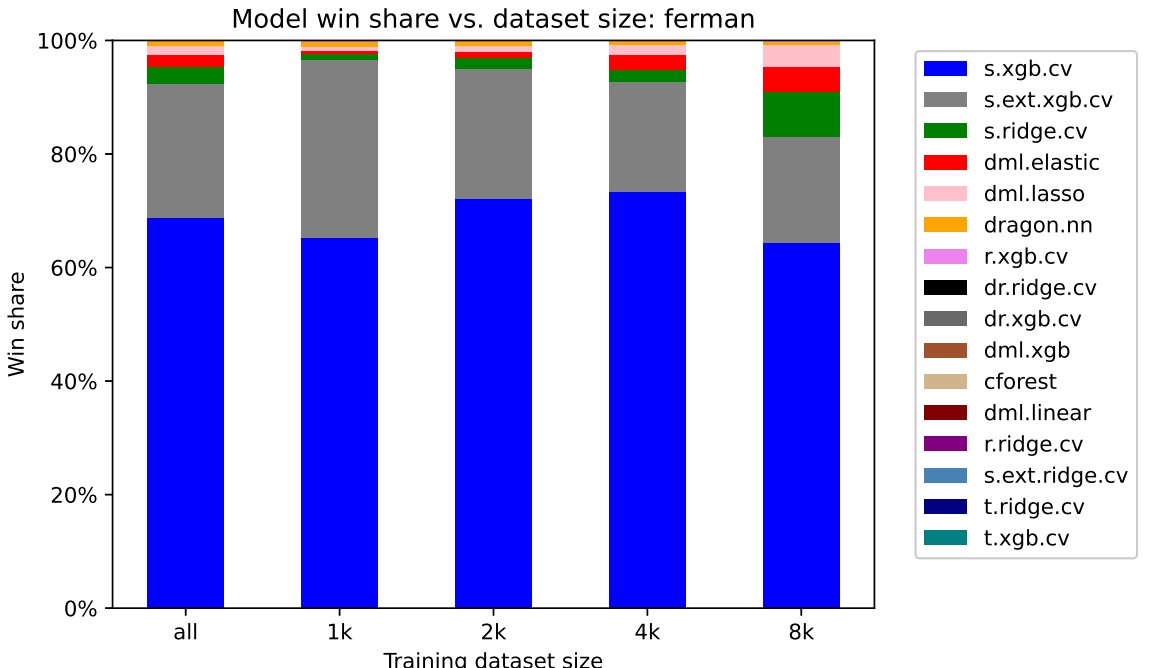

Figure 21: Model win share by estimation data, ferman

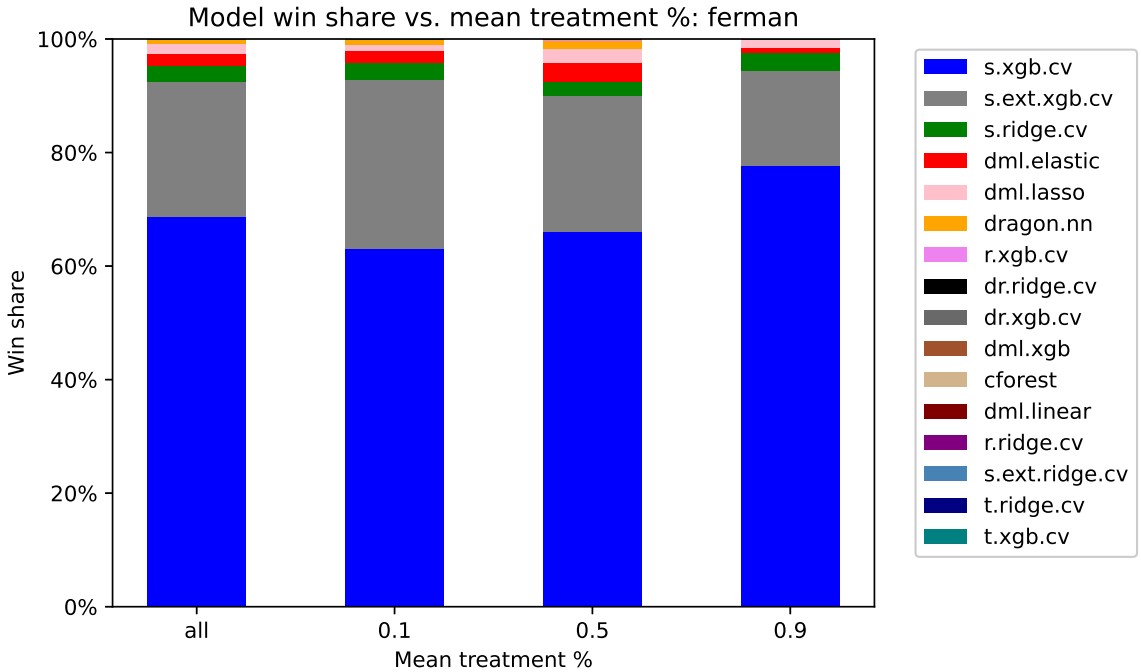

Figure 22: Model win share by treatment ratio, `ferman`

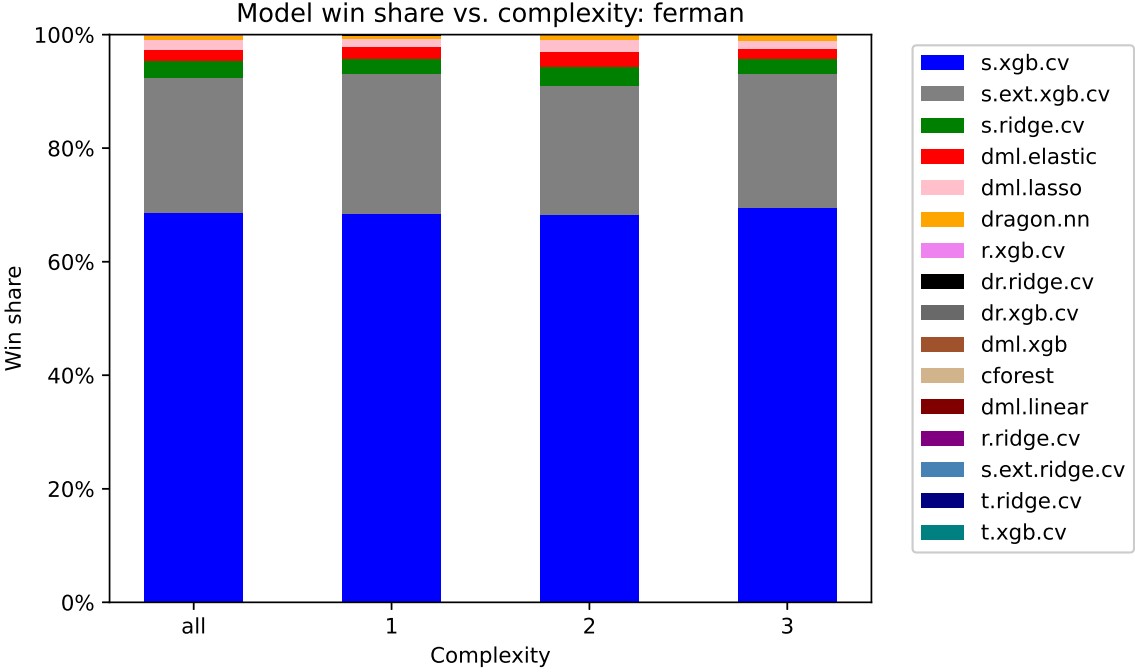

Figure 23: Model win share by assignment mechanism complexity, `ferman`

## I.5 SANDERCOCK

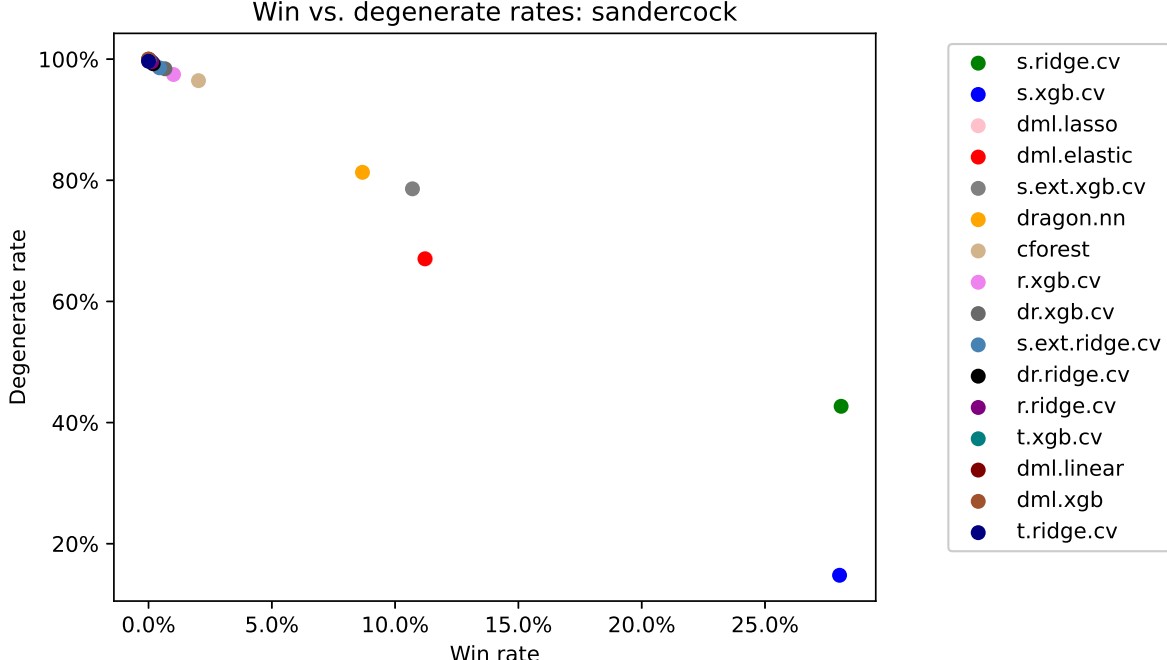

Figure 24: Win share vs. degenerate percentage, by model on `sandercock`

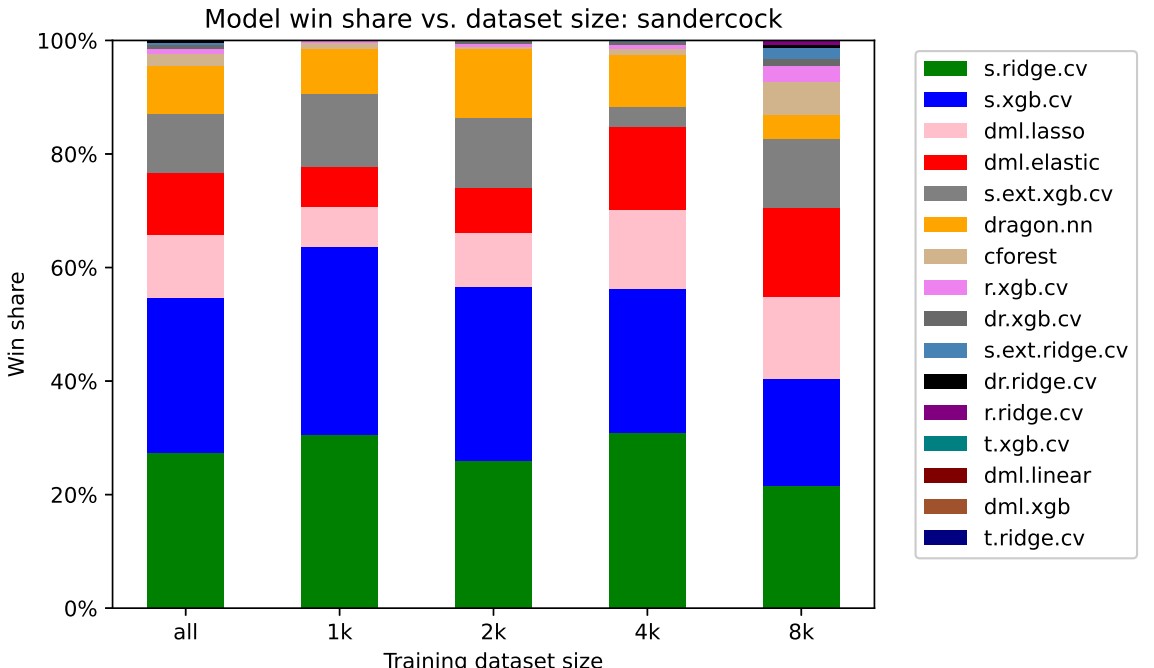

Figure 25: Model win share by estimation data, `sandercock`

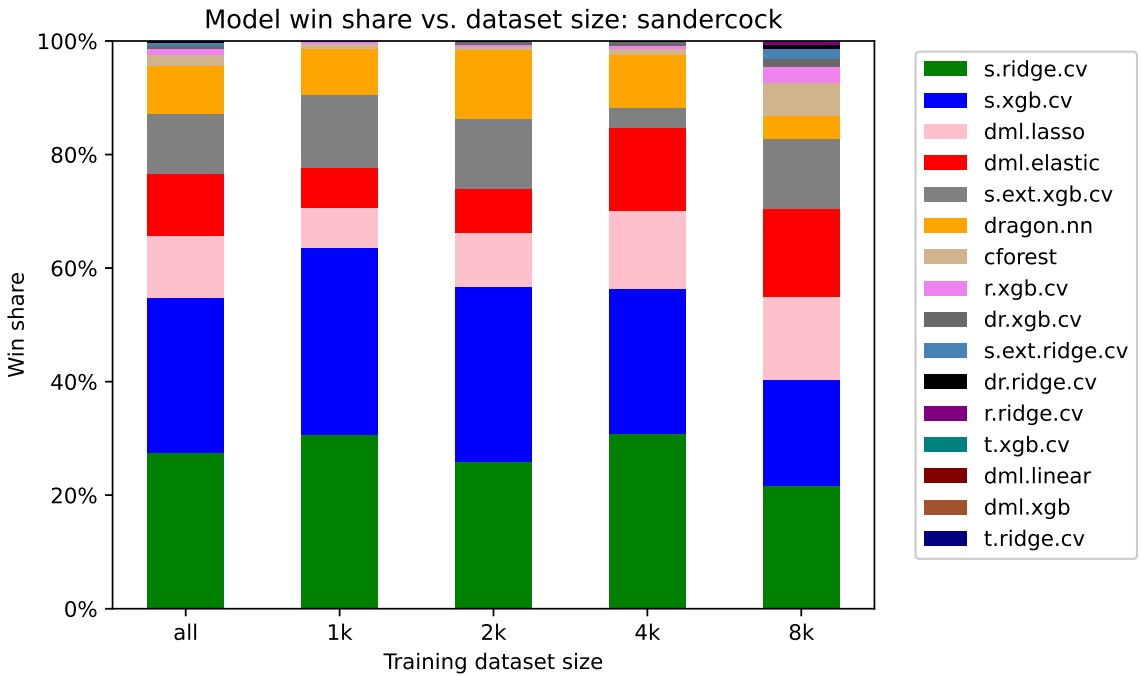

Figure 26: Model win share by treatment ratio, `sandercock`

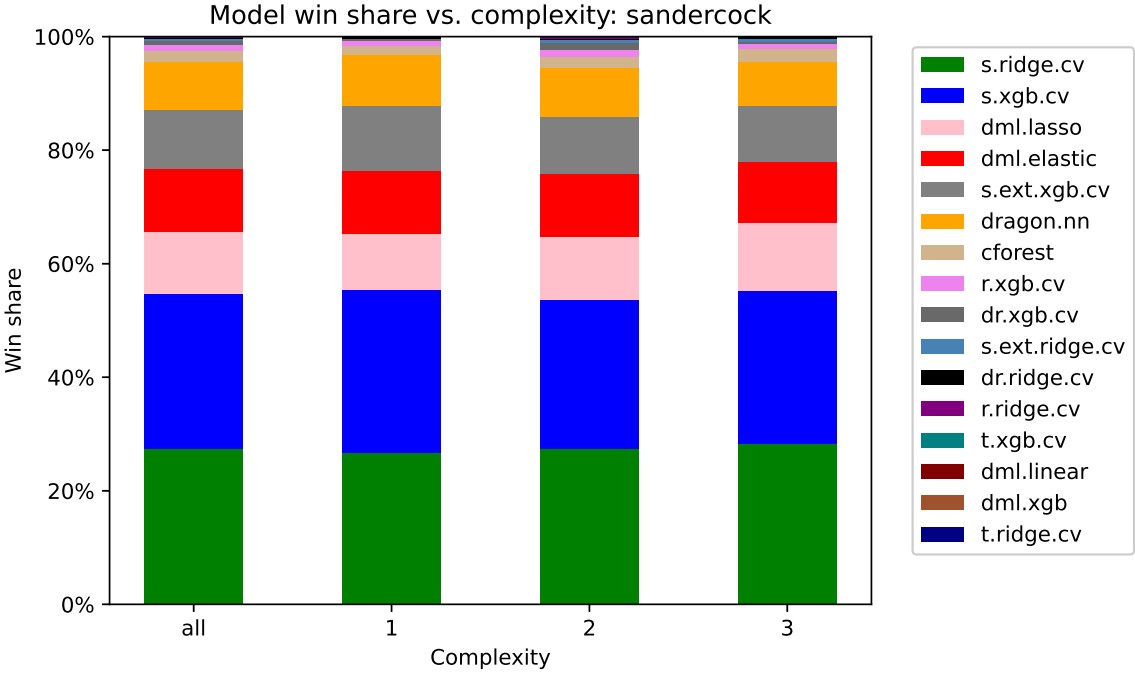

Figure 27: Model win share by assignment mechanism complexity, `sandercock`

I.6 `NATAID`

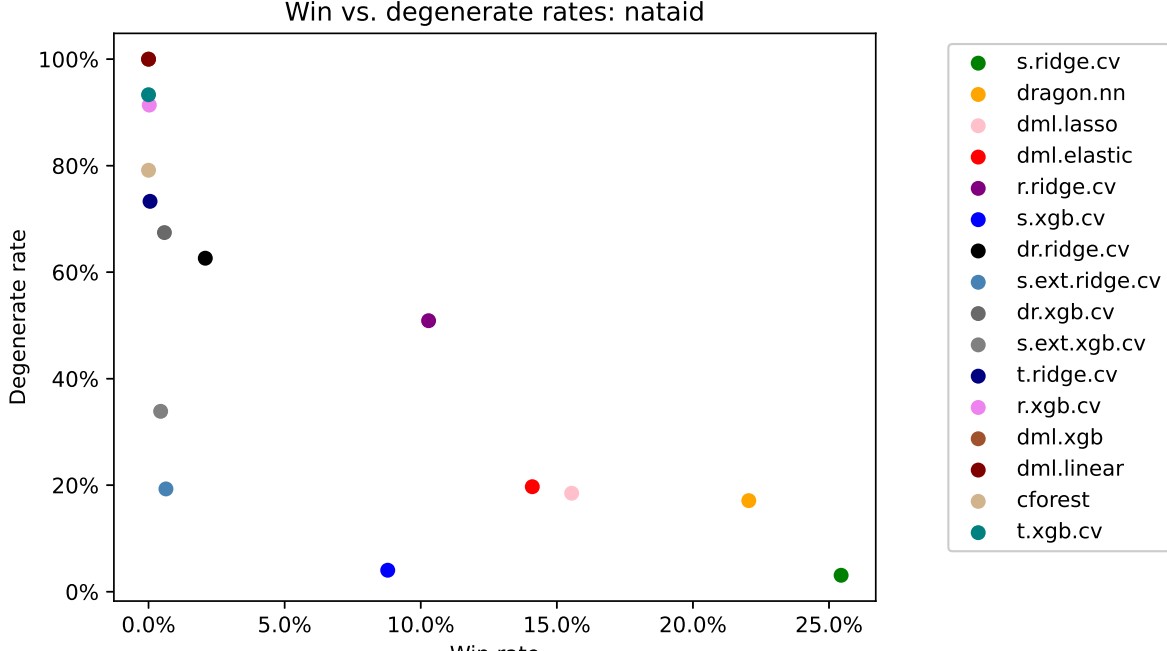

Figure 28: Win share vs. degenerate percentage, by model on `nataid`

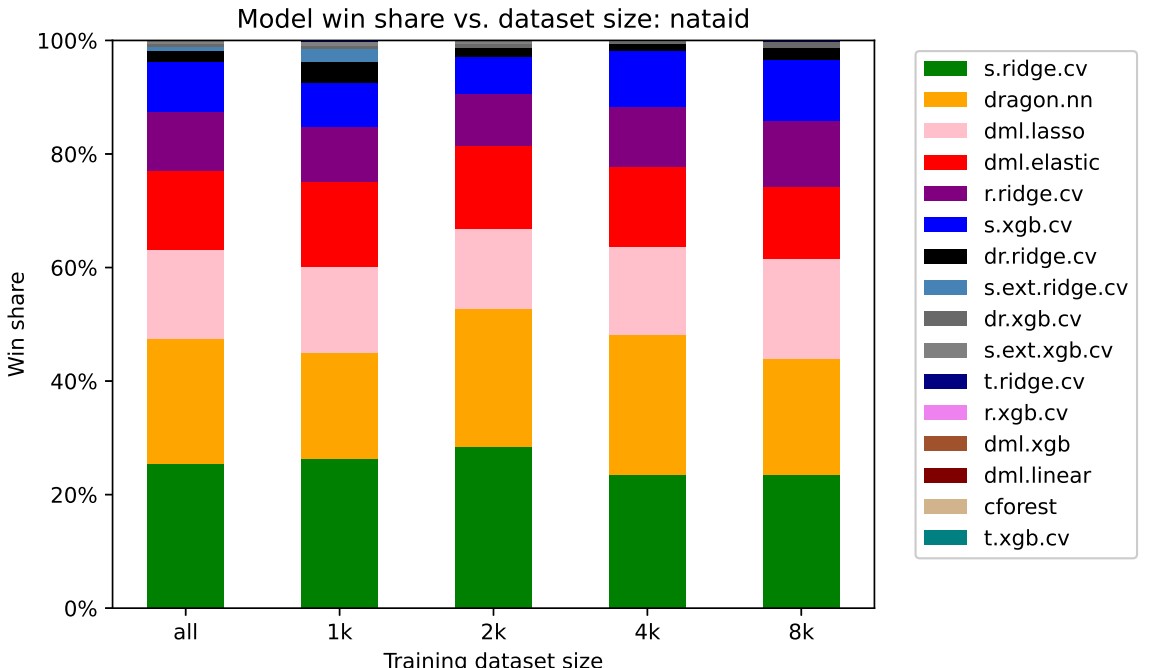

Figure 29: Model win share by estimation data, `nataid`

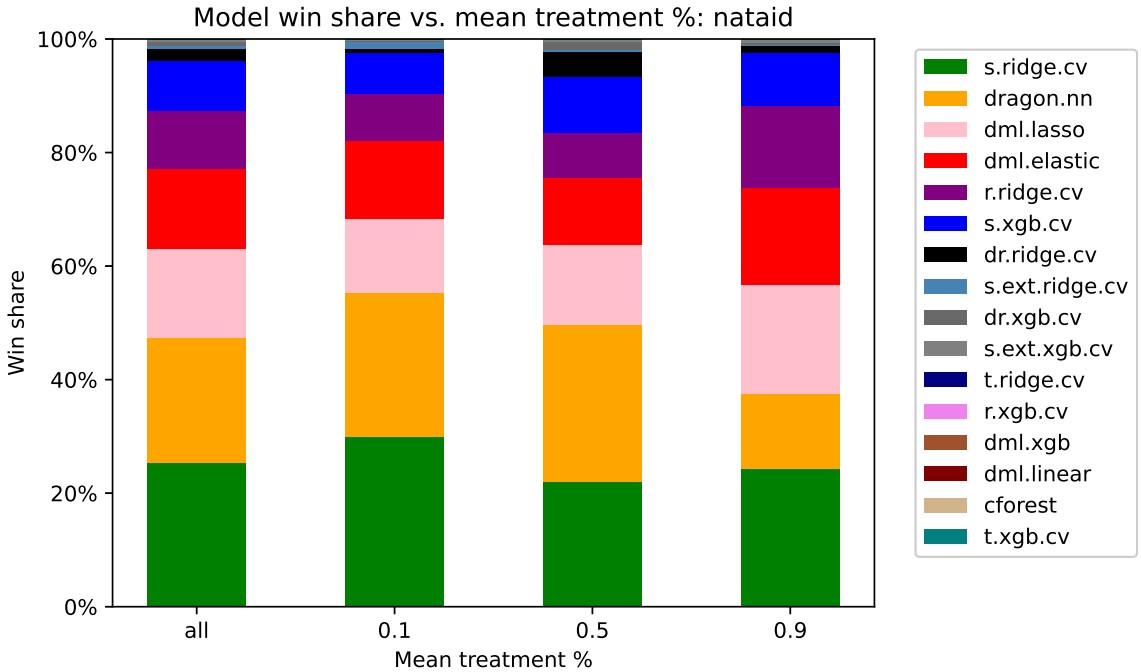

Figure 30: Model win share by treatment ratio, `nataid`

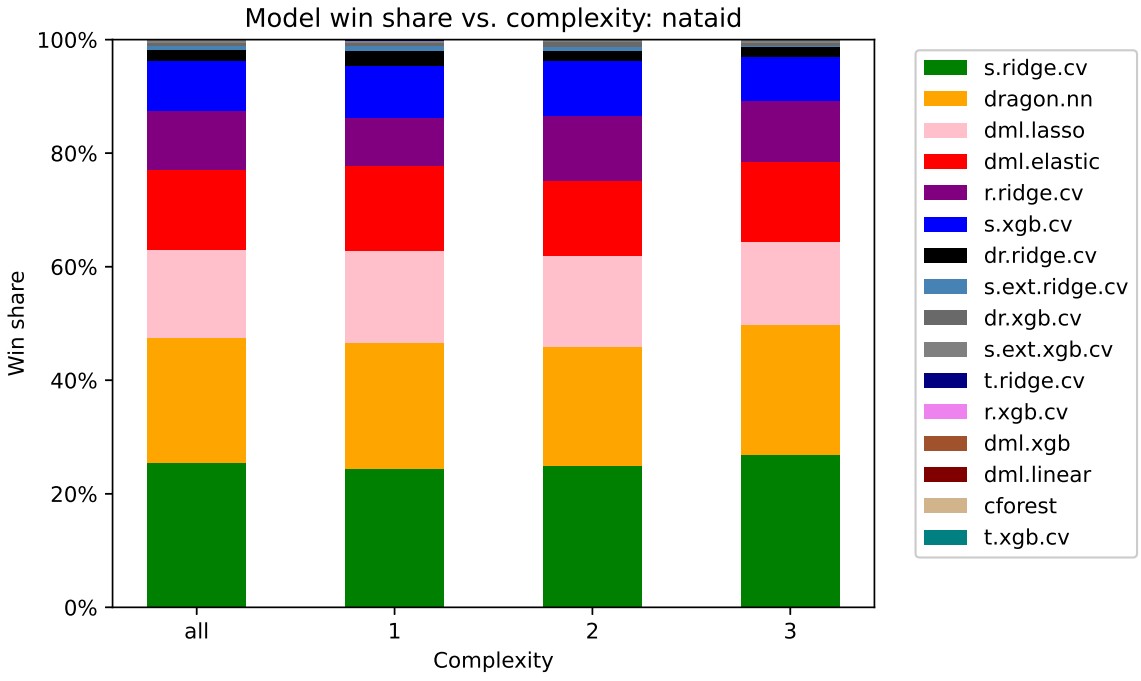

Figure 31: Model win share by assignment mechanism complexity, `nataid`

### I.7  `NATARMS`

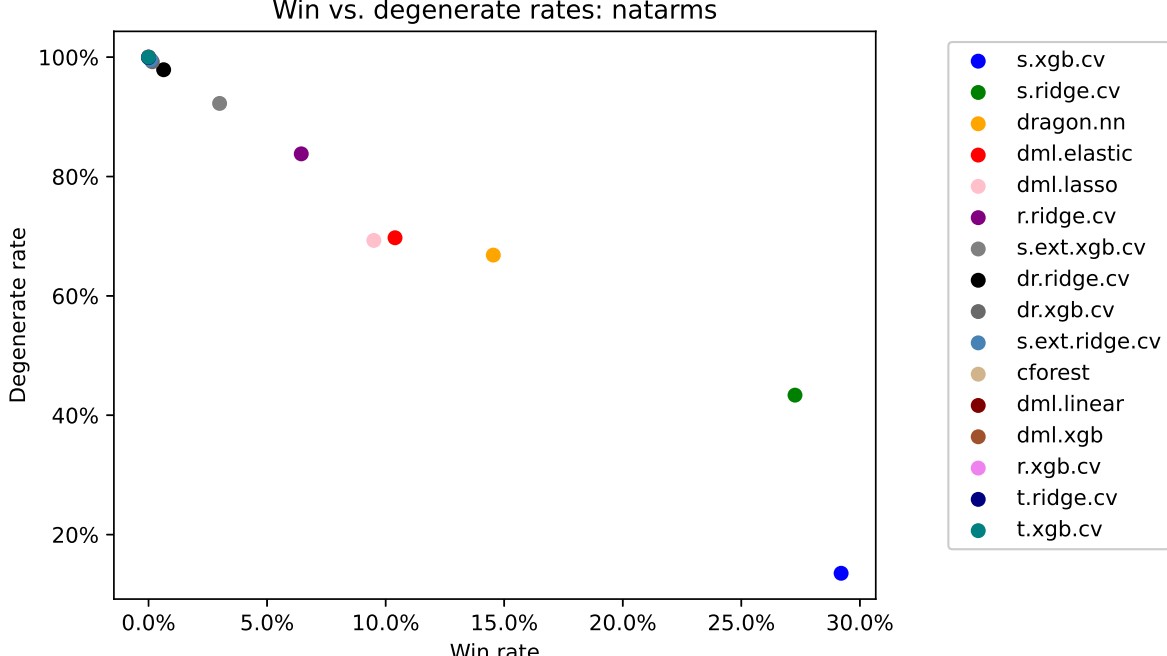

Figure 32: Win share vs. degenerate percentage, by model on `natarms`

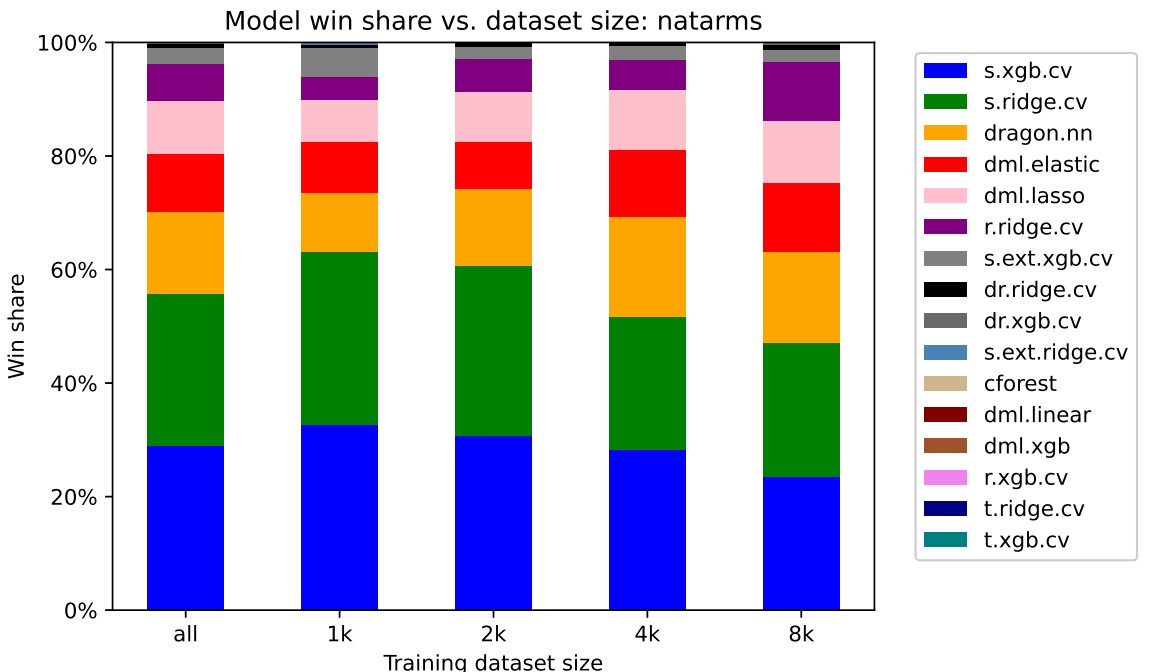

Figure 33: Model win share by estimation data, `natarms`

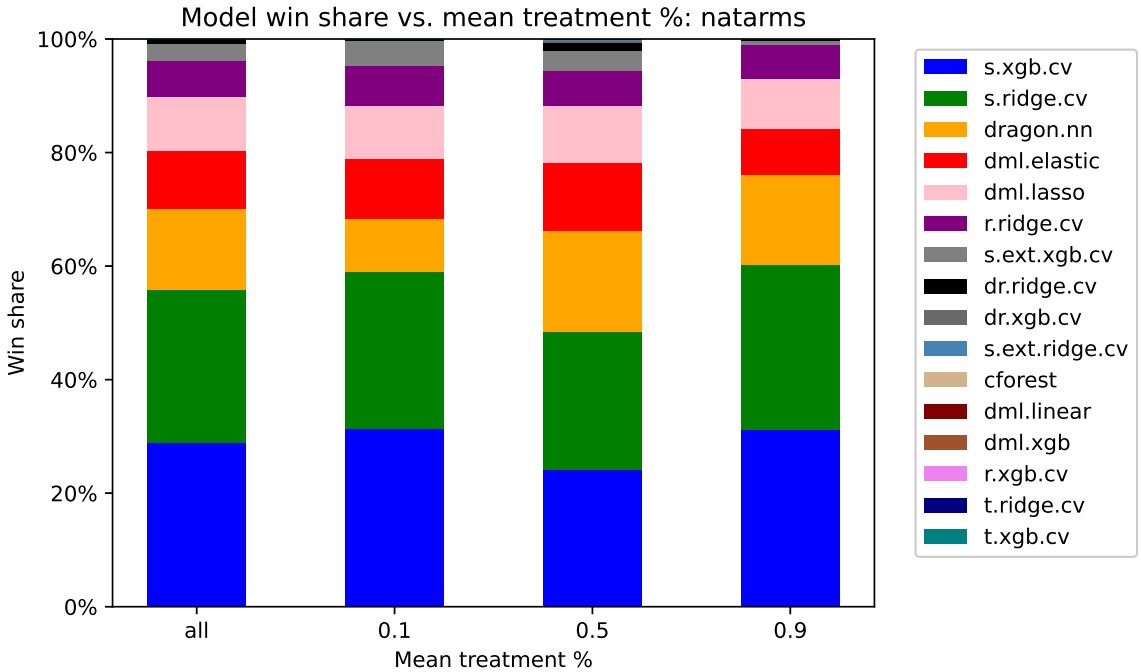

Figure 34: Model win share by treatment ratio, `natarms`

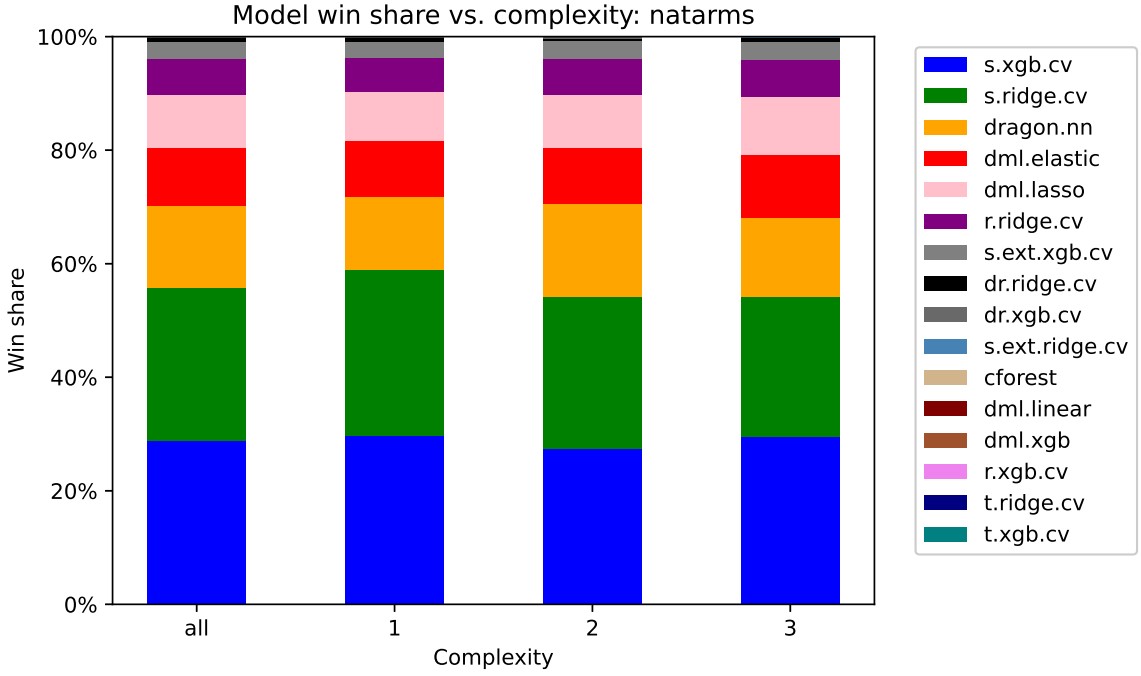

Figure 35: Model win share by assignment mechanism complexity, `natarms`

## I.8 NATCITY

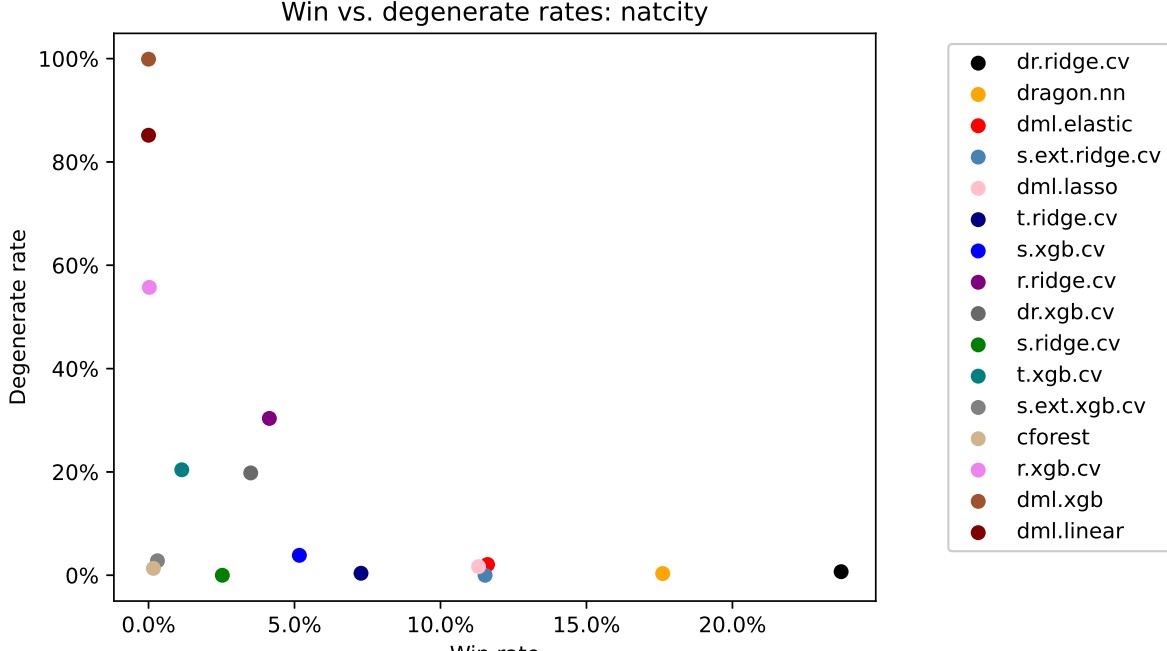

Figure 36: Win share vs. degenerate percentage, by model on natcity

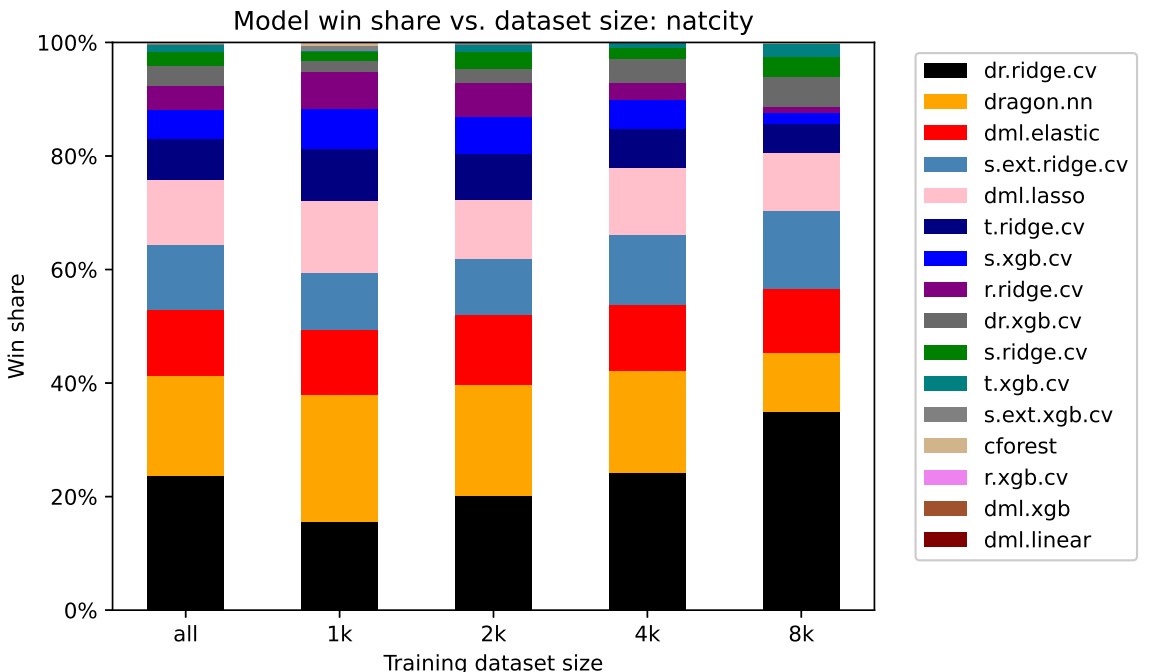

Figure 37: Model win share by estimation data, natcity

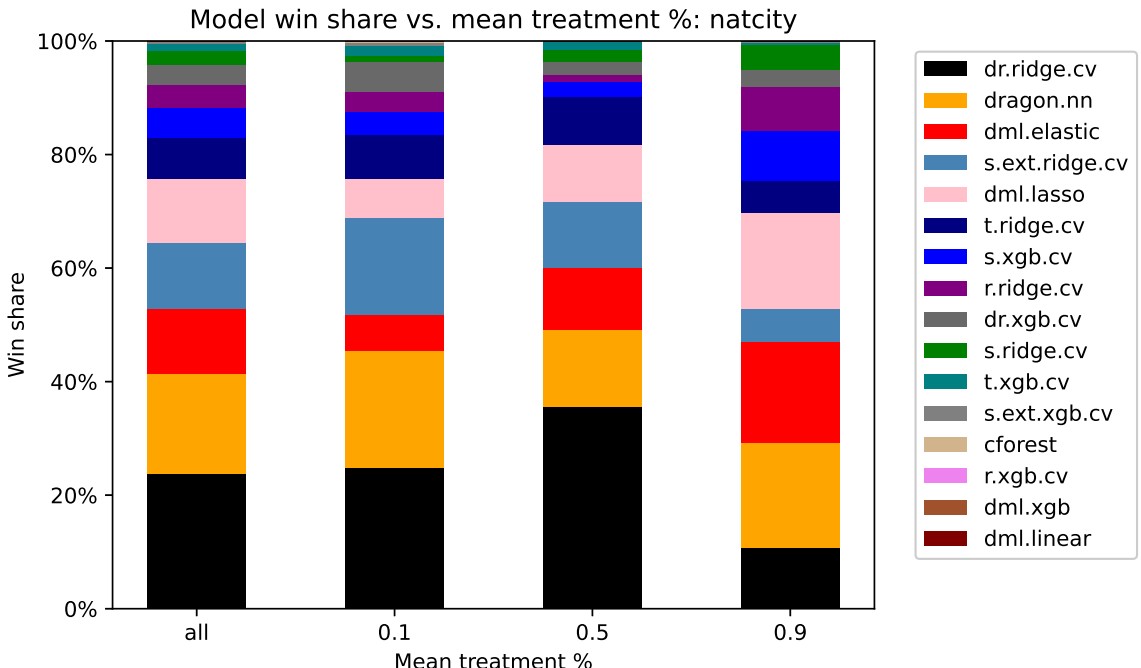

Figure 38: Model win share by treatment ratio, `natcity`

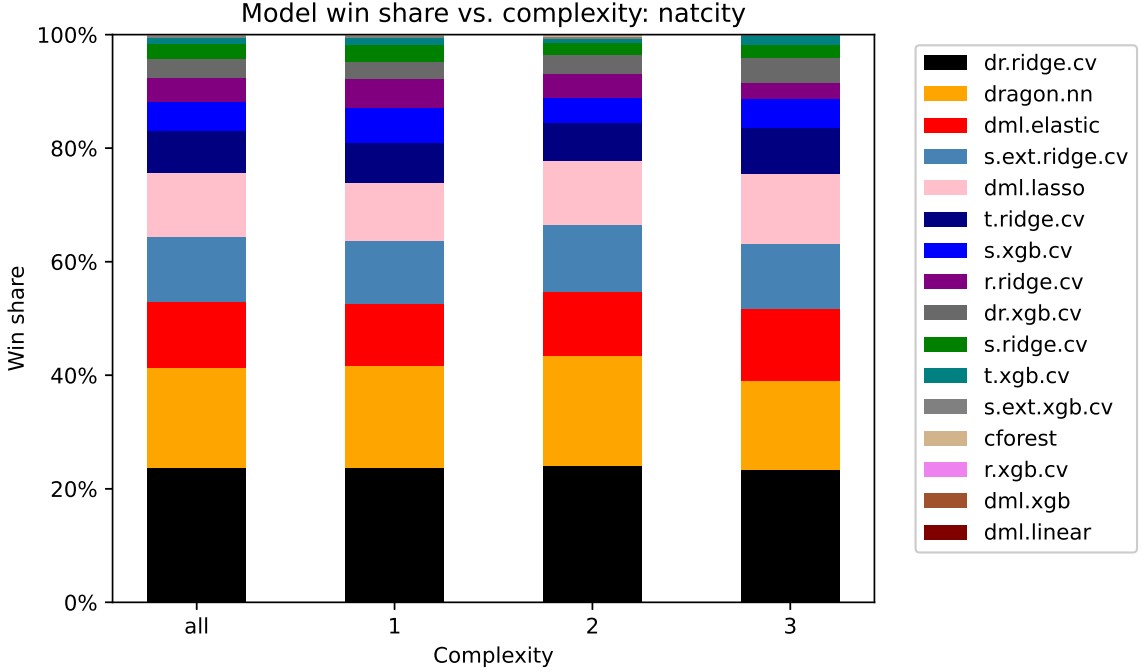

Figure 39: Model win share by assignment mechanism complexity, `natcity`

## I.9 `NATCRIME`

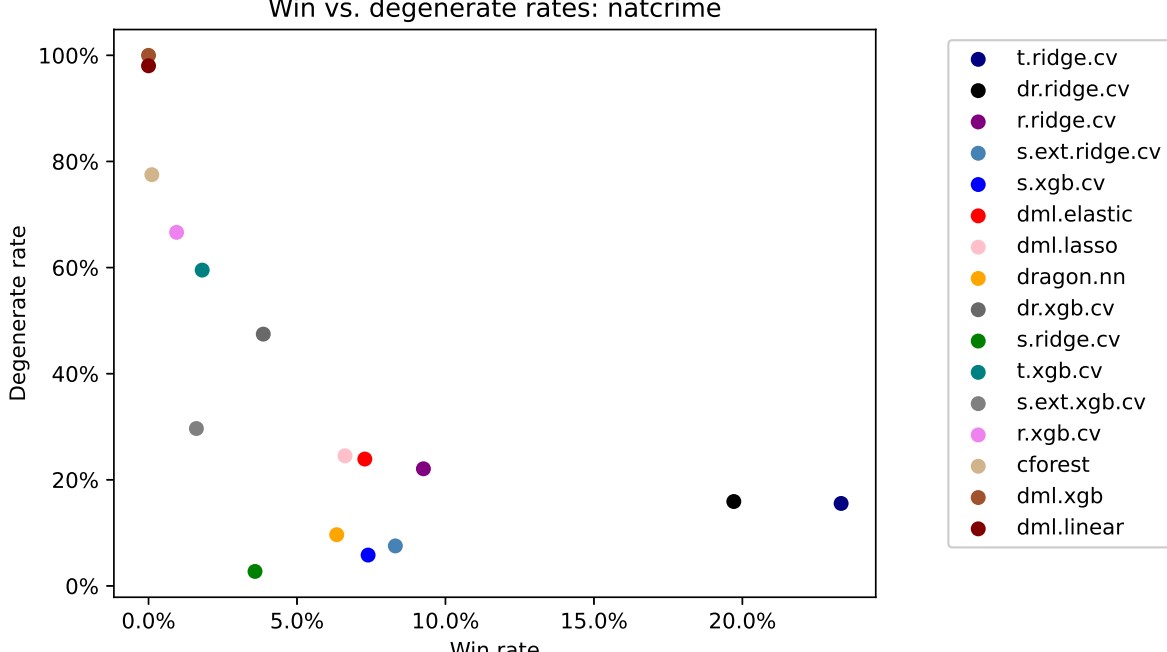

Figure 40: Win share vs. degenerate percentage, by model on `natcrime`

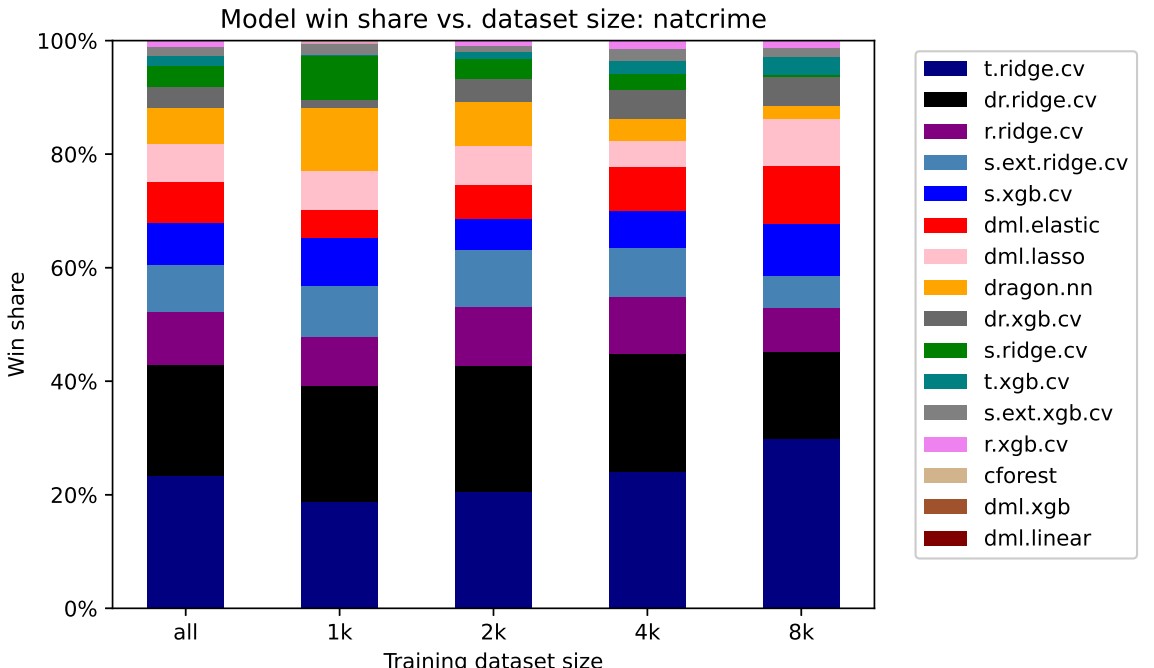

Figure 41: Model win share by estimation data, `natcrime`

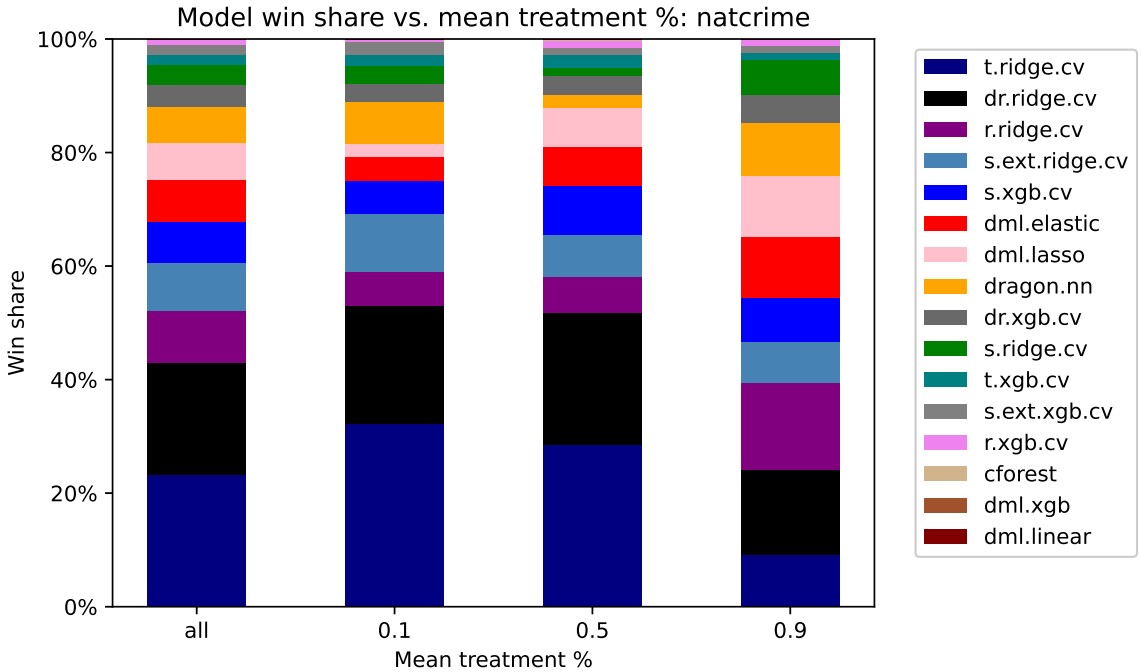

Figure 42: Model win share by treatment ratio, `natcrime`

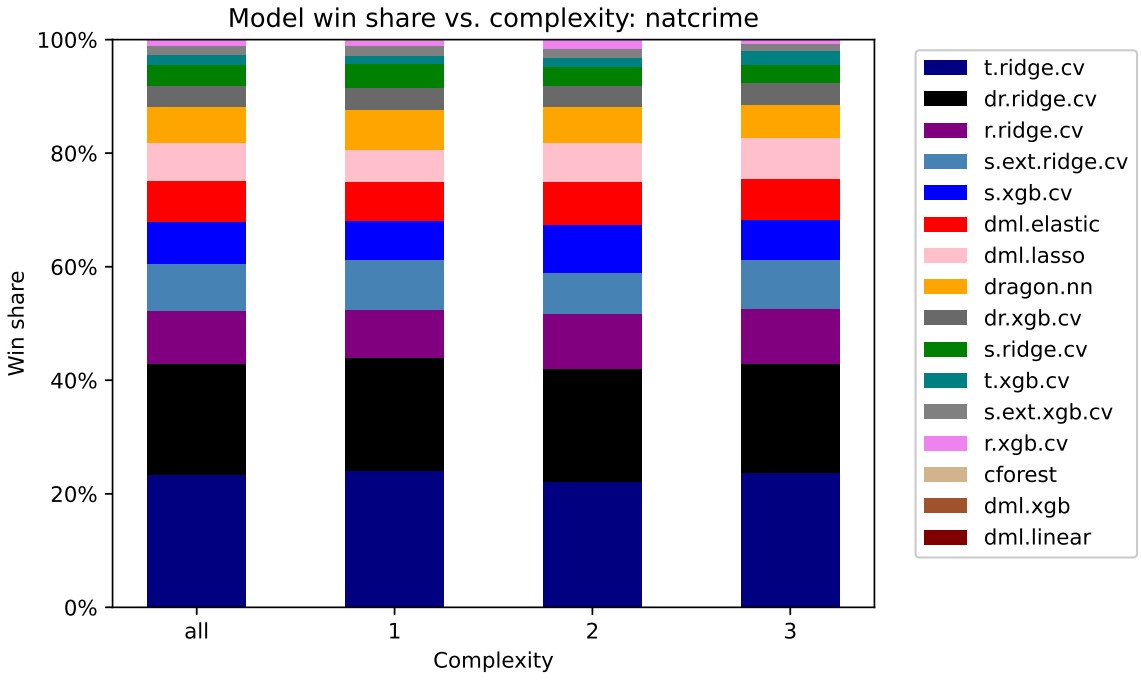

Figure 43: Model win share by assignment mechanism complexity, `natcrime`

## I.10 NATDRUG

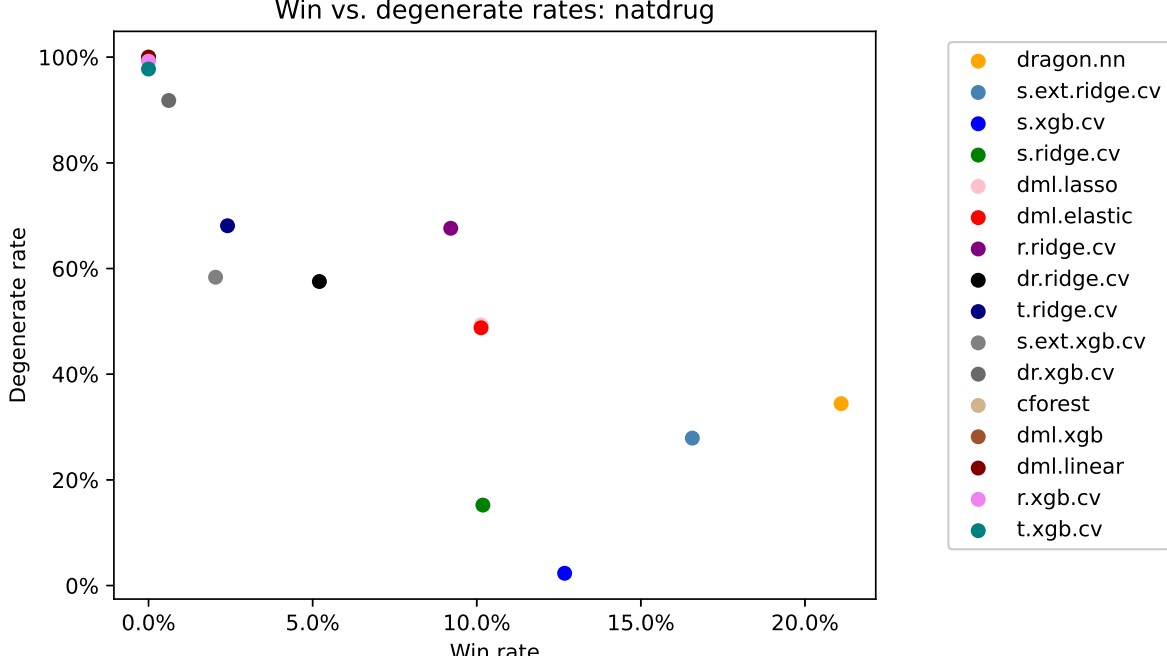

Figure 44: Win share vs. degenerate percentage, by model on natdrug

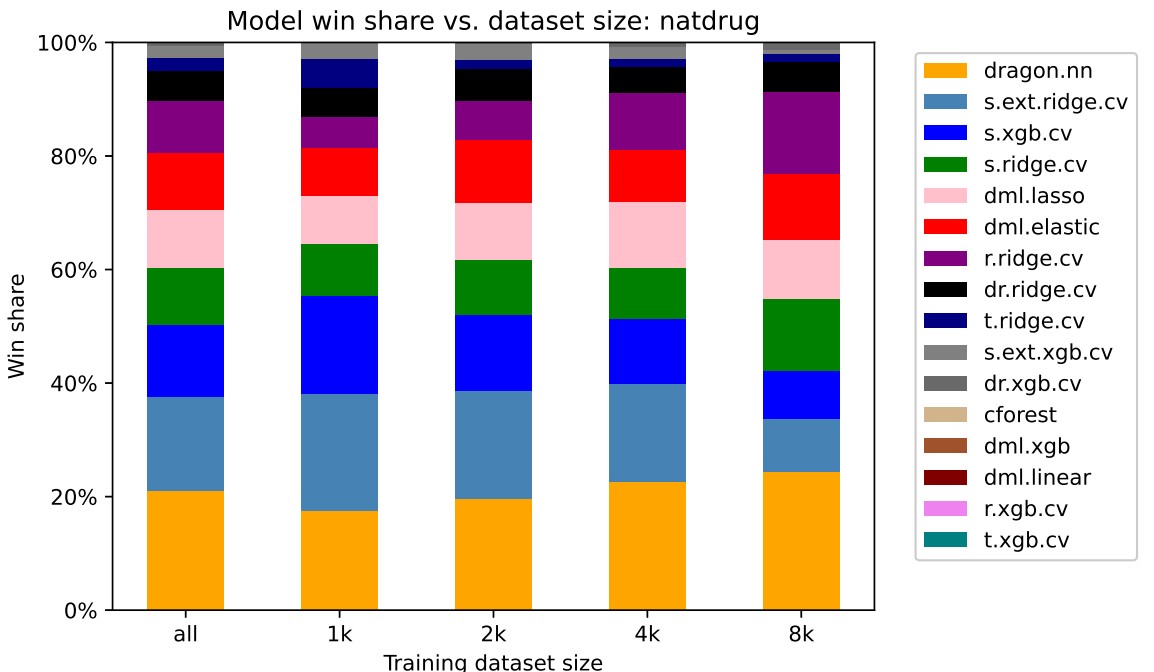

Figure 45: Model win share by estimation data, natdrug

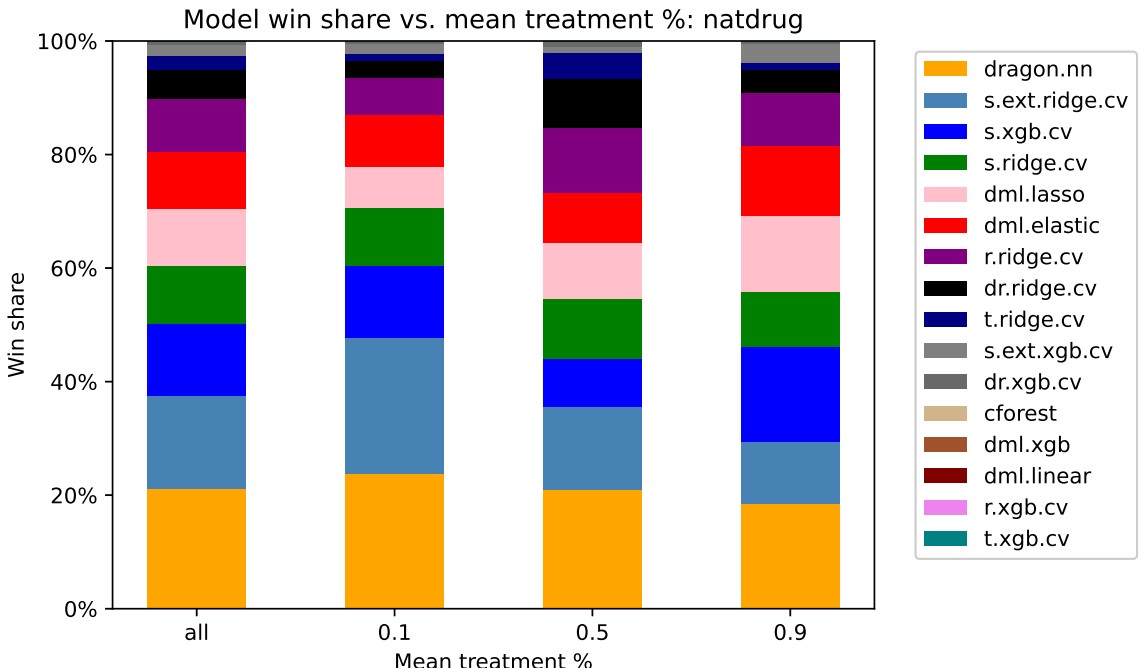

Figure 46: Model win share by treatment ratio, `natdrug`

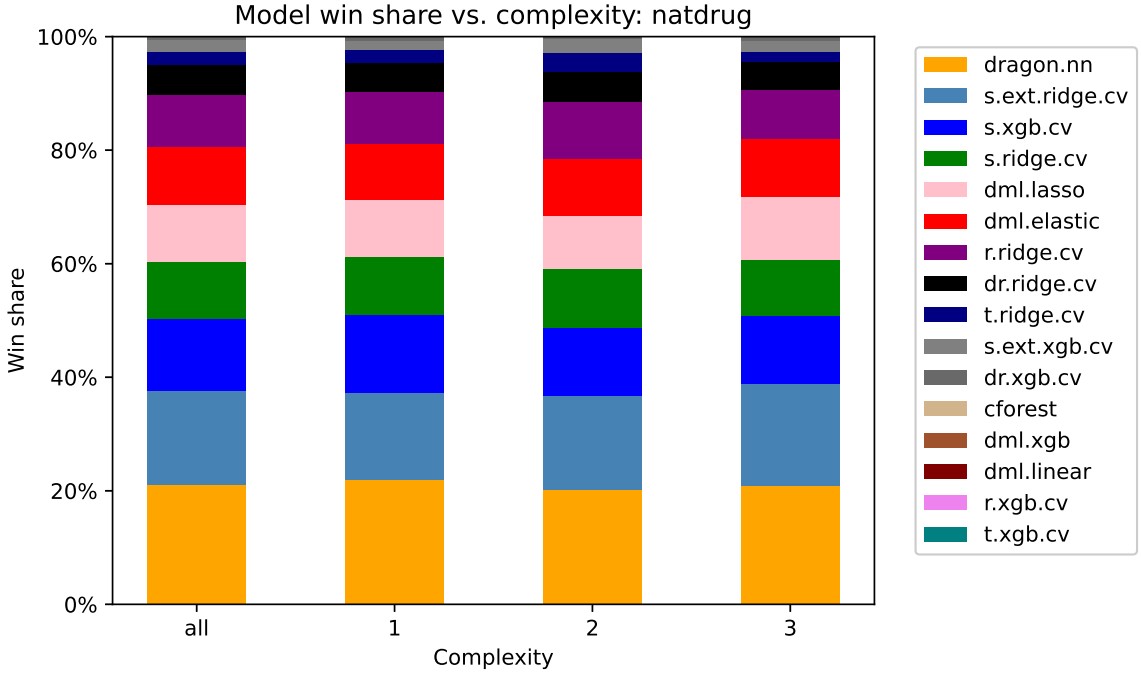

Figure 47: Model win share by assignment mechanism complexity, `natdrug`

## I.11 NATEDUC

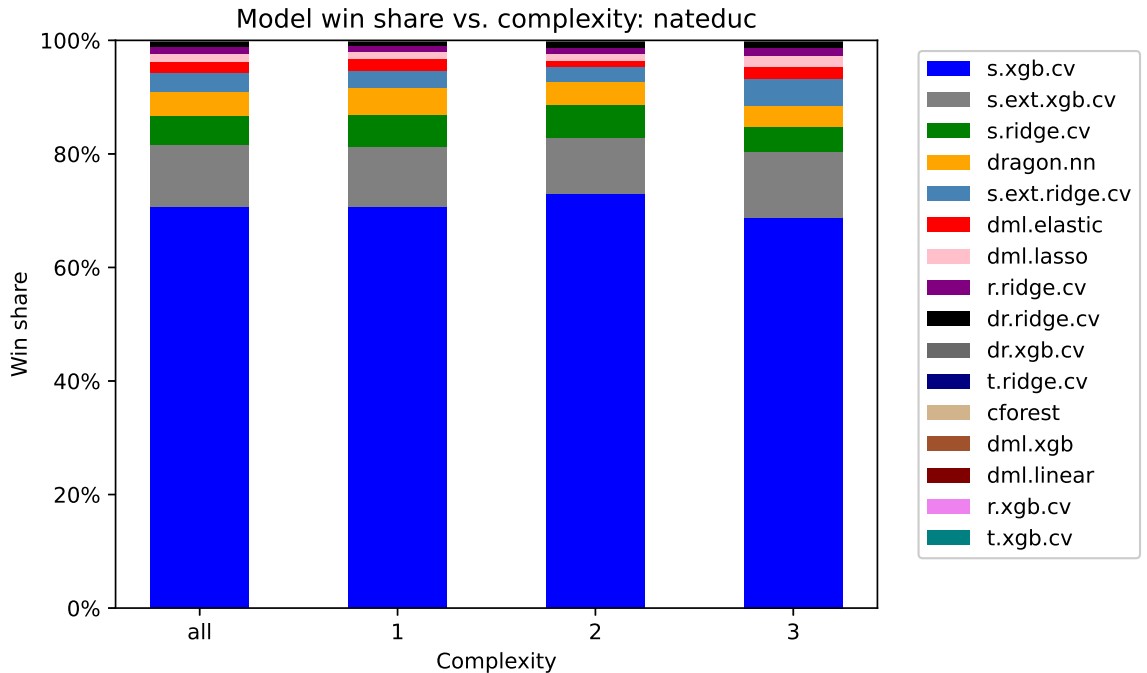

Figure 48: Win share vs. degenerate percentage, by model on `nateduc`

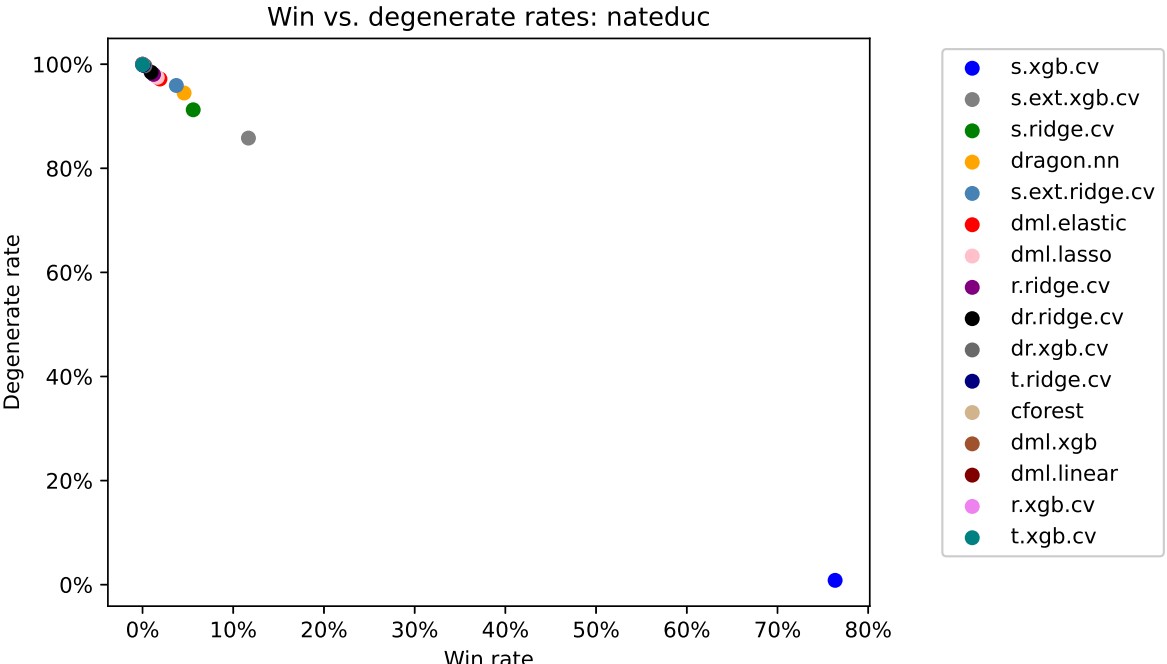

Figure 49: Model win share by estimation data, `nateduc`

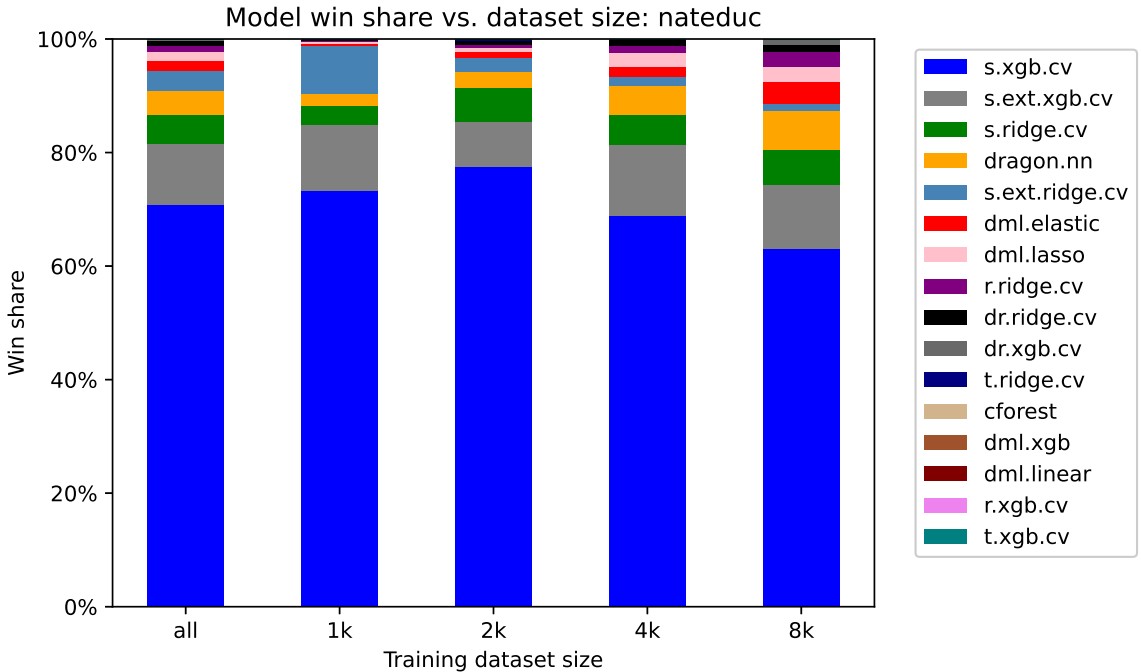

Figure 50: Model win share by treatment ratio, `nateduc`

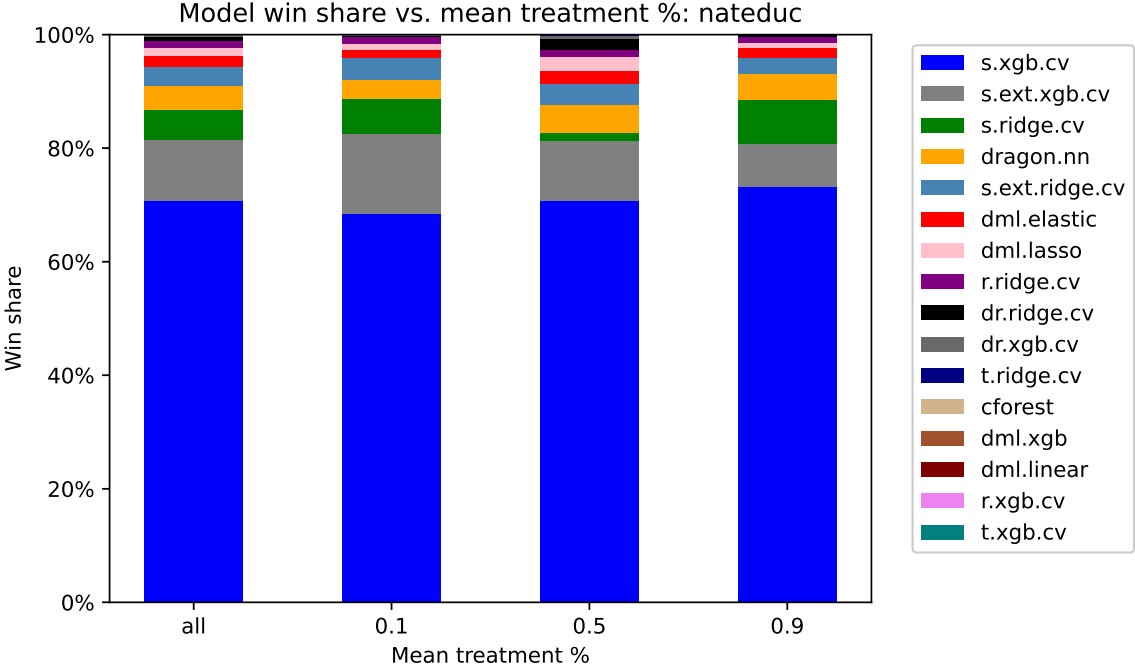

Figure 51: Model win share by assignment mechanism complexity, `nateduc`

### I.12 `NATENVIR`

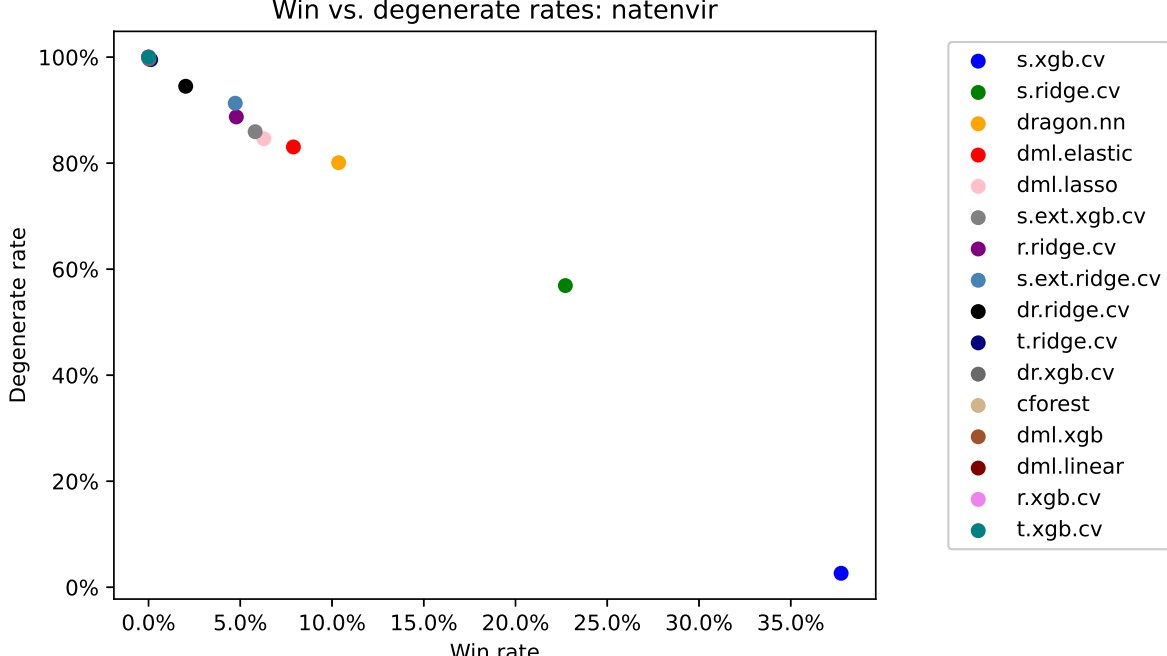

Figure 52: Win share vs. degenerate percentage, by model on `natenvir`

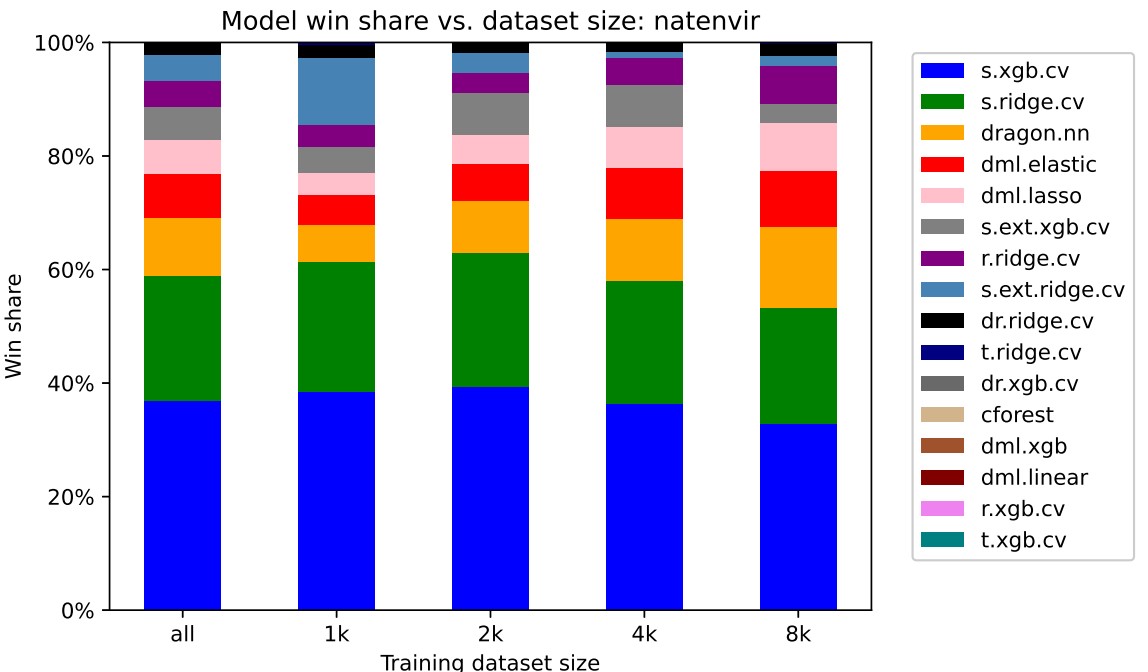

Figure 53: Model win share by estimation data, `natenvir`

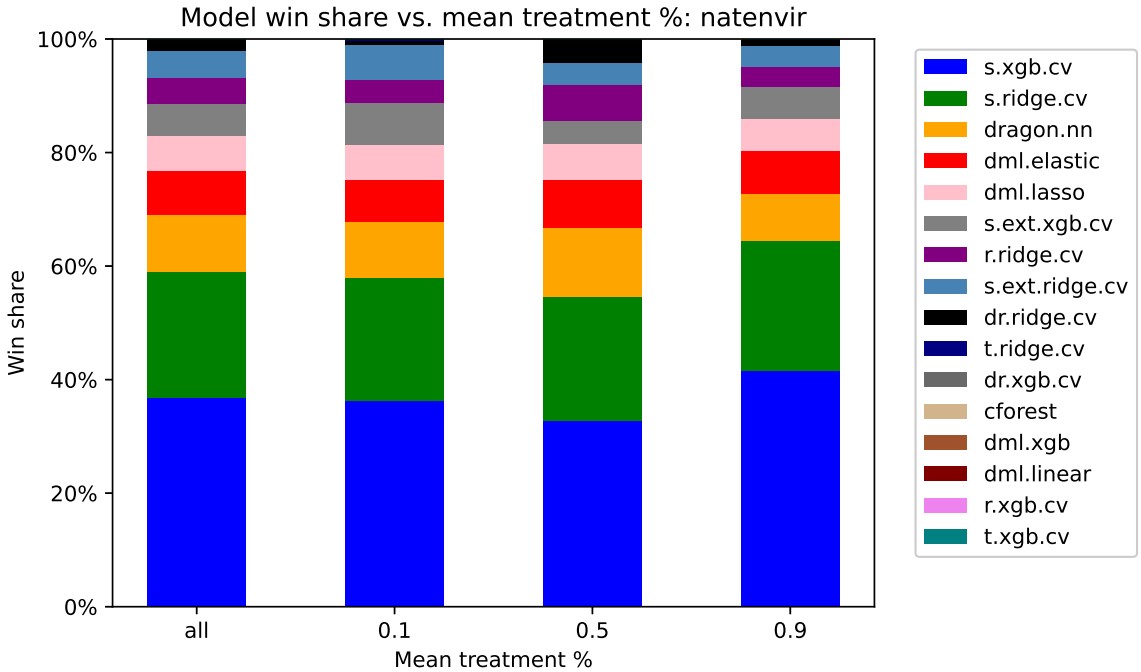

Figure 54: Model win share by treatment ratio, `natenvir`

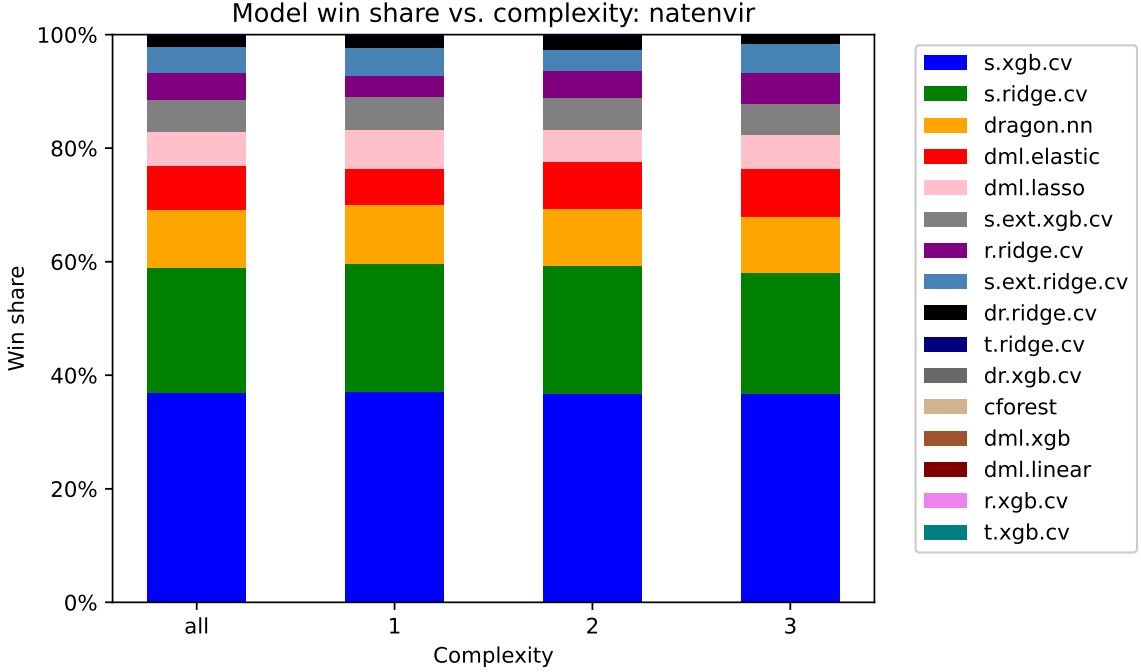

Figure 55: Model win share by assignment mechanism complexity, `natenvir`

## I.13  `NATFARE`

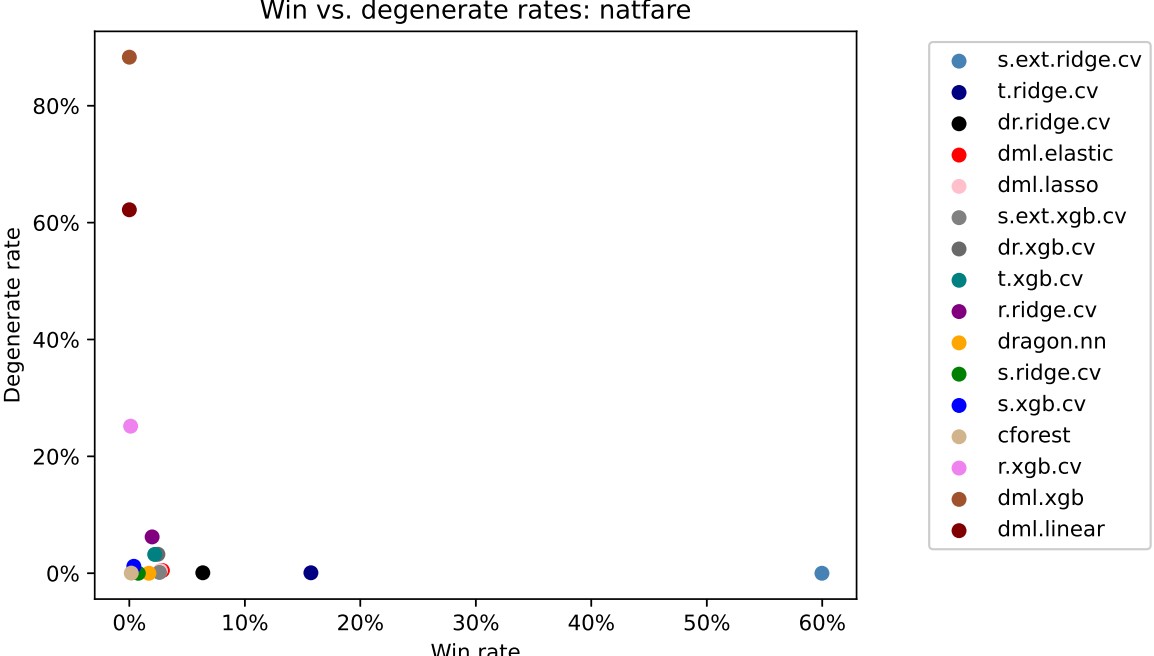

Figure 56: Win share vs. degenerate percentage, by model on `natfare`

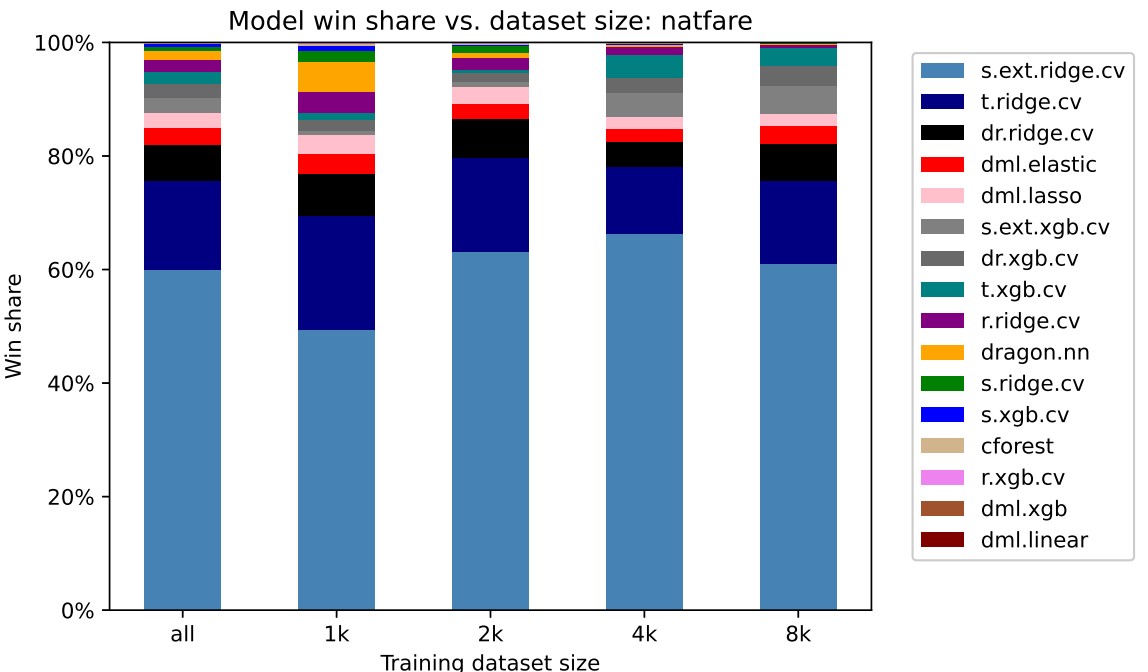

Figure 57: Model win share by estimation data, `natfare`

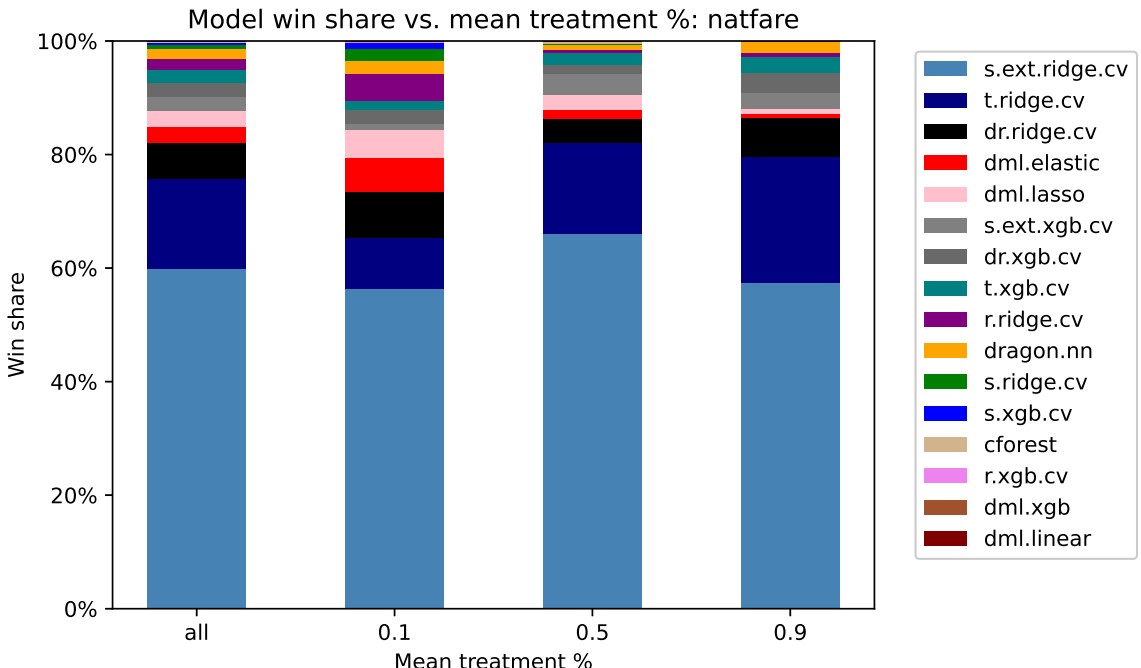

Figure 58: Model win share by treatment ratio, `natfare`

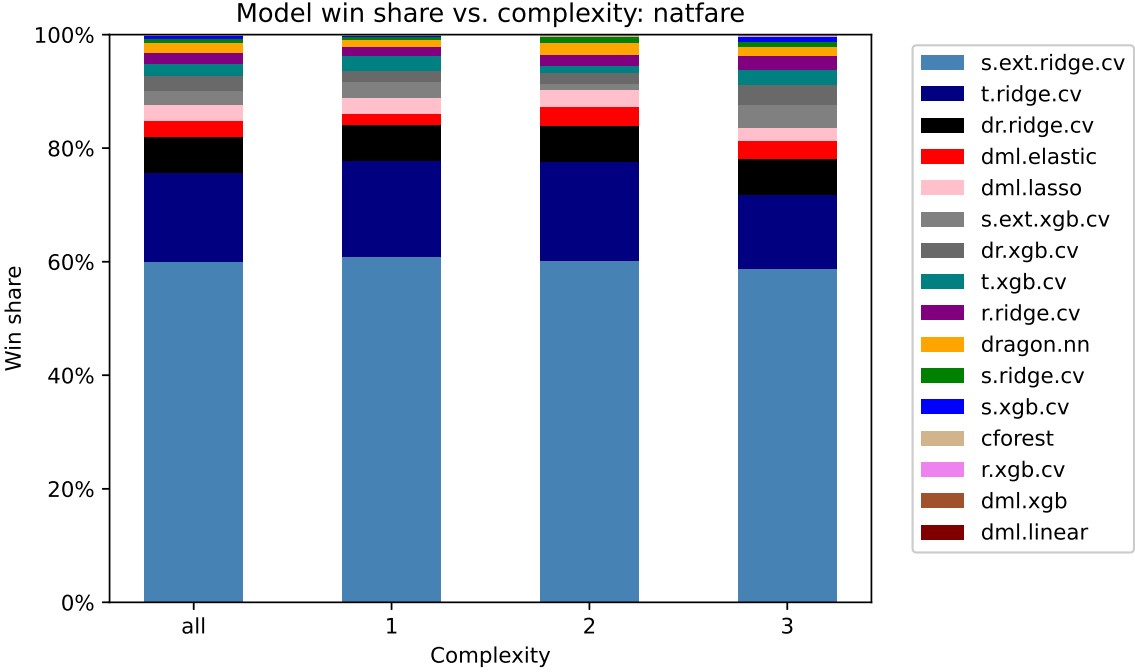

Figure 59: Model win share by assignment mechanism complexity, `natfare`

