# OpenReview forum: "Do Contemporary Causal Inference Models Capture Real-World Heterogeneity? Findings from a Large-Scale Benchmark"
_ICLR.cc/2025/Conference — ICLR 2025 Poster_

### Official Review · Reviewer_jZnM · 2024-10-31

**Soundness:** 3
**Presentation:** 2
**Contribution:** 3
**Rating:** 6
**Confidence:** 4

**Summary:**

This paper presents a framework to evaluate CATE estimation methods in observational studies. To address challenges in existing approaches that either leverage semi-synthetic data or rely on untestable modeling assumptions, this work proposes to rely on propensity modeling for evaluation. Estimators for the MSE (minus a constant) are proposed and their theoretical properties are discussed under conditions on the propensity models. Then, this paper provides a comprehensive benchmark by deriving observational studies via resampling RCTs, which leads to unbiased evaluation of existing CATE methods. The benchmark reveals surprising findings about the insufficiency of the existing methods for capturing heterogeneity in CATE.

**Strengths:**

1. This paper studies the important problem of reliable evaluation of CATE methods.
2. The paper is comprehensive, containing fruitful results.
3. The paper contributes new tools and benchmarks to the field, which may benefit other researchers and motivate future works.
4. The paper is clearly structured.

**Weaknesses:**

1. The writing quality of the paper can still be improved. Sometimes the advantages of the proposed method is a bit overclaimed. It will be helpful to also discuss the limitation of the proposed approach.
2. The discussion on the surprising results can be more in depth. Why are existing methods even worse than constant treatment effect estimation? Is this because they are too variable (so that variance in MSE is large)? Is this because the heterogeneity in the datasets is actually small?

Please see my questions for more comments.

**Questions:**

1. Why is Theorem 3.4 better than the double robustness results? It seems to require consistency of the propensity model, but double robustness estimation can be consistent if either the outcome model or the propensity model is correct.

2. What are the assumptions for Theorem 3.13 to hold? Does the propensity score need to be estimated? If so, why didn't its estimation error enter the result?

3. It's weird to count randomly split data as a "new dataset" so that there are 43200 benchmark datasets. Please consider revise your claim on the number of datasets.

4. In section 4.3, the performance of $\hat{Q}$ is a bit worse than model-augmented versions of $\hat{Q}$. It would be nice to provide some suggestions on what estimators to use in practice. Does the model-augmented version has additional bias/modeling concerns?

5. What's special about the Horwitz-Thompson estimator for $\eta$? It seems that $\eta$ can be anything whose conditional expectation given $X$ equals $\tau(X)$. Is it the point that it doesn't use any outcome model? Can it be replaced by other reasonable choices?

6. While addressing the concerns of semi-synthetic evaluation and outcome modeling, here the method requires the propensity model to be well estimated (though I understand that the evaluation part doesn't have this concern due to controlled sampling process). Is this over-simplifying real-world evaluation scenarios where the treatment assignment in observational studies can be complicated as well? What is the method evaluating if an inconsistent propensity model is used? Is it estimating some other quantity that is related to MSE of CATE estimator?

7. I would suggest separating the theoretical results for the most general evaluation case (eg in a real observational study) and the evaluation part in this paper. In real observational studies, consistent evaluation inevitably rely on some good modeling. However, it is important to be clearly stated that the evaluation in this paper doesn't need this because of the controlled sampling process.

---

> ### Author Response · Authors · 2024-11-20
> **rebuttal**
>
> W1. The writing quality of the paper can still be improved. Sometimes the advantages of the proposed method is a bit overclaimed. It will be helpful to also discuss the limitation of the proposed approach.
>
> Thank you for your feedback regarding the writing quality and the balance of claims in the paper. We will carefully review the manuscript to ensure that the advantages of our proposed method are presented accurately and without overstatement. Additionally, while we have discussed some limitations of our approach in Section 4.3 Result Validity and Considerations, we recognize the importance of a balanced perspective and will expand on this discussion in the revised manuscript to make these limitations clearer.
>
> Specifically, we have taken steps to minimize risks such as implementation errors by reusing codebases validated in prior large-scale benchmarks (e.g., Curth 2023) and relying on established CATE estimators and evaluation criteria when possible. While resource constraints limited the number of models included in our evaluation, we selected 16 widely used CATE models that span major strategies in the literature. Finally, while our datasets cover diverse real-world scenarios, we acknowledge that they may not generalize to every researcher's specific needs. These points, along with our findings, highlight both the strengths and limitations of our approach, and we will ensure these aspects are presented clearly in the revision.
>
> We appreciate your suggestion and will strive to further improve the clarity, balance, and overall quality of the writing in the next revision.
>
>
> W2. The discussion on the surprising results can be more in depth. Why are existing methods even worse than constant treatment effect estimation? Is this because they are too variable (so that variance in MSE is large)? Is this because the heterogeneity in the datasets is actually small?
>
> Response. We appreciate the reviewer’s thoughtful comment. While we do not yet have a comprehensive answer, we are actively analyzing the data. We already know that CATE estimators face two key challenges (and potentially more): (1) Unlike supervised learning, they never have access to ground truth labels in training, and (2) our observational sampling process introduces selection bias to emulate challenging real-world conditions, further complicating the problem. Additionally, we suspect that outcome variability and limited heterogeneity in some datasets contribute to these findings. This remains an area of active research, and we plan to explore these factors further in future work.
>
>
> Q1. Why is Theorem 3.4 better than the double robustness results? It seems to require consistency of the propensity model, but double robustness estimation can be consistent if either the outcome model or the propensity model is correct.
>
> Response. We thank the reviewer for this thoughtful comment. The reviewer is correct that results from double robustness (and orthogonality-based theory) are \textit{theoretically} stronger, as they ensure consistency if either the outcome model or the propensity model is correct. However, our \textit{empirical} results in Section 4.2 indicate that CATE estimates based on these assumptions often fall short of expectations in practice. This discrepancy raises important questions about whether other underlying assumptions (discussed in lines 67–70 of the paper) required by such results are indeed satisfied in real-world applications.
>
> When a theory demonstrates strong theoretical appeal but lacks empirical support, it highlights a gap that warrants further investigation. Until such investigations are thoroughly conducted, we propose Theorem 3.4 as a seemingly weaker alternative based on simpler and fewer assumptions for scenarios where verifying all the conditions for orthogonality is challenging.
>
> Q2. What are the assumptions for Theorem 3.13 to hold? Does the propensity score need to be estimated? If so, why didn't its estimation error enter the result?
>
> Response. Theorem 3.13 holds for RCT (used in observational sampling) and cases with known propensities, thus it does not need estimation of propensity. That's why there is no estimation error term in the result. It is grouped under Section 3.2 Improved Results For Observational Sampling. We will further clarify this part in next revision.

---

> ### Author Response · Authors · 2024-11-20
> **rebuttal**
>
> Q3. It's weird to count randomly split data as a "new dataset" so that there are 43200 benchmark datasets. Please consider revise your claim on the number of datasets.
>
> Response. Thank you for pointing out the need to clarify our claims regarding the number of datasets. We fully agree that it is important to distinguish between the 12 unique datasets and the 43,200 sampled variants. The latter are not "new datasets" in the conventional sense but rather represent variations created through controlled sampling strategies applied to the original 12 datasets. These variants were generated to rigorously test CATE models under diverse conditions, and their use is critical for evaluating performance across a wide range of settings.
>
> In the revised manuscript, we will explicitly clarify this distinction to avoid any misunderstanding and ensure that our claims are accurately represented.
>
> We also note that such sampling approaches are common in causal inference research, where the availability of diverse real-world datasets is often limited. By generating sampled variants, researchers can simulate different scenarios and stress-test models, which is a standard practice for benchmarking purposes.
>
> Q4. In section 4.3, the performance of is a bit worse than model-augmented versions of. It would be nice to provide some suggestions on what estimators to use in practice. Does the model-augmented version has additional bias/modeling concerns?
>
> Response: Which variant to use depends on theoretical and practical considerations: the original $\hat{Q}$ and $\hat{Q}(r_{LI})$ are model-free and easy to implement even without a scientist; if a large dataset is available, use them. Meanwhile, the variance reduction variants $\hat{Q}(r)$, including its special cases $\hat{Q}(r_R)$ and $\hat{Q}(r_{DR})$, offers the potential benefit of even lower variance, at the price of fitting and saving the extra plug-in estimators. The model-augmented version are also unbiased, but they do incur the cost of fitting models.
>
> Q5. What's special about the Horwitz-Thompson estimator for $\eta$? It seems that $\hat{eta}$ can be anything whose conditional expectation given equals $\eta$. Is it the point that it doesn't use any outcome model? Can it be replaced by other reasonable choices?
>
> Response. HT estimator can be replaced with other unbiased estimates to $\eta$. It is simple to use though (see previous comment)
>
> Q6. While addressing the concerns of semi-synthetic evaluation and outcome modeling, here the method requires the propensity model to be well estimated (though I understand that the evaluation part doesn't have this concern due to controlled sampling process). Is this over-simplifying real-world evaluation scenarios where the treatment assignment in observational studies can be complicated as well? What is the method evaluating if an inconsistent propensity model is used? Is it estimating some other quantity that is related to MSE of CATE estimator?
>
> Response. We agree that real-world scenarios with complex treatment assignments require further research. Until better alternatives are established, we recommend practitioners follow best practices from the literature, such as Chapter 15 of Applied Causal Inference Powered by ML and AI by Chernozhukov et al.
>
> In terms of future research, while the technical aspects of orthogonality-based models require further analysis, we believe the concept of orthogonality remains sound in principle, though refinements may be necessary. Additionally, the debiasing mechanism through sample splitting should mitigate some of the risks mentioned.
>
> Q7. I would suggest separating the theoretical results for the most general evaluation case (eg in a real observational study) and the evaluation part in this paper. In real observational studies, consistent evaluation inevitably rely on some good modeling. However, it is important to be clearly stated that the evaluation in this paper doesn't need this because of the controlled sampling process.
>
> Response. Thank you for this insightful suggestion. We fully agree that separating the theoretical results for the most general evaluation case (e.g., in a real observational study) from the specific evaluation conducted in this paper would add clarity. In particular, we acknowledge that consistent evaluation in real observational studies often relies on strong modeling assumptions, whereas the controlled sampling process in our work eliminates the need for such assumptions. We will make this distinction explicit in the revised manuscript.

---

> ### Author Response · Authors · 2024-11-25
>
> Dear Reviewer jZnM,
>
> Thanks again for your efforts in reviewing our paper. This is a gentle reminder that the author discussion deadline is approaching very soon (11/26). We'd like to know if our response has addressed your questions. We are happy to discuss more if there are any further questions.
>
> Thanks,
>
> Authors

---

> ### Comment · Reviewer_jZnM · 2024-11-25
>
> Thank you for the responses! I have some brief follow-up comments.
>
> For W2, I would appreciate seeing more concrete responses with your active analysis.
>
> For Q1, my comment was purely from a theoretical perspective. Theorem 3.4 is also a bit surprising to me because in many numerical experiments, the propensity score model is less robust to outcome regression. I didn't intend to mean that you need to switch to double robustness, but it will be helpful to emphasize using propensity model instead of both for general readers.
>
> For Q5, what if other $\hat\eta$ are used, like those with outcome models?

---

> > ### Author Response · Authors · 2024-11-26
> >
> > FOQ1. For Q1, my comment was purely from a theoretical perspective. Theorem 3.4 is also a bit surprising to me because in many numerical experiments, the propensity score model is less robust to outcome regression. I didn't intend to mean that you need to switch to double robustness, but it will be helpful to emphasize using propensity model instead of both for general readers.
> >
> > Response. We appreciate the comment and agree. We will emphasize the point here in the revised manuscript.
> >
> > FOQ2. For Q5, what if other $\eta$ are used, like those with outcome models?
> >
> > Response. Results from another $\hat{Q}_{R}$ show the overall degeneracy rate and model's relative performance (ranked by win share) stays same as Table 1.

---

> ### Author Response · Authors · 2024-11-26
> **follow up on W2**
>
> Here we dive deeper to find out when model tends to under-perform constant-effect baseline. In particular, we compute the % of cases each model out-performs the baseline constant effect model, and aggregate them by different dimensions (the global mean is 20%). Note that, this analysis focuses on 25,440 (out of 43,200) cases with non-degenerate constant effect baseline; it is a biased sample for all the datasets.
>
> At a high level, we find model performance vary by many dimensions (see details below). We don't have the complete diagnosis at this moment, but our current hypothesis is that beating a constant effect baseline is harder than we think -- when there is ground truth label. This is best seen when comparing CATE estimation to regression. In a regression model, we take the ground truth label for granted, and a constant effect model can be fit by simply fitting the intercept (in case for linear regression). But we don't have such ground truth for CATE estimation.
>
> Model performance relative to constant-effect baseline:
>
> We find the "percent better" metric vary by model, and model variations.
> | model          |   % better             |
> |:---------------|-----------------------:|
> | dragon.nn      |                 0.4292 |
> | dml.elastic    |                 0.3967 |
> | s.ridge.cv     |                 0.3701 |
> | s.ext.ridge.cv |                 0.3697 |
> | dr.ridge.cv    |                 0.3303 |
> | t.ridge.cv     |                 0.2715 |
> | s.xgb.cv       |                 0.2237 |
> | r.ridge.cv     |                 0.1949 |
> | dml.lasso      |                 0.1939 |
> | dr.xgb.cv      |                 0.1545 |
> | s.ext.xgb.cv   |                 0.1255 |
> | t.xgb.cv       |                 0.1053 |
> | cforest        |                 0.0401 |
> | r.xgb.cv       |                 0.0300 |
> | dml.linear     |                 0.0009 |
> | dml.xgb        |                 0.0000 |
>
>
> We find the "percent better" metric vary by dataset. More work needs to be done to quantify the level of heterogeneity.
> | dataset    |   % better             |
> |:-----------|-----------------------:|
> | natfare    |                 0.4183 |
> | natcrime   |                 0.3527 |
> | natcity    |                 0.1991 |
> | hillstrom  |                 0.1756 |
> | criteo     |                 0.1262 |
> | natdrug    |                 0.1195 |
> | nataid     |                 0.1106 |
> | natarms    |                 0.0729 |
> | sandercock |                 0.0699 |
> | natenvir   |                 0.0593 |
> | nateduc    |                 0.0351 |
> | ferman     |                 0.0161 |
>
>
> We find the "percent better" metric vary by estimation dataset size. This is consistent with our observation that more estimation data is helpful.
> |   estimation dataset size|   % better             |
> |-------------:|-----------------------:|
> |      0008000 |                 0.2455 |
> |      0004000 |                 0.2102 |
> |      0002000 |                 0.1777 |
> |      0001000 |                 0.1589 |
>
>
> We find the model performs better when the estimation dataset is balanced.
> |   treatment % |   % better             |
> |--------:|-----------------------:|
> |      50 |                 0.2360 |
> |      10 |                 0.1844 |
> |      90 |                 0.1804 |
>
>
> We didn't find major fluctuation in the "percent better" metric when the assignment mechanism becomes more nonlinear (complex).
> |   complexity|   % better             |
> |--------:|-----------------------:|
> |       1 |                 0.2049 |
> |       3 |                 0.2013 |
> |       2 |                 0.2006 |

---

### Official Review · Reviewer_UrBr · 2024-11-02

**Soundness:** 3
**Presentation:** 3
**Contribution:** 3
**Rating:** 8
**Confidence:** 4

**Summary:**

This paper conducts an empirical study of the accuracy of CATE estimation methods, asking whether existing methods in the literature reliably outperform trivial benchmarks. The main idea is to use RCT datasets, where the propensity is known, to construct benchmarks where a simulated propensity score is used to construct an "observational" sample. Since the propensity score is known for the original data, estimators on the observational sample can be benchmarked without having to rely on accurate estimates of propensity or outcome models (effectively using a HT estimator, potentially augmented with other regression models just for variance reduction purposes).

**Strengths:**

This work is definitely important for the field. Evaluating CATE models is very difficult due to the risk of self-serving bias that the paper discusses, and it is unclear when in practice modern techniques are helpful. Constructing benchmarks and evaluation methods which use only real data, via RCT datasets, is important. Using semi-synthetic datasets with simulated outcomes, as is common in the field, is much less convincing.

**Weaknesses:**

There are some respects in which the execution of the paper could be improved:

(1) Based on plots in the appendix, there appears to be significant variation in the performance of different methods across datasets (as would be expected). This deserves more discussion, and investigation. E.g. why do double ML methods perform well in some cases and not in many others? One hypothesis, based on benchmarks for ATE estimation, would be that outcome regression methods work best as long as the outcome models are reasonably well estimated, with double-ML style methods providing benefits when the outcomes are not as well captured.

As is, there is little in the way of takeaways about the conditions under which existing methods do or don't perform well. This sort of diagnosis would be at least as useful as the headline results about % of cases where methods work well or don't, since the headline numbers are very specific to the composition of this specific benchmark.

(2) Relatedly, the benchmark appears to be weighted quite heavily towards a set of multiple tasks all drawn from the same dataset: out of 12 tasks, 8 are different outcomes taken from the general social survey. We might reasonably expect different tasks from the same dataset to share some similar characteristics so the amount of actual diversity in the benchmark is less than what the number of tasks would suggest.

(3) Regarding the degeneracy rate, are the results different if we test for whether Q is significantly larger than 0 (i.e., a CI excludes 0) rather than if the point estimate is > 0? I would guess that most of the datasets are large enough for the estimates of Q to be fairly precise, but it would be helpful to verify this.

While I think the paper makes a worthwhile contribution even with these weaknesses, the contribution would be significantly strengthened with a more diverse set of tasks and more analysis of what drives performance variation across those tasks.

**Questions:**

Any comments/clarifications, particularly related to points 1 and 3 above, would be helpful.

---

> ### Author Response · Authors · 2024-11-20
> **rebuttal**
>
> W1. Based on plots in the appendix, there appears to be significant variation in the performance of different methods across datasets (as would be expected). This deserves more discussion, and investigation. E.g. why do double ML methods perform well in some cases and not in many others? One hypothesis, based on benchmarks for ATE estimation, would be that outcome regression methods work best as long as the outcome models are reasonably well estimated, with double-ML style methods providing benefits when the outcomes are not as well captured. As is, there is little in the way of takeaways about the conditions under which existing methods do or don't perform well. This sort of diagnosis would be at least as useful as the headline results about \% of cases where methods work well or don't, since the headline numbers are very specific to the composition of this specific benchmark.
>
> Response. We appreciate the reviewer’s insightful comment highlighting the importance of diagnosing the variation in method performance across datasets. We agree that understanding the conditions under which different methods succeed or fail would be highly valuable and complementary to the primary contribution of this paper, which focuses on identifying these performance differences. While we are actively investigating these patterns, a comprehensive diagnosis would require a more detailed analysis, which we aim to address in future work.
>
>
> Our current hypothesis is that performance variation arises from a combination of factors, including the domain-specific data-generating process (e.g., type of response, effect size, and heterogeneity), data characteristics (e.g., sample size and feature dimensionality), and modeling choices (e.g., assumptions and base learners).
>
> For Double ML methods specifically, we suspect that their performance could be influenced by the same list of factors. While the sample-splitting and debiasing mechanisms should, in theory, mitigate risks from poorly specified outcome models, other practical challenges—such as violations of assumptions required for orthogonality conditions to hold—may play a role. However, we do not yet have a definitive explanation and believe further theoretical and empirical investigations are needed to reconcile these findings with existing theory.
>
> We will incorporate these points into the revised manuscript and look forward to contributing to this important line of inquiry in future research.
>
> W2. Relatedly, the benchmark appears to be weighted quite heavily towards a set of multiple tasks all drawn from the same dataset: out of 12 tasks, 8 are different outcomes taken from the general social survey. We might reasonably expect different tasks from the same dataset to share some similar characteristics so the amount of actual diversity in the benchmark is less than what the number of tasks would suggest.
>
> Response. We appreciate the reviewer’s observation regarding the potential lack of diversity due to multiple tasks being drawn from the same dataset. As discussed in our response to Reviewer vxwd, these datasets represent all that met our data selection criteria.
>
> The 8 tasks from the General Social Survey do share the same set of covariates. However, to assess diversity among these tasks, we measured the Pearson correlation among their outcome variables and found the correlations ranged from -0.05 to 0.25, with a mean of 0.09, indicating weak relationships among the outcome distributions. Additionally, these tasks yielded 6 different winner models, suggesting the tasks have capabilities to detect model performance variation.
>
> We hope these points help clarify the level of diversity within the benchmark, despite the overlap in covariates.
>
> W3. Regarding the degeneracy rate, are the results different if we test for whether Q is significantly larger than 0 (i.e., a CI excludes 0) rather than if the point estimate is greater than 0? I would guess that most of the datasets are large enough for the estimates of Q to be fairly precise, but it would be helpful to verify this.
>
> Response. The reviewer's intuition is right: 94\% of the degenerate model $\hat{Q}$ have p-value (from t-test) less than 0.05.

---

> ### Author Response · Authors · 2024-11-25
>
> Dear Reviewer UrBr,
>
> Thanks again for your efforts in reviewing our paper. This is a gentle reminder that the author discussion deadline is approaching very soon (11/26). We'd like to know if our response has addressed your questions. We are happy to discuss more if there are any further questions.
>
> Thanks,
>
> Authors

---

### Official Review · Reviewer_vxwd · 2024-11-03

**Soundness:** 2
**Presentation:** 2
**Contribution:** 2
**Rating:** 6
**Confidence:** 2

**Summary:**

This paper presents a large-scale benchmark study evaluating the performance of contemporary Conditional Average Treatment Effect (CATE) estimation models. The authors use a novel application of observational sampling to evaluate 16 modern CATE models across 43,200 datasets.  Their key findings challenge the effectiveness of current CATE models in capturing real-world heterogeneity.

**Strengths:**

Large-scale and comprehensive benchmark study.

Novel approach to CATE evaluation using observational sampling and the Q statistic.

Rigorous theoretical analysis and proofs supporting the proposed methodology.

Use of real-world datasets provides valuable insights into the limitations of current CATE models.

**Weaknesses:**

While aiming for diversity, the selection of datasets may still not fully represent the breadth of real-world applications, potentially limiting the generalizability of the findings. More explanation of dataset selection criteria would strengthen the paper.

The paper focuses on MSE as the primary evaluation metric. While justified by its practical relevance, exploring alternative evaluation metrics could provide additional insights.

While innovative, the reliance on the Q statistic is new and requires further validation and adoption by the wider research community.

**Questions:**

The paper discusses the generalization of the Q statistic to new distributions. Could you provide more details about the assumptions underlying these generalization results? How sensitive are these results to violations of these assumptions?

What are the main limitations of the current study, and what directions for future research do you suggest based on these findings? What kinds of new CATE models or evaluation methods should be explored? What types of additional data would be useful for further validation?

How can the findings of this study be used to guide the practical application of CATE models in various domains (medicine, economics, etc.)? What advice would you give to practitioners regarding the selection and interpretation of CATE models?

---

> ### Author Response · Authors · 2024-11-20
> **rebuttal**
>
> W1. While aiming for diversity, the selection of datasets may still not fully represent the breadth of real-world applications, potentially limiting the generalizability of the findings. More explanation of dataset selection criteria would strengthen the paper.
>
> Response. We appreciate the reviewer’s comment regarding the potential limitations in the generalizability of our findings due to the dataset selection. We fully agree that evaluating CATE algorithms comprehensively requires a diverse range of datasets. However, our dataset selection was constrained by the following criteria: (1) RCT datasets with over 18,000 samples, (2) availability of covariates, (3) real-world outcomes (not simulated), and (4) public accessibility. Based on these criteria, we searched multiple RCT data registries (e.g., Harvard DataVerse, AEA registry, and ICTRP) and screened thousands of datasets. The 12 datasets represent all that met our requirements.
>
> To address generalization concerns, we also provide theoretical results in Theorems 3.8 and 3.9, which offer insights into the applicability of our findings beyond the datasets used. While we acknowledge this limitation, we believe our approach represents a meaningful step forward, given the constraints of current data availability.
>
> W2. The paper focuses on MSE as the primary evaluation metric. While justified by its practical relevance, exploring alternative evaluation metrics could provide additional insights.
>
> Response. We appreciate the reviewer’s suggestion to explore alternative evaluation metrics. We agree that additional metrics, such as proxy losses or rank-based metrics, could provide complementary insights. These alternatives are discussed in the "Related Work" section. However, our study focuses on MSE (commonly referred to as PEHE in the causal inference literature) due to its practical relevance in evaluating the accuracy of CATE estimates. Accuracy is foundational to the utility of CATE estimators, and MSE remains the widely accepted criterion for this purpose. Without a reliable measure of accuracy, the usefulness of CATE estimates becomes uncertain.
>
> We also note that the agreement between alternative score functions and MSE is not yet well established, as highlighted in prior work (Curth and van der Schaar, 2021; 2023; Neal et al., 2021; Mahajan et al., 2023) and in our discussion in Section 4.3.
>
> We are encouraged that $\hat{Q}$ effectively reveals the limitations of current CATE algorithms, underscoring its practical value. However, we acknowledge that no single metric is perfect, and we plan to explore alternative metrics in future work to provide a more comprehensive evaluation framework.
>
> W3. While innovative, the reliance on the Q statistic is new and requires further validation and adoption by the wider research community.
>
> Response. We agree with the reviewer that any new statistic, including $\hat{Q}$, requires further validation and scrutiny by the research community. We view this as a natural and necessary step for the adoption of any metric, rather than a weakness. In the paper, we have addressed potential limitations of $\hat{Q}$ in Section 4.3. We also acknowledge that for
> the metric  to reach its full potential as a practical and reliable evaluation metric, it will require collaborative efforts from the research community to apply it to a broader and more diverse set of datasets.

---

> ### Author Response · Authors · 2024-11-20
> **rebuttal**
>
> Q1. The paper discusses the generalization of the Q statistic to new distributions. Could you provide more details about the assumptions underlying these generalization results? How sensitive are these results to violations of these assumptions?
>
> Response. For Theorem 3.8 (IPW), the key assumptions are:
>
> 1. The two data distributions $\Pi_1$ and $\Pi_2$ share the same conditional outcome distribution $\tau(x)$. Violating this assumption means we are studying two different (potentialy unrelated) conditional outcome distributions, and results from one may not translate to the other.
>
> 2. Their marginal covariate distribution $\mathcal{\chi}_1$ and $\mathcal{\chi}_1$ share a common support. Violating this assumption means we need to extrapolate beyond the observed data, which can lead to invalid conclusions.
>
> 3. The density ratio $\zeta(x)$ is known; violating this assumption likely means result is no longer unbiased.
>
> For Theorem 3.9 (Ranking), the key assumption is that the difference between $Q$ values for two data distribution $\Pi_1$ and $\Pi_2$ is large enough to overcome their distribution difference, so that a better model on $\Pi_1$ remains better on $\Pi_2$. If the two distributions are far apart, the ranking result can change.
>
> Q2. What are the main limitations of the current study, and what directions for future research do you suggest based on these findings? What kinds of new CATE models or evaluation methods should be explored? What types of additional data would be useful for further validation?
>
> Response.We appreciate the reviewer’s thoughtful questions regarding the limitations of the current study and potential future research directions. See Section 4.3 Result Validity and Considerations for more discussions.
>
> Regarding the limitations of our study, we recognize the need for more diverse real-world datasets, a broader range of tested models, and evaluations under a wider variety of experimental settings. On the dataset front, we are actively working to collect additional real-world datasets, particularly from underrepresented domains such as pharmaceuticals, where personalized treatment decisions could benefit significantly from robust CATE estimation. Expanding the dataset diversity will help validate our findings across a broader spectrum of applications.
>
> In terms of models, we aim to include a wider array of methodologies, such as deep learning and Gaussian process-based CATE estimation models, in future iterations of this work. Additionally, we plan to investigate model performance driven by other factors, such as structural assumptions, regularization techniques, and hyper-parameter choices, to provide a more nuanced understanding of CATE model behaviors.
>
> On the topic of model evaluation, we believe that establishing a more comprehensive and standardized benchmark for CATE estimation is a critical next step. Similar to the role of ImageNet in driving progress in computer vision, a well-curated CATE benchmark could facilitate systematic comparison, foster innovation, and guide researchers toward addressing key challenges in the field. We hope this work contributes to laying the foundation for such a benchmark and inspires collaborative efforts toward its realization.
>
> Q3. How can the findings of this study be used to guide the practical application of CATE models in various domains (medicine, economics, etc.)? What advice would you give to practitioners regarding the selection and interpretation of CATE models?
>
> Response. Thank you for raising this important question about the practical implications of our findings. We believe the results of our study offer several key takeaways for guiding the application of CATE models; meanwhile we caveat that our result is the starting point for more research in this direction.
>
> First, we encourage practitioners to interpret CATE estimates with caution. Our results show that contemporary CATE models can struggle in certain settings or when evaluated against simple baselines. Practitioners should consider running multiple model, carefully fine-tuning base learners, comparing results against ATE estimates, and applying domain knowledge to gain confidence their findings, before taking actions.
>
> Second, when taking actions backed by causal inference insights, practitioners should account for the risk of using CATE estimates with high MSE. For their problem on hand, the amount of risk associated with CATE may be very low (e.g., when deciding who should receive a marketing email) to high (e.g., when deciding who should receive a new research treatment).
>
> Third, we see this work as part of a broader call to action for for experimentation, and experimentation data collection and sharing. Improved data sharing and the creation of comprehensive benchmarks will be critical to advancing the practical utility of CATE models. We hope our study serves as a starting point for these efforts and encourages practitioners to actively engage in this effort.

---

> ### Author Response · Authors · 2024-11-25
>
> Dear Reviewer vxwd,
>
> Thanks again for your efforts in reviewing our paper. This is a gentle reminder that the author discussion deadline is approaching very soon (11/26). We'd like to know if our response has addressed your questions. We are happy to discuss more if there are any further questions.
>
> Thanks,
>
> Authors

---

### Official Review · Reviewer_x6Rc · 2024-11-04

**Soundness:** 3
**Presentation:** 2
**Contribution:** 2
**Rating:** 5
**Confidence:** 3

**Summary:**

In this work, the authors propose a new way to estimate treatment effect and perform model selection when the counterfactual is not available. To do this, the authors propose a new metric called Q, which is derived from the Mean Squared Error (MSE) but adjusted to exclude terms that do not depend on the estimator.

**Strengths:**

Lacking good metrics for evaluation is a big problem in casual inference due to no access to counterfactuals. This paper proposes a new metric, which could be helpful for practitioners when comparing different models.

**Weaknesses:**

1. For the experiments given in this work, the estimation (training) size is much smaller than the evaluation size. This is not intuitive to me as in typical ML settings, we have training-test ratio to be 4:1. Using a much lower ratio of training vs test samples could potentially lead to the problem of unfitting, which could be a reason why many baselines have too many degenerate cases. I think it is necessary to use a large number to training samples to rule out this hypothetical scenario.

**Questions:**

1. Can you explain why your estimation size is much smaller than the evaluation size?

2. Can you re-conduct the experiments when the estimation size is much larger than the evaluation size? Let's use the more standard ML experimental setup, where the ration between training and test samples is 4:1. You should apply this to all datasets in your Table 4. Then rerun the whole experimental pipeline and report results similar to those shown in Table 1.

3. Also, there is no standard deviation for Table 1. It might be a little bit misleading in using the description "43,200 datasets" in Table 1. What you are essentially doing is repeated resampling, similar to bootstrap. In this case, in addition to what I ask for #2 above, could you also report results on the same datasets, with same percentage of treatment and other parameters fixed, with both mean and standard deviation, where the standard deviation solely comes from bootstrap?

---

> ### Author Response · Authors · 2024-11-20
> **rebuttal**
>
> W1. For the experiments given in this work, the estimation (training) size is much smaller than the evaluation size. This is not intuitive to me as in typical ML settings, we have training-test ratio to be 4:1. Using a much lower ratio of training vs test samples could potentially lead to the problem of unfitting, which could be a reason why many baselines have too many degenerate cases. I think it is necessary to use a large number to training samples to rule out this hypothetical scenario.
>
> Response. Thank you for highlighting the difference between the typical supervised learning paradigm and the causal inference paradigm, particularly with regard to training-test size ratios. We agree that a much lower ratio of training to test samples could potentially lead to underfitting. We are working on testing the 4:1 ratio and post results when ready.
>
> However, our design reflects the unique challenges and goals of our study, which differ from traditional machine learning frameworks. Specifically, our estimation-evaluation framework is distinct in two key ways:
>
> Estimation Stage: this stage is designed to mimic realistic constraints faced by CATE estimation, such as limited training data availability and selection bias. For example, in fields like medical research, obtaining datasets larger than a few thousand samples is often infeasible due to cost, privacy, and regulatory constraints. Practitioners frequently rely on such limited datasets to make critical decisions. Small estimation dataset sizes are common in causal inference research. For instance, the IHDP dataset has 747 data points, the ACIC2016 dataset has 4,802, and the JOBS dataset has 3,212.
>
> Evaluation Stage: this stage focuses on assessing whether CATE models perform well. Without access to ground truth labels, it is fundamentally different from evaluation (using test dataset with labels) in supervised learning. As a result, we rely on large evaluation datasets to accurately evaluate CATE models' ability to recover real-world heterogeneity, compensating for the lack of ground truth labels.
>
> In short, our design aims to mimic challenges faced by CATE estimation models in real-world settings, and maximize the chance to accurately evaluate model performance (MSE) to compensate for lack of ground truth data.
>
> Q1. Can you explain why your estimation size is much smaller than the evaluation size? \\
>
> Response. see response to previous point.
>
> Q2. Can you re-conduct the experiments when the estimation size is much larger than the evaluation size? Let's use the more standard ML experimental setup, where the ration between training and test samples is 4:1. You should apply this to all datasets in your Table 4. Then rerun the whole experimental pipeline and report results similar to those shown in Table 1.
>
> Response. We will test a 4:1 ratio and report results as soon as they are available. However, re-running the entire experimental pipeline across all datasets in Table 4 is a time-intensive process. While we will prioritize this analysis, it may not be feasible to complete it before rebuttal ends. Regardless, we are committed to adding these results to the next revision of this paper to ensure a thorough exploration of the proposed setup.
>
> Q3. Also, there is no standard deviation for Table 1. It might be a little bit misleading in using the description 43,200 datasets in Table 1. What you are essentially doing is repeated resampling, similar to bootstrap. In this case, in addition to what I ask for W2 above, could you also report results on the same datasets, with same percentage of treatment and other parameters fixed, with both mean and standard deviation, where the standard deviation solely comes from bootstrap?
>
> Response. Just to clarify, for this particular request are you looking for results from the 4:1 data split, or the results from the incumbent data split? We will update the results as requested.

---

> ### Author Response · Authors · 2024-11-21
> **supplemental info on estimation dataset size**
>
> If the reviewer wants to better understand whether more estimation data improves model performance, more granular data from the experiment discussed in the paper may be helpful:
>
> We find the degeneracy rate decreases with the amount of estimation data: 67.3\% (1k estimation data), 64.1\% (2k), 60.7\% (4k), and 57\% (8k).
>
> One may find this trend comforting as degeneracy risk may continue to decline when estimation datasets grow larger. But if the roughly log-linear trend indeed continues, it will take much more data to reduce the degeneracy rate to a more useful level (e.g., 5 or 10\%). Furthermore, we note that degeneracy is a very low bar for evaluating the quality of CATE models. For CATE estimates to be practically useful in real-world settings, degeneracy rates need to be significantly lower than those reported in the current paper.

---

> ### Author Response · Authors · 2024-11-25
>
> Dear Reviewer x6Rc,
>
> Thanks again for your efforts in reviewing our paper. This is a gentle reminder that the author discussion deadline is approaching very soon (11/26). We'd like to know if our response has addressed your questions. We are happy to discuss more if there are any further questions.
>
> Thanks,
>
> Authors

---

> > ### Comment · Reviewer_x6Rc · 2024-11-26
> >
> > Thank you very much for the previous responses. With regards to the 4:1 train vs test data split experiment I requested, please just post what you have got so far before the rebuttal deadline. Reporting even unfinished experiments shows more sincerity in addressing this concern than promising to report the results and end up not doing it for the final version (if the paper does get accepted at the end).
> >
> > This is a premature comment right now (since I have not seen the 4:1 data split experiments), but I am not fully buying the answer in your R1. If you do not have the correct train vs test splits, I am just not sure how much takeaway I can get from some of the conclusions in the paper. The issue I am seeing is not so much as how large/small the entire dataset is, but the percentages of train vs test data are. If you do have a large/huge dataset (they do exist for real-world application today; for example, the healthcare real-world datasets MIMIC III and MIMIC IV datasets are huge; you can for sure do causal inference on this kind of large scale), which I believe you do have for some of them in your experiments , it doesn't make sense to train on only a small portion of it. An analogy would be restricting ourselves to train neural networks on only a small number of samples and test on a large number of samples and conclude in the end that SVM performs much better than neural networks. If this were the standard experimental setup, we would have never started the deep learning revolution in the ML community.
> >
> > With regards to reporting the standard deviations, you can report it in either way you want at this point, given the time constraint. However, I think the correct way is to do a 5-split cross validation and report the mean and standard deviation based on the 5-split results.
> >
> > At any rate, please post whatever you have for the 4:1 train vs test split results I requested during my initial evaluation, before the rebuttal period ends. This is the most important result I need to see in order to make a correct assessment of this work.

---

> ### Author Response · Authors · 2024-11-26
> **results from 4:1 ratio**
>
> Here we show 4:1 estimation/evaluation ratio with the same estimation dataset size (1k,2k,4k,and 8k). see result below. As an example, when estimation dataset is 8k, the evaluation dataset has 8k x 0.25 = 2k samples. Due to time constraint, we only have results from criteo dataset.
>
> The Standard Error (SE) and 95% Confidence Interval (CI) comes from 10k bootstrap.
>
> | Model          | Win Rate Mean (SE)   | Win Rate 95% CI    | Degeneracy Rate Mean (SE)   | Degeneracy Rate 95% CI   | Avg Rank Mean (SE)   | Avg Rank 95% CI   |
> |:---------------|:---------------------|:---------------|:----------------------------|:---------------------|:---------------------|:-------------------|
> | cforest        | 0.122 ( 0.006 )      | (0.111, 0.133) | 0.644 ( 0.008 )             | (0.628, 0.660)       | 7.8 ( 0.1 )          | (7.7, 7.9)         |
> | dml.elastic    | 0.052 ( 0.004 )      | (0.045, 0.060) | 0.630 ( 0.008 )             | (0.613, 0.646)       | 7.9 ( 0.1 )          | (7.8, 8.1)         |
> | dml.lasso      | 0.061 ( 0.004 )      | (0.053, 0.070) | 0.623 ( 0.008 )             | (0.607, 0.639)       | 7.9 ( 0.1 )          | (7.8, 8.0)         |
> | dml.linear     | 0.011 ( 0.002 )      | (0.007, 0.014) | 0.973 ( 0.003 )             | (0.968, 0.979)       | 15.0 ( 0.0 )         | (14.9, 15.0)       |
> | dml.xgb        | 0.013 ( 0.002 )      | (0.010, 0.018) | 0.977 ( 0.003 )             | (0.972, 0.982)       | 15.3 ( 0.0 )         | (15.2, 15.4)       |
> | dr.ridge.cv    | 0.039 ( 0.003 )      | (0.033, 0.046) | 0.630 ( 0.008 )             | (0.614, 0.646)       | 7.6 ( 0.1 )          | (7.4, 7.7)         |
> | dr.xgb.cv      | 0.058 ( 0.004 )      | (0.050, 0.066) | 0.703 ( 0.008 )             | (0.688, 0.718)       | 8.6 ( 0.1 )          | (8.5, 8.8)         |
> | dragon.nn      | 0.068 ( 0.004 )      | (0.059, 0.076) | 0.522 ( 0.009 )             | (0.505, 0.539)       | 6.3 ( 0.1 )          | (6.2, 6.4)         |
> | r.ridge.cv     | 0.033 ( 0.003 )      | (0.027, 0.039) | 0.534 ( 0.009 )             | (0.517, 0.551)       | 6.5 ( 0.1 )          | (6.4, 6.6)         |
> | r.xgb.cv       | 0.088 ( 0.005 )      | (0.079, 0.098) | 0.738 ( 0.007 )             | (0.723, 0.752)       | 9.4 ( 0.1 )          | (9.2, 9.5)         |
> | s.ext.ridge.cv | 0.077 ( 0.005 )      | (0.068, 0.086) | 0.569 ( 0.008 )             | (0.552, 0.586)       | 6.5 ( 0.1 )          | (6.4, 6.6)         |
> | s.ext.xgb.cv   | 0.116 ( 0.005 )      | (0.105, 0.126) | 0.527 ( 0.009 )             | (0.511, 0.544)       | 6.6 ( 0.1 )          | (6.4, 6.7)         |
> | s.ridge.cv     | 0.035 ( 0.003 )      | (0.029, 0.041) | 0.479 ( 0.009 )             | (0.462, 0.496)       | 5.8 ( 0.1 )          | (5.7, 5.9)         |
> | s.xgb.cv       | 0.108 ( 0.005 )      | (0.097, 0.118) | 0.173 ( 0.006 )             | (0.161, 0.186)       | 5.8 ( 0.1 )          | (5.6, 5.9)         |
> | t.ridge.cv     | 0.071 ( 0.004 )      | (0.063, 0.080) | 0.713 ( 0.008 )             | (0.698, 0.727)       | 9.0 ( 0.1 )          | (8.8, 9.1)         |
> | t.xgb.cv       | 0.067 ( 0.004 )      | (0.059, 0.076) | 0.760 ( 0.007 )             | (0.746, 0.774)       | 10.0 ( 0.1 )         | (9.9, 10.2)        |

---

> ### Author Response · Authors · 2024-11-26
>
> In comparison, we also share the subsection of Table 1 on criteo, as below. Note that in this original setting, the evaluation dataset is much larger than the 4:1 ratio.
>
> | Model          | Win Rate Mean (SE)   | Win Rate 95% CI    | Degeneracy Rate Mean (SE)   | Degeneracy Rate 95% CI   | Avg Rank Mean (SE)   | Avg Rank 95% CI   |
> |:---------------|:---------------------|:---------------|:----------------------------|:---------------------|:---------------------|:-------------------|
> | cforest        | 0.038 ( 0.003 )      | (0.031, 0.044) | 0.904 ( 0.005 )             | (0.893, 0.914)       | 8.5 ( 0.1 )          | (8.4, 8.6)         |
> | dml.elastic    | 0.065 ( 0.004 )      | (0.057, 0.073) | 0.728 ( 0.008 )             | (0.713, 0.743)       | 7.7 ( 0.1 )          | (7.5, 7.8)         |
> | dml.lasso      | 0.056 ( 0.004 )      | (0.048, 0.064) | 0.727 ( 0.008 )             | (0.712, 0.742)       | 7.8 ( 0.1 )          | (7.6, 7.9)         |
> | dml.linear     | 0.000 ( 0.000 )      | (0.000, 0.000) | 1.000 ( 0.000 )             | (1.000, 1.000)       | 15.3 ( 0.0 )         | (15.3, 15.3)       |
> | dml.xgb        | 0.000 ( 0.000 )      | (0.000, 0.000) | 1.000 ( 0.000 )             | (1.000, 1.000)       | 15.7 ( 0.0 )         | (15.6, 15.7)       |
> | dr.ridge.cv    | 0.068 ( 0.004 )      | (0.060, 0.077) | 0.800 ( 0.007 )             | (0.786, 0.813)       | 7.6 ( 0.1 )          | (7.5, 7.8)         |
> | dr.xgb.cv      | 0.000 ( 0.000 )      | (0.000, 0.001) | 0.976 ( 0.003 )             | (0.971, 0.981)       | 10.0 ( 0.0 )         | (10.0, 10.1)       |
> | dragon.nn      | 0.138 ( 0.006 )      | (0.126, 0.150) | 0.550 ( 0.009 )             | (0.533, 0.566)       | 4.6 ( 0.0 )          | (4.5, 4.7)         |
> | r.ridge.cv     | 0.164 ( 0.006 )      | (0.152, 0.177) | 0.513 ( 0.009 )             | (0.496, 0.529)       | 4.7 ( 0.1 )          | (4.6, 4.8)         |
> | r.xgb.cv       | 0.010 ( 0.002 )      | (0.006, 0.013) | 0.969 ( 0.003 )             | (0.963, 0.975)       | 10.6 ( 0.1 )         | (10.5, 10.7)       |
> | s.ext.ridge.cv | 0.107 ( 0.005 )      | (0.097, 0.118) | 0.829 ( 0.006 )             | (0.816, 0.842)       | 6.6 ( 0.1 )          | (6.5, 6.7)         |
> | s.ext.xgb.cv   | 0.082 ( 0.005 )      | (0.073, 0.091) | 0.743 ( 0.007 )             | (0.729, 0.758)       | 6.3 ( 0.1 )          | (6.2, 6.4)         |
> | s.ridge.cv     | 0.135 ( 0.006 )      | (0.124, 0.147) | 0.385 ( 0.008 )             | (0.369, 0.401)       | 3.3 ( 0.0 )          | (3.3, 3.4)         |
> | s.xgb.cv       | 0.166 ( 0.006 )      | (0.154, 0.178) | 0.223 ( 0.007 )             | (0.209, 0.237)       | 4.0 ( 0.0 )          | (3.9, 4.1)         |
> | t.ridge.cv     | 0.002 ( 0.001 )      | (0.001, 0.004) | 0.983 ( 0.002 )             | (0.978, 0.987)       | 11.0 ( 0.0 )         | (10.9, 11.1)       |
> | t.xgb.cv       | 0.000 ( 0.000 )      | (0.000, 0.000) | 0.999 ( 0.001 )             | (0.998, 1.000)       | 12.3 ( 0.0 )         | (12.2, 12.3)       |

---

> ### Author Response · Authors · 2024-11-26
>
> Note that, the result from 4:1 ratio looks different from Table 1 (and its criteo subsection), primarily because the evaluation dataset is only 25% of training; in comparison, the evaluation dataset in Table 1 is much larger. In fact, we estimate that 80% or 90% of the qhat computed from this smaller evaluation set falls outside of the 95% confidence interval computed from the full evaluation dataset. In other words, such evaluation is not trustworthy. When the evaluation dataset size is 25% of estimation (i.e., 25% of 1k~8k), we also see that only 22% of the degenerate cases (qhat > 0) have p-value less than or equal to 0.05 (compared to 94% from observational sampling evaluation).
>
> The estimation/evaluation split in observational sampling is different from a supervised learning setting. In supervised learning, a high (e.g., 4:1) train/test ratio reflect the trade-off between the risk of under-fitting a model and the need for accurate evaluation. It is reasonable to think that one can move data between train and test to slightly fine-tune the trade-off, as the reviewer alluded to.
>
> Such trade-off is absent from observational sampling, which starts from a RCT data and create “what-if” estimation dataset to stress test CATE models. It is perhaps easier to understand from the following angle: a scientist collects an observational dataset (e.g. with 2,000 samples) to fit some model and show promising results. They then proceed to run an RCT which gives them another 10,000 samples. The second 10k sample allows them to validate “accuracy” of the model fit from the first 2k sample post-mortem, but these 10k samples were not available at the moment of estimation.
>
> To summarize our discussion:
>
> 1. Everything else being equal, more estimation (aka training) data allows model to improve. This is consistent between supervised learning and CATE estimation paradigms. This is the benefit from large estimation dataset as pointed out by the reviewer, and we agree with that.
> 2. Everything else being equal, more evaluation (aka test) data allows model evaluation to improve. This is not necessarily a high priority for supervised learning, but a top one for observational sampling evaluation paradigm. This is because counterfactual ground truth is not available, fundamentally different from SL.
> 3. Unlike the supervised learning paradigm, observational sampling wants to both stress test CATE estimation algorithms when only small datasets are available, and evaluate such estimators, when counterfactual ground truth is not available. The different train/test split is a design choice.

---

> ### Author Response · Authors · 2024-11-26
> **Q3. follow up. standard error and confidence interval for Table 1**
>
> We added bootstrap standard deviation to Table 1; see below.
>
> | Model          | Win Rate Mean (SE)   | Win Rate 95% CI    | Degeneracy Rate Mean (SE)   | Degeneracy Rate 95% CI   | Avg Rank Mean (SE)   | Avg Rank 95% CI   |
> |:---------------|:---------------------|:---------------|:----------------------------|:---------------------|:---------------------|:-------------------|
> | cforest        | 0.005 ( 0.000 )      | (0.004, 0.006) | 0.760 ( 0.002 )             | (0.756, 0.764)       | 10.5 ( 0.0 )         | (10.4, 10.5)       |
> | dml.elastic    | 0.083 ( 0.001 )      | (0.080, 0.086) | 0.484 ( 0.002 )             | (0.479, 0.488)       | 5.6 ( 0.0 )          | (5.6, 5.6)         |
> | dml.lasso      | 0.080 ( 0.001 )      | (0.077, 0.082) | 0.484 ( 0.002 )             | (0.479, 0.489)       | 5.7 ( 0.0 )          | (5.6, 5.7)         |
> | dml.linear     | 0.000 ( 0.000 )      | (0.000, 0.000) | 0.942 ( 0.001 )             | (0.940, 0.945)       | 14.3 ( 0.0 )         | (14.3, 14.3)       |
> | dml.xgb        | 0.000 ( 0.000 )      | (0.000, 0.000) | 0.990 ( 0.000 )             | (0.989, 0.991)       | 15.9 ( 0.0 )         | (15.9, 15.9)       |
> | dr.ridge.cv    | 0.061 ( 0.001 )      | (0.058, 0.063) | 0.592 ( 0.002 )             | (0.588, 0.597)       | 7.1 ( 0.0 )          | (7.1, 7.2)         |
> | dr.xgb.cv      | 0.012 ( 0.001 )      | (0.011, 0.013) | 0.714 ( 0.002 )             | (0.710, 0.719)       | 9.9 ( 0.0 )          | (9.8, 9.9)         |
> | dragon.nn      | 0.121 ( 0.002 )      | (0.118, 0.124) | 0.438 ( 0.002 )             | (0.433, 0.443)       | 5.1 ( 0.0 )          | (5.1, 5.2)         |
> | r.ridge.cv     | 0.062 ( 0.001 )      | (0.059, 0.064) | 0.612 ( 0.002 )             | (0.608, 0.617)       | 8.7 ( 0.0 )          | (8.7, 8.8)         |
> | r.xgb.cv       | 0.003 ( 0.000 )      | (0.002, 0.003) | 0.846 ( 0.002 )             | (0.842, 0.849)       | 12.4 ( 0.0 )         | (12.4, 12.4)       |
> | s.ext.ridge.cv | 0.111 ( 0.002 )      | (0.108, 0.114) | 0.504 ( 0.002 )             | (0.499, 0.509)       | 5.6 ( 0.0 )          | (5.6, 5.6)         |
> | s.ext.xgb.cv   | 0.064 ( 0.001 )      | (0.062, 0.067) | 0.518 ( 0.002 )             | (0.514, 0.523)       | 6.9 ( 0.0 )          | (6.9, 6.9)         |
> | s.ridge.cv     | 0.129 ( 0.002 )      | (0.126, 0.133) | 0.312 ( 0.002 )             | (0.307, 0.316)       | 4.2 ( 0.0 )          | (4.2, 4.2)         |
> | s.xgb.cv       | 0.255 ( 0.002 )      | (0.251, 0.259) | 0.063 ( 0.001 )             | (0.061, 0.066)       | 4.4 ( 0.0 )          | (4.4, 4.4)         |
> | t.ridge.cv     | 0.043 ( 0.001 )      | (0.041, 0.045) | 0.641 ( 0.002 )             | (0.636, 0.646)       | 8.2 ( 0.0 )          | (8.2, 8.3)         |
> | t.xgb.cv       | 0.005 ( 0.000 )      | (0.004, 0.005) | 0.767 ( 0.002 )             | (0.763, 0.771)       | 11.4 ( 0.0 )         | (11.4, 11.4)       |

---

### Official Review · Reviewer_tCgC · 2024-11-09

**Soundness:** 2
**Presentation:** 3
**Contribution:** 2
**Rating:** 5
**Confidence:** 3

**Summary:**

The paper proposes a new statistical parameter to evaluate the performance of a method to estimate CATE. Then the paper shows that this parameter can be consistently estimated. Finally the paper demonstrates that this parameter can be used to select estimation methods for CATE.

**Strengths:**

- The statistical parameter proposed in the paper is simple to implement and makes intuitive sense.
- Theoretical guarantees are provided.
- The paper compares many estimation methods for CATE in the empirical application.

**Weaknesses:**

- The paper largely overclaims the number of datasets. Essentially, there are only 12 unique datasets and the paper uses a bunch of sampling methods to create many more variants of the original 12 datasets.
- The findings in the first paragraph of the abstract have been known for a long time, and they are not unexpected, as claimed in the paper. The estimation of CATE is known to be very noisy, and that's why the causal inference literature has primarily focused on the estimation of ATE (but not CATE) for decades.
- The clarity and rigor of the paper need to be improved. For example, the term "CATE model" is a bit awkward and is rarely used in the literature. CATE is an estimand commonly defined based on potential outcomes. When "CATE model" is used, it is unclear whether this term refers to an outcome model to define CATE or an estimation method for CATE.

**Questions:**

Please address the weaknesses above.

---

> ### Author Response · Authors · 2024-11-20
> **rebuttal**
>
> W1. The paper largely overclaims the number of datasets. Essentially, there are only 12 unique datasets and the paper uses a bunch of sampling methods to create many more variants of the original 12 datasets.
>
> Response. Thank you for highlighting this important point. We agree that the distinction between the 12 unique datasets and the 43,200 sampled variants could be more clearly articulated. In the next revision, we will ensure this distinction is explicitly clarified in the main text. Specifically, we will emphasize that the 43,200 variants are generated through diverse sampling strategies applied to the 12 unique datasets, and we will avoid overstating their independence.
>
> Meanwhile, we would like to highlight that this clarification does not diminish the contribution of our work. The inclusion of 12 large RCTs represents a significant improvement over traditional causal inference studies, which often benchmark algorithms on only a few datasets (e.g., IHDP or JOBS). By leveraging these 12 diverse datasets, our work offers a more robust and comprehensive benchmarking framework. See further discussion on dataset selection in our response to Reviewer vwxd.
>
> W2. The findings in the first paragraph of the abstract have been known for a long time, and they are not unexpected, as claimed in the paper. The estimation of CATE is known to be very noisy, and that's why the causal inference literature has primarily focused on the estimation of ATE (but not CATE) for decades.
>
> Response. We appreciate the reviewer’s observation regarding the noisiness of CATE estimates and its historical context in causal inference. However, we respectfully note that our findings extend beyond this general understanding. While experts in causal inference may recognize the challenges of CATE estimation, the broader scientific community—particularly practitioners outside this domain—may not fully grasp the risks associated with applying CATE models. This is particularly relevant as these models, supported by accessible tools like EconML and DoubleML, are increasingly used in critical decision-making across diverse fields. Our work quantifies these risks by demonstrating that state-of-the-art CATE methods can fail to outperform simple baselines in real-world heterogeneous datasets, highlighting a gap not addressed in prior literature. If the reviewer is aware of specific studies on this topic, we would greatly appreciate the opportunity to incorporate them into our discussion.
>
> W3. The clarity and rigor of the paper need to be improved. For example, the term "CATE model" is a bit awkward and is rarely used in the literature. CATE is an estimand commonly defined based on potential outcomes. When "CATE model" is used, it is unclear whether this term refers to an outcome model to define CATE or an estimation method for CATE.
>
> Response. Thank you for pointing out the potential ambiguity in our use of the term "CATE model." By CATE model we mean Conditional Average Treatment Effect estimation models. We adopted the term "CATE model" primarily for brevity. However, we recognize the importance of clarity and rigor in our terminology. In the next revision, we will ensure that the term is either replaced with a more precise alternative or explicitly defined at its first mention to avoid misunderstanding. We appreciate the reviewer bringing this to our attention and will make every effort to improve the clarity of the paper as a whole.

---

> ### Author Response · Authors · 2024-11-25
>
> Dear Reviewer tCgC,
>
> Thanks again for your efforts in reviewing our paper. This is a gentle reminder that the author discussion deadline is approaching very soon (11/26). We'd like to know if our response has addressed your questions. We are happy to discuss more if there are any further questions.
>
> Thanks,
>
> Authors

---

### Meta-Review · Area_Chair_XfBj · 2024-12-21

**Metareview:**

The paper proposes a statistical framework (Q-metric) to evaluate CATE (Conditional Average Treatment Effect) estimation methods in observational studies without relying on synthetic data or untestable modeling assumptions

Strengths:

+ Addresses an important problem in causal inference by providing a more reliable way to evaluate CATE estimation methods

Weaknesses:

+ The benchmark is heavily skewed towards one source (8 out of 12 tasks from the general social survey), potentially limiting diversity

+ Training sample sizes are unusually small compared to evaluation sizes, which could explain poor performance of some methods and needs further investigation

**Additional Comments On Reviewer Discussion:**

The reviewers have mixed opinions; while some agree that the paper addresses an important problem, the study can be more thorough.

---

### Decision · Program_Chairs · 2025-01-22

Accept (Poster)